# DIG2DIG: DIG INTO DIFFUSION INFORMATION GAINS FOR IMAGE FUSION

## ABSTRACT

Image fusion integrates complementary information from multiple sources to generate more informative results. Recently, the diffusion model, which demonstrates unprecedented generative potential, has been explored in the context of image fusion. During diffusion model generation, information emerges at unequal rates, so the fusion should dynamically weight the source modalities. To address this issue, we reveal a significant spatio-temporal imbalance in image denoising; specifically, the diffusion model produces dynamic information gains in different image regions with denoising steps. Based on this observation, we dive into the Diffusion Information Gains (DIG) and theoretically derive a diffusion-based dynamic image fusion framework that provably reduces its upper bound of the generalization error. Accordingly, we introduce diffusion information gains to quantify the information contribution of each modality at different denoising steps, thereby providing dynamic guidance during the fusion process. Experiments on multiple fusion scenarios confirm that our method outperforms existing diffusion-based approaches in terms of both fusion quality and inference efficiency.

## 1 INTRODUCTION

Image fusion integrates complementary information from various sources to generate informative fused images with high visual quality (Kaur et al., 2021; Liang et al., 2022), thus substantially improving the performance of downstream vision tasks through enhanced scene representations and enriched visual perception. Image fusion can be mainly grouped into three categories: multi-modal image fusion, multi-exposure image fusion, and multi-focus image fusion. Multi-modal image fusion (MMF) mainly encompasses Visible-Infrared Image Fusion (VIF) and Medical Image Fusion (MIF) tasks. VIF aims to combine the highlighted thermal targets, especially under extreme conditions, in infrared images and the textural details contributed by visible images (Zhang et al., 2020; Ma et al., 2021). MIF incorporates the active regions of various medical imaging modalities, thereby contributing to diagnostic capabilities (Basu et al., 2024). Different from MMF, MEF (Cao et al., 2025a) reconciles the disparity between high- and low-dynamic range images in visual modality, ensuring harmonious lighting appearance, while MFF (Kaur et al., 2021) produces all-in-focus images by blending multiple images captured at different focal depths.

Deep learning-based image fusion techniques, such as CNNs, GANs, and Transformers, have outperformed traditional methods; however, their generative capacity usually restricts the detail and realism of the fused images.Later, diffusion models have emerged as a powerful generative model (Dhariwal & Nichol, 2021), demonstrating unprecedented potential in image fusion (Zhao et al., 2023b). Some works aim to generate fused images by extracting effective feature representations or incorporating diverse constraints (Cao et al., 2025b) into diffusion models. However, they often employ fixed multi-modal fusion guidance to the denoising diffusion steps, overlooking the structural dynamism of denoising and failing to produce qualified fusion results in complex scenarios with changing image quality, highlighting the importance of performing dynamic fusion.

Recently, some studies (Tang et al., 2022b) have explored the dynamism in image fusion. For instance, MoE-Fusion (Cao et al., 2023) introduced a dynamic fusion CNN framework with a mixture of experts model, adaptively extracting comprehensive features from diverse modalities. Text-IF (Yi et al., 2024) pioneered the dynamic controllability of image fusion utilizing various text guidance. Furthermore, TTD (Cao et al., 2025a) first studied the theoretical foundation of dynamic image fu-

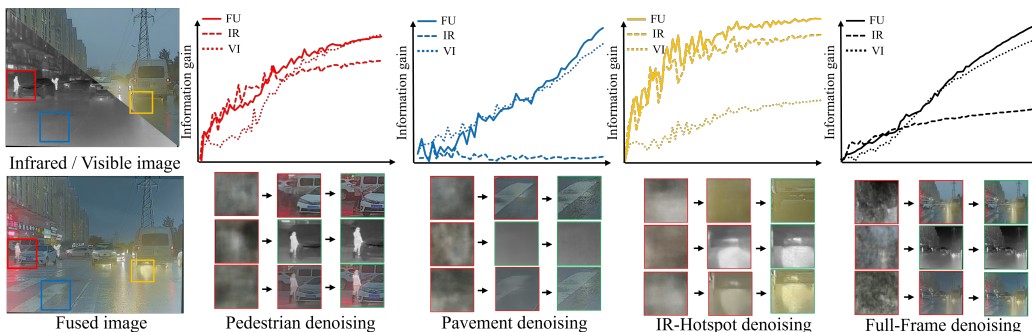

Figure 1: The spatio-temporal imbalance of diffusion. We observe that diffusion models restore different regions of an image at unequal rates. Throughout the denoising process, not only do information discrepancies exist between different modalities, but such spatio-temporal imbalance of information gain also persists across various regions of the image. For clarity, the "information gain" in this figure refers to the $\ell_2$ difference between the denoised image at timestep $t$ and the image at $t = T/2$ during the denoising process, which is slightly different from the DIG defined later.

sion during inference. Despite their notable empirical performance, these dynamic-oriented fusion methods are mainly limited to CNN-based frameworks, and few works dive into the dynamism of diffusion modeling. Furthermore, many of these techniques fundamentally rely on heuristic approaches that lack theoretical validation and clear interpretability, leading to unstable fusion results.

To address these issues, we reveal the objective of image fusion and dig into diffusion information gains for denoising image fusion with theoretical guarantee. Intuitively, image fusion aims to maximize information retention across all modalities (Li et al., 2017; Liu et al., 2024). Given that multi-source images jointly determine the fusion result at each step of the diffusion process, the more incremental information of one modality gains at the denoising step contributes more to the overall fusion result, and vice versa. As illustrated in Figure 1, each modality involved in the fusion process demonstrates a distinct denoising pace within the diffusion framework. Specifically, regions with salient structures converge during the early denoising steps, whereas texture-rich details are recovered only in later iterations. This spatio-temporal heterogeneity reveals that the information contribution of each modality is uneven across denoising steps. This highlights the dynamic guidance strength of different modalities to effectively preserve and integrate the complementary information offered by each modality. The information recovery speed of the fused image also shows a similar pattern. Building on this insight, we revisit the generalized form of denoising image fusion from the perspective of generalization error, and for the first time prove that the key to enhancing generalization in denoising diffusion fusion lies in the positive correlation of the modality fusion weight and the residual modality information. Consequently, we derive the Diffusion Information Gains (DIG) as the dynamic fusion weight, which quantifies the contribution of each modality, theoretically enhancing the generalization of the image fusion model, and dynamically highlights the *informative* regions of different sources. Extensive experiments on multiple datasets and diverse image fusion tasks demonstrate our superiority in terms of fusion quality and efficiency.

- We introduce Dig2DIG, a simple yet effective dynamic denoising fusion framework. By taking DIG as the dynamic fusion weight, our approach enhances the generalization of the image fusion model while adaptively integrating informative regions from each source.

- We theoretically prove that dynamic denoising image fusion outperforms static denoising fusion from the generalization error perspective provably, the key of which lies in the positive covariance between the fusion weight and the residual modality information.

- We compute per-modality, per-region Diffusion Information Gain (DIG) at each reverse-denoising step and use these gains as fusion weights to inject only the currently informative regions from each modality; information-deficient regions are automatically downweighted. Guided by DIG, steps with less information gain can be skipped, saving 70% of time consumption. This differs from conventional fusion that enforces proximity to all inputs at every step with fixed or heuristic weights, which can bias the fused result toward weak modalities and uninformative regions.

## 2 RELATED WORKS

### 2.1 IMAGE FUSION

Image fusion aims to integrate complementary information from various sources, such as visible-infrared images, multi-exposure images, and multi-focus images, into a single fused image, thereby improving its visual appearance and downstream task performance. Traditional approaches often employ wavelet transforms, multi-scale pyramids, or sparse representations to perform fusion in a transform domain (Wang et al., 2005), while deep learning-based methods (e.g., CNN-, GAN-, or Transformer-based models) learn end-to-end fusion mappings directly in data-driven scheme, which significant enhances the fusion quality compared to traditional methods (Archana & Jeevaraj, 2024). Recently, several fusion approaches based on diffusion models have emerged. For example, DDFM (Zhao et al., 2023b) frames the fusion problem as conditional generation within a DDPM framework, utilizing an unconditional pretrained model and expectation-maximization (EM) inference to generate high-quality fused images. CCF (Cao et al., 2025b) introduces controllable constraints into a pretrained DDPM, allowing the fusion process to adapt to various requirements at each reverse diffusion step, thereby enhancing versatility and controllability. Moreover, Text-IF (Yi et al., 2024) incorporates textual semantic guidance into the fusion process, enabling joint image restoration and fusion interactively. Although some studies have explored dynamic image fusion, the absence of theoretical foundations may yield unstable and unreliable performance in practice.

### 2.2 CONDITIONAL GUIDANCE

Conditional guidance (Ho & Salimans, 2022) in diffusion models typically involves injecting additional priors (such as multi-modal features or textual semantics) at each denoising step, providing a flexible way to steer the final generation or editing outcome. Existing studies (Tumanyan et al., 2023; Xu et al., 2024) have shown that the guidance on different denoising stages can produce substantially different results, highlighting the importance of dynamic guidance within denoising steps (Cao et al., 2025b). Recently, some dynamic fusion methods were proposed not only for image fusion, but also for more general multi-modal learning. For instance, Xue & Marculescu (2023) employ a Mixture-of-Experts mechanism to integrate multiple experts for multimodal fusion. Han et al. (2022) assign the Evidence-driven dynamic weights at the decision level to obtain the trusted fusion decisions, and Zhang et al. (2023) explored the advantages of dynamic fusion and further proposed uncertainty-based fusion weights to enhance the robustness of multimodal learning. Although these methods validated the effectiveness of performing dynamic learning, few works reveal the dynamism of conditional guidance in diffusion-based image fusion. Most existing methods often assume equal importance for all modalities, overlooking the variations in the information retained by each modality at different denoising stages. This highlights the need for a dynamic guidance mechanism capable of quantifying and utilizing the information gain of each modality.

## 3 METHOD

In this paper, we dig into the diffusion information gains and propose a denoising-oriented dynamic image fusion framework. We proceed to reveal the DDPM (Song et al., 2020), the forward diffusion process gradually adds noise to a clean sample $x_0$ until it becomes nearly Gaussian as $x_t = \sqrt{\bar{\alpha}_t}\, x_0 + \sqrt{1 - \bar{\alpha}_t}\, \epsilon$, $\epsilon \sim \mathcal{N}(0, I)$, where $\alpha_t = 1 - \beta_t$, $\bar{\alpha}_t = \prod_{i=1}^{t} \alpha_i$, and $\{\beta_t\}$ is a predefined variance schedule. During inference, the noise $x_T$ iteratively denoises via the reverse update:

$$x_{t-1} = \frac{1}{\sqrt{\alpha_t}} \left( x_t - \frac{1 - \alpha_t}{\sqrt{1 - \bar{\alpha}_t}}\, \epsilon_\theta(x_t, t) \right) + \sigma(t)\, z, \tag{1}$$

where $\sigma^2(t) = (1 - \alpha_t)(1 - \bar{\alpha}_{t-1})/(1 - \bar{\alpha}_t)$, $\epsilon_\theta(\cdot)$ is the network's noise prediction, and $z \sim \mathcal{N}(0, I)$.

### 3.1 MULTIMODAL GUIDANCE

For the forward process, if $\epsilon_\theta$ accurately reflects the noise in $x_t$, the gradient of $\log p(x_t)$ can be approximated by the score function as $\nabla_{x_t} \log p(x_t) = -\frac{\epsilon_\theta(x_t, t)}{\sqrt{1 - \bar{\alpha}_t}}$. Comprehensively, the final update

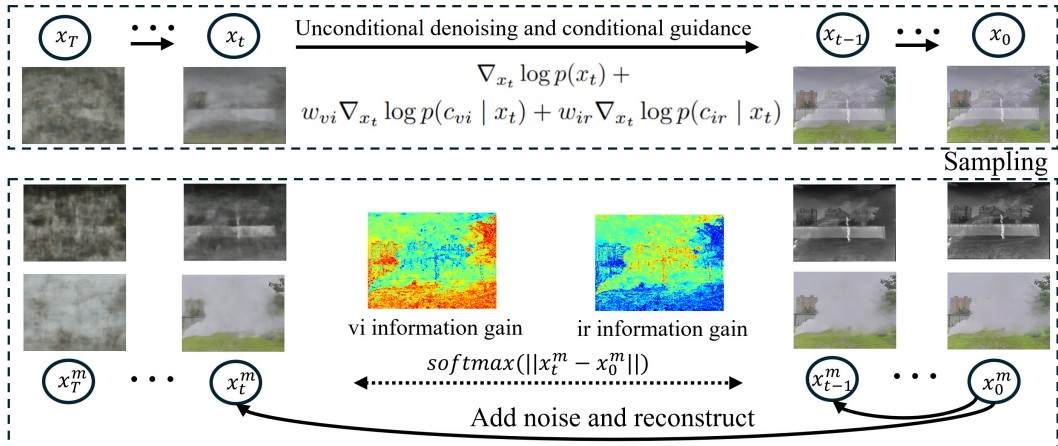

Figure 2: The framework of our Dig2DIG. Deriving from generalization theory, we find that the key to tightening the fusion generalization bound is to ensure that the guidance weight assigned to each modality is positively correlated with the amount of residual information from that modality that has not yet been incorporated into the current fused image. To achieve this, we utilize DIG to estimate this residual information, providing theoretical guidance for reducing generalization error.

step is given as follows. More details are presented in Appendix A and C.

$$x_{t-1} \approx \frac{1}{\sqrt{\alpha_t}} x_t + \underbrace{\sigma(t)\, z}_{\text{Noise}} + \underbrace{\frac{1}{\sqrt{\alpha_t}}(1-\alpha_t)\nabla_{x_t}\log p(x_t)}_{\text{Unconditional Guidance}} + \underbrace{\frac{1}{\sqrt{\alpha_t}}(1-\alpha_t)\sum_{k=1}^{K} w_k \nabla_{x_t}\log p(c_k \mid x_t)}_{\text{Multimodal Guidance}}$$

(2)

## 3.2 Generalization Error Upper Bound

Given images $\{c_k\}_{k=1}^{K}$, $c_k \in \mathbb{R}^{H \times W \times N}$ from $K$ sources, the input image combination can be represented as $c = \{c_1, \ldots, c_K\}$. In the diffusion model, we use $x_t$ to denote the image in the $t$ step of the reverse diffusion process, and the final denoised (fused) result is $x_0 \in \mathbb{R}^{H \times W \times N}$. The overall denoising operator of the diffusion model can be denoted as $F$, i.e., $x_0 = F(c)$. We write $I_{k,t} = \Delta_I(c_k, x_t, x^*(c)), k = 1, \ldots, K$, to denote the residual information of modality $k$ that has not yet been incorporated into $x_t$; When $x^*(c)$ is available, we instantiate $\Delta_I$ as the modality-$k$ projection energy of the residual, i.e., $I_{k,t} \triangleq \|\Pi_k(x^*(c) - x_t)\|^2$.

Let $c \sim D$ denote the multimodal input, and let $z = \{z_t\}_{t=1}^{T}$ be the algorithmic sampling noise in the reverse diffusion, where each $z_t \sim \mathcal{N}(0, I)$ is independent of $(c, x_t)$.

Let $x^*(c)$ represents the *ideal* fused image conditioned on the multimodal input $c$, and let $\zeta(\cdot)$ be a loss function that measures the discrepancy between a fused image and the ideal image. Assume that $\zeta(\cdot)$ is an $L$-Lipschitz function (i.e., $|\zeta(u) - \zeta(v)| \le L \|u - v\|$ for a suitable norm; in image tasks $\zeta$ is often chosen as the $\ell_1$ or $\ell_2$ distance) , under these assumptions, for any unseen data $c \sim D$, we define the Generalization Error as follows:

$$\text{GError}(F) = \mathbb{E}_{c,z}\big[\zeta\big(F(c), x^*(c)\big)\big].$$

(3)

Here, $x^*(c)$ denotes the ideal fused image tailored to the input $c$, which reflects the optimal fusion result that we aim to approximate. This expectation quantifies the mean discrepancy between the fused output $F(c)$ and the ideal fused image $x^*(c)$, evaluated on the actual data distribution $D$ and averaged over the internal sampling noise $z$. A smaller Generalization Error indicates that the model performs better in terms of fusion accuracy on unseen multimodal data.

**Theorem 1** *For a multi-source image-fusion operator $F$ that employs diffusion-based conditional guidance, the Generalization Error (GError) can be decomposed into (i) a linear combination of*

covariance terms, *each capturing the interaction between the guidance weight $w_k$ and the* residual information $I_{k,t}$ *of modality $k$ that has not yet been incorporated into the current fused image $x_t$, and (ii) a constant term that is independent of both the weights $\{w_k\}$ and the input data $c$, given that $\sum_{k=1}^{K} w_k = 1$. The detailed proof and detailed explanation of $I_{k,t}$ are provided in Appendix B.*

$$\text{GError}(F) \leq C - \sum_{t=1}^{T} \sum_{k=1}^{K} A_{k,t} \, \text{Cov}(w_k, I_{k,t}), \tag{4}$$

where $C$ and the coefficients $A_{k,t}$ are constants that do not depend on $w_k$ or $c$. $I_{k,t}$ quantifies the amount of residual information from modality $k$ that has not yet been fused into $x_t$. A larger $I_{k,t}$ indicates that modality $k$ still contains substantial information that can reduce the discrepancy between $x_t$ and the ideal fused image $x^*(c)$.

In practice, the ideal fused image $x^*(c)$ and $I_{k,t}$ are unobservable; consequently, the covariance $\text{Cov}(w_k, I_{k,t})$ cannot be evaluated. A common workaround in diffusion-based fusion systems is to assign uniform weights, implicitly assuming equal importance for all modalities. However, empirical studies (Du et al., 2023; Dinh et al., 2023) have shown that the rate at which information is restored during the reverse process depends on spatial frequency and the current time step, leading different modalities to exhibit heterogeneous informational contributions at a given $x_t$.

Spectral analyses in diffusion (Lee et al., 2025) demonstrate that low-frequency content is synthesized early whereas high-frequency details emerge later in diffusion models. Because a fused image must simultaneously reconstruct the complete frequency content of all source modalities (Wang et al., 2024), its step-wise information restoration naturally inherits the generation speed of each modality. The results in Figure 1 also support this view. Therefore, the information gain obtained at time step $t$ from a single-modal reconstruction serves as a reasonable proxy for the amount of modality-$k$ information that still remains to be incorporated into the fused image, which is $I_{k,t}$.

## 3.3 DYNAMIC FUSION WITH DIFFUSION INFORMATION GAINS

Accordingly, we introduce the concept of *Diffusion Information Gains* (DIG), which quantifies how much residual discrepancy still separates a single-modal reconstruction from its clean target at each reverse-diffusion step. Specifically, for an individual modality $c_k$, let $c_k^t$ denote its noisy observation at timestep $t$, and let $\hat{c}_k^t$ be the corresponding one-step denoised result. We define

$$\text{DIG}_k(t) \;=\; l(\hat{c}_k^t, \, c_k), \tag{5}$$

where $l(\cdot, \cdot)$ is any image-to-image discrepancy measure (e.g., the $\ell_2$ distance). A larger $\text{DIG}_k(t)$ means that, at step $t$, the current single-modal reconstruction is still far from its clean counterpart $c_k$, indicating that modality $k$ can potentially supply a larger amount of information to the ongoing fusion process. Therefore, we refer to this quantity as DIG.

Following the standard diffusion framework, the noisy image $c_k^t$ at timestep $t$ is generated by

$$c_k^t \;=\; \sqrt{\bar{\alpha}_t}\, c_k \;+\; \sqrt{1 - \bar{\alpha}_t}\, \epsilon, \quad \epsilon \sim \mathcal{N}(0, I), \tag{6}$$

where $\bar{\alpha}_t$ controls the noise level. The denoised result $\hat{c}_k^t$ is obtained from $c_k^t$ via the estimated noise:

$$\hat{c}_k^t \;=\; \frac{1}{\sqrt{\bar{\alpha}_t}}\Big(c_k^t - \sqrt{1 - \bar{\alpha}_t}\, \epsilon_\theta\big(c_k^t, t\big)\Big). \tag{7}$$

Recalling the upper bound of the Generalization Error in equation 4, the not-yet-fused information $I_{k,t}$ can be characterized by the status of the single-modal reconstruction at step $t$. Because a larger $\text{DIG}_k(t)$ indicates that more residual information from modality $k$ remains to be incorporated, $\text{DIG}_k(t)$ serves as a practical, observable proxy for the latent residual-information term $I_{k,t}$ discussed earlier.

Given $\text{DIG}_k(t)$ for each modality $c_k$, we propose to dynamically weight the guidance contributions to the fused image based on their diffusion information gains. At each denosing step $t$, the weights $\{w_k\}$ is computed by normalizing the DIG values across the modalities (e.g., softmax):

$$w_k(t) \;=\; \frac{\exp\big(\text{DIG}_k(t)\big)}{\sum_{j=1}^{K} \exp\big(\text{DIG}_j(t)\big)}. \tag{8}$$

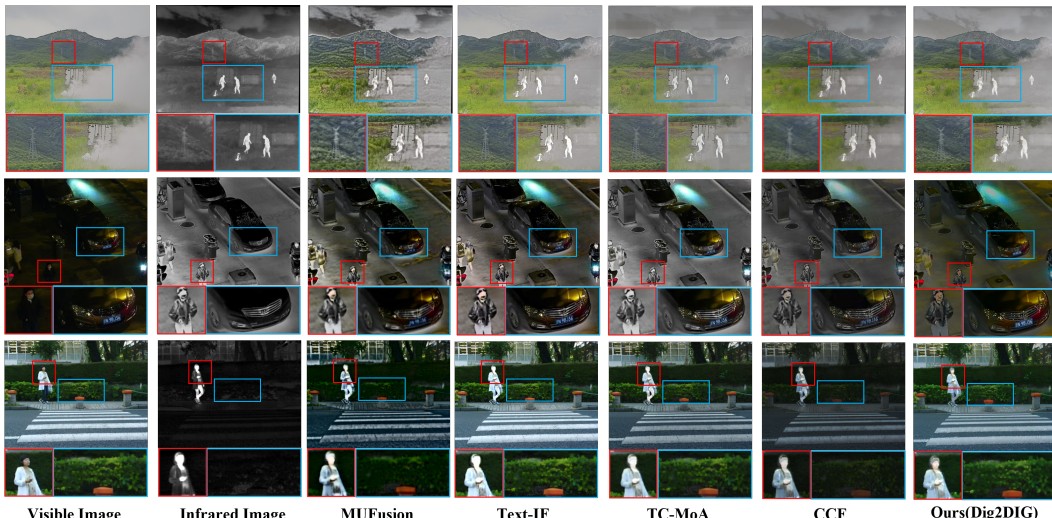

| Visible Image | Infrared Image | MUFusion | Text-IF | TC-MoA | CCF | Ours(Dig2DIG) |

Figure 3: Qualitative comparisons of our method on M3FD, LLVIP, and MSRS datasets.

Table 1: Quantitative comparisons on LLVIP, M3FD, and MSRS.

| Method | LLVIP Dataset | | | | | | M3FD Dataset | | | | | | MSRS Dataset | | | | | |
|---|---|---|---|---|---|---|---|---|---|---|---|---|---|---|---|---|---|---|
| | PSNR↑ | SSIM↑ | MSE↓ | Nabf↓ | CC↑ | LPIPS↓ | PSNR↑ | SSIM↑ | MSE↓ | Nabf↓ | CC↑ | LPIPS↓ | PSNR↑ | SSIM↑ | MSE↓ | Nabf↓ | CC↑ | LPIPS↓ |
| SwinFusion | 32.33 | 0.81 | 2845 | 0.023 | 0.67 | 0.321 | 31.73 | 1.40 | 3853 | 0.021 | 0.51 | 0.289 | 39.34 | 1.41 | 1755 | 0.002 | 0.59 | 0.298 |
| DIVFusion | 21.60 | 0.82 | 6450 | 0.044 | 0.66 | 0.350 | 26.19 | 1.20 | 4099 | 0.083 | 0.51 | 0.377 | 18.49 | 0.69 | 10054 | 0.100 | 0.52 | 0.462 |
| MoE-Fusion | 31.70 | 1.12 | 2402 | 0.034 | 0.69 | 0.324 | 33.15 | 1.37 | 3462 | 0.012 | 0.47 | 0.303 | 38.21 | 1.35 | 2637 | 0.030 | 0.60 | 0.298 |
| MUFusion | 31.64 | 1.10 | 2069 | 0.030 | 0.65 | 0.320 | 29.82 | 1.29 | 2733 | 0.071 | 0.50 | 0.349 | 36.02 | 1.25 | 1701 | 0.037 | 0.62 | 0.370 |
| CDDFuse | 32.13 | 1.18 | 2545 | 0.016 | 0.67 | 0.335 | 31.75 | 1.40 | 3715 | 0.030 | 0.52 | 0.278 | 37.76 | 1.30 | 2485 | 0.022 | 0.59 | 0.335 |
| DDFM | 36.10 | 1.18 | 2056 | 0.004 | 0.67 | 0.310 | 30.87 | 1.40 | 2221 | 0.007 | 0.56 | 0.303 | 38.19 | 1.39 | 1367 | 0.004 | 0.66 | 0.287 |
| Text-IF | 31.22 | 1.18 | 2460 | 0.031 | 0.69 | 0.312 | 34.01 | 1.39 | 3470 | 0.037 | 0.48 | 0.277 | 41.93 | 1.37 | 2494 | 0.027 | 0.60 | 0.298 |
| TC-MoA | 33.00 | 1.20 | 2790 | 0.017 | 0.67 | 0.332 | 31.07 | 1.40 | 2516 | 0.011 | 0.53 | 0.289 | 37.73 | 1.40 | 1640 | 0.005 | 0.62 | 0.293 |
| CCF | 33.12 | 1.22 | 1658 | 0.006 | 0.70 | 0.334 | 31.51 | 1.40 | 2271 | 0.010 | 0.56 | 0.291 | 38.00 | 1.38 | 1410 | 0.006 | 0.64 | 0.319 |
| DCEvo | 32.42 | 1.15 | 2575 | 0.014 | 0.66 | 0.321 | 31.45 | 1.40 | 3812 | 0.071 | 0.50 | 0.290 | 38.00 | 1.41 | 2464 | 0.039 | 0.60 | 0.299 |
| MMDRFuse | 33.28 | 1.20 | 2159 | 0.025 | 0.69 | 0.302 | 31.51 | 1.40 | 3508 | 0.014 | 0.54 | 0.301 | 39.01 | 1.40 | 2199 | 0.190 | 0.60 | 0.323 |
| LFDT | 33.31 | 1.20 | 2534 | 0.019 | 0.66 | 0.302 | 30.54 | 1.39 | 3714 | 0.027 | 0.47 | 0.318 | 38.97 | 1.41 | 2525 | 0.031 | 0.60 | 0.306 |
| Dig2DIG | 33.74 | 1.23 | 1464 | 0.001 | 0.73 | 0.298 | 31.83 | 1.41 | 2216 | 0.009 | 0.57 | 0.287 | 39.07 | 1.42 | 1366 | 0.001 | 0.63 | 0.282 |

By incorporating DIG-based weights, the fused result more accurately reflects the relative contributions of each modality at each timestep, ultimately leading to a lower fusion error and better generalization performance.

# 4 EXPERIMENTS

## 4.1 EXPERIMENTAL SETTING

**Datasets.** In our experiments, we evaluate the proposed method on three key image fusion tasks: Visible-Infrared Image Fusion (VIF), Multi-Focus Fusion (MFF), and Multi-Exposure Fusion (MEF). For VIF, we use the LLVIP (Jia et al., 2021), M3FD (Liu et al., 2022), and MSRS (Tang et al., 2022a) datasets, each providing paired visible and infrared images under a variety of scenarios. In the MFF task, we adopt the MFFW dataset (Zhang, 2021a) to merge images that focus on different regions into a single, fully focused output. For the MEF task, we employ the MEFB dataset (Zhang, 2021b) to assess the performance of combining images captured at various exposure levels.

**Implementation Details.** Our approach is built upon a single pre-trained diffusion model (Dhariwal & Nichol, 2021), and it does not require any additional training or fine-tuning. The same pretrained network is reused without any modification to denoise every modality across all fusion tasks, thereby avoiding modality-specific supervision. See Appendix C for more details.

**Evaluation Metrics.** We evaluate fusion quality using both qualitative and quantitative approaches. Qualitative assessment relies on subjective visual inspection, focusing on clear textures and natural color representation. For the Visible-Infrared Image Fusion task, we use Peak Signal-to-Noise Ratio (PSNR), Structural Similarity (SSIM), Mean Squared Error (MSE), Noise Amplification (Nabf),

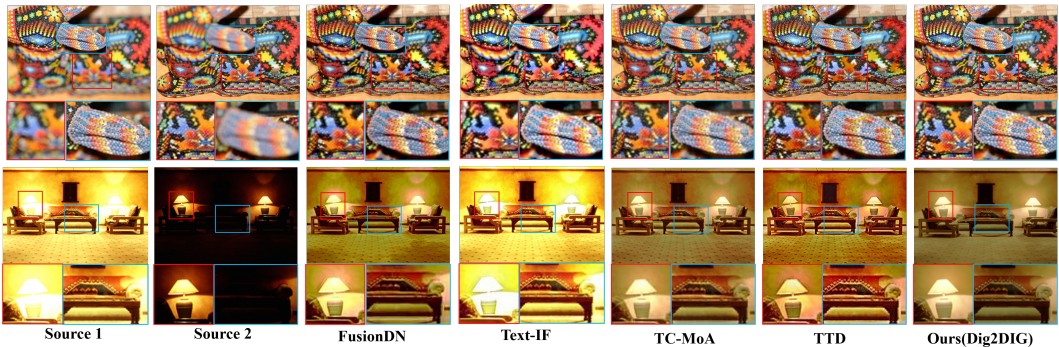

Figure 4: Qualitative comparisons of our method on MFFW Dataset and MEFB Dataset.

Table 2: Performance comparison on MFFW and MEFB datasets.

| Method | MFFW Dataset | | | | | | MEFB Dataset | | | | | |
|---|---|---|---|---|---|---|---|---|---|---|---|---|
| | SD↑ | EI↑ | EN↑ | AG↑ | SF↑ | MI↑ | SD↑ | EI↑ | EN↑ | AG↑ | SF↑ | MI↑ |
| FusionDN | 66.59 | 17.20 | 7.45 | 6.74 | 22.27 | 3.37 | 61.50 | 19.55 | 7.29 | 7.56 | 21.05 | 3.47 |
| U2Fusion | 64.88 | 11.97 | 6.93 | 5.56 | 18.74 | 3.25 | 67.83 | 19.54 | 7.37 | 8.08 | 22.19 | 3.38 |
| DeFusion | 52.75 | 10.60 | 6.80 | 4.32 | 14.12 | 2.92 | 54.75 | 12.55 | 7.28 | 4.76 | 12.72 | 3.89 |
| DDFM | 67.30 | 14.32 | 7.51 | 3.82 | 13.40 | 5.71 | 56.34 | 11.95 | 7.30 | 4.47 | 12.21 | 8.49 |
| Text-IF | 62.51 | 12.73 | 6.39 | 4.82 | 17.26 | 3.41 | 66.27 | 20.01 | 7.37 | 7.72 | 21.58 | 3.30 |
| TC-MoA | 50.27 | 12.18 | 7.07 | 4.82 | 15.64 | 3.39 | 57.55 | 17.65 | 7.35 | 6.95 | 20.67 | 4.45 |
| TTD | 52.86 | 15.94 | 7.10 | 6.38 | 21.99 | 4.54 | 54.22 | 19.10 | 7.39 | 7.70 | 23.51 | 3.59 |
| CCF | 69.71 | 14.85 | 7.75 | 6.70 | 21.49 | 4.23 | 71.01 | 19.99 | 7.35 | 8.03 | 22.71 | 3.93 |
| Dig2DIG | 72.95 | 16.64 | 7.87 | 6.75 | 22.60 | 5.97 | 75.05 | 20.21 | 7.38 | 8.10 | 23.60 | 6.87 |

Correlation Coefficient (CC), and Learned Perceptual Image Patch Similarity (LPIPS). For the MFF and MEF tasks, we employ Standard Deviation (SD), Edge Intensity (EI), Entropy (EN), Average Gradient (AG), Spatial Frequency (SF), and Mutual Information (MI). Following DDFM/CCF, we compute SSIM as the sum over two references, this keeps comparisons protocol-consistent.

## 4.2 COMPARISON ON VISIBLE-INFRARED IMAGE FUSION

For VIF, we compare our method with the state-of-the-art methods: SwinFusion (Ma et al., 2022), DIVFusion (Tang et al., 2023), MOEFusion (Cao et al., 2023), MUFusion (Cheng et al., 2023), CDDFuse (Zhao et al., 2023a), DDFM (Zhao et al., 2023b), Text-IF (Yi et al., 2024), TC-MoA Zhu et al. (2024), CCF (Cao et al., 2025b), DCEvo (Liu et al., 2025), MMDRFuse (Deng et al., 2024), and LFDT-Fusion (Yang et al., 2025).

**Quantitative Comparisons.** Table 1 presents the quantitative results on three infrared-visible datasets (LLVIP, M3FD, and MSRS) under six evaluation metrics. Our proposed method (Dig2DIG) achieves leading performance on the majority of these metrics without requiring any training procedure. On the **LLVIP** dataset, Dig2DIG attains the best SSIM, MSE, Nabf, CC, and LPIPS scores. For instance, our MSE (1464) not only outperforms the second-best value (1658) but is also indicative of improved fidelity to the original images. Additionally, our SSIM (1.23) surpasses previous methods, demonstrating superior structural preservation. In the **M3FD** dataset, our method again secures top rankings in several metrics, including SSIM and CC. The reduction of MSE from 2221 (second-best) to 2216 underlines our consistent fidelity benefits, while the improvements in SSIM highlight enhanced structural similarity. Meanwhile, on the **MSRS** dataset, Dig2DIG achieves the best SSIM, MSE, and LPIPS scores. The lower MSE (1366) suggests stronger detail retention, and the improved LPIPS (0.282) indicates better perceptual quality. Consistent with Theorem 1, across all three VIF benchmarks we observe that positive covariance outperforms the uncorrelated baseline , which in turn outperforms negative covariance , as shown in Appendix Table 8. This observed monotone ordering supports treating $\mathrm{DIG}_k(t)$ as a proxy for $I_{k,t}$.

Table 3: Performance of different DIG intervals $S$ on the M3FD dataset.

| $S$ | PSNR↑ | SSIM↑ | MSE↓ | Nabf↓ | CC↑ | LPIPS↓ |
|---|---|---|---|---|---|---|
| 1 | 30.35 | 1.35 | 2562 | 0.035 | 0.521 | 0.308 |
| 5 | 30.92 | 1.38 | 2321 | 0.021 | 0.533 | 0.294 |
| 10 | **31.83** | **1.41** | **2215** | **0.009** | **0.573** | **0.287** |
| 20 | 31.51 | 1.40 | 2220 | 0.010 | 0.570 | 0.289 |

Table 4: Comparison of different distance measures on the M3FD dataset.

| Metric | PSNR↑ | SSIM↑ | MSE↓ | Nabf↓ | CC↑ | LPIPS↓ |
|---|---|---|---|---|---|---|
| $\varnothing$ | 30.29 | 1.36 | 2381 | 0.040 | 0.535 | 0.322 |
| $\ell_1$ | 31.41 | 1.39 | 2220 | 0.017 | 0.569 | 0.293 |
| SSIM | 31.45 | 1.41 | 2245 | 0.011 | 0.571 | 0.297 |
| $\ell_2$ | **31.83** | **1.41** | **2216** | **0.009** | **0.573** | **0.287** |

Table 5: Ablation of region-wise and time-wise DIG weighting on the M3FD dataset.

| Metric | PSNR↑ | SSIM↑ | MSE↓ | Nabf↓ | CC↑ | LPIPS↓ |
|---|---|---|---|---|---|---|
| Region+Time | **31.83** | **1.41** | **2216** | **0.009** | **0.573** | **0.287** |
| Region-only | 31.40 | 1.40 | 2269 | 0.012 | 0.570 | 0.290 |
| Time-only | 30.67 | 1.37 | 2350 | 0.031 | 0.542 | 0.310 |
| No weighting | 30.29 | 1.36 | 2381 | 0.040 | 0.535 | 0.322 |

Table 6: Comparison of different weighting functions for deriving $w_k$ on the M3FD dataset.

| Metric | PSNR↑ | SSIM↑ | MSE↓ | Nabf↓ | CC↑ | LPIPS↓ |
|---|---|---|---|---|---|---|
| Softmax | **31.83** | **1.41** | **2216** | **0.009** | **0.573** | **0.287** |
| Sigmoid | 31.62 | 1.40 | 2239 | 0.022 | 0.556 | 0.292 |
| ReLU | 31.67 | 1.41 | 2240 | 0.017 | 0.571 | 0.299 |
| No weighting | 30.29 | 1.36 | 2381 | 0.040 | 0.535 | 0.322 |

**Qualitative Comparisons.** With the aid of DIG, Dig2DIG persistently preserves fine-grained structures and salient infrared cues across all three benchmarks, as shown in Fig. 3. On **M3FD**, Our method retains the sharpest texture patterns, while TC-MoA and CCF produce noticeably blurred results. Within **LLVIP** scenes, Dig2DIG keeps licence-plate characters and facial details intact; MUFusion and Text-IF, by contrast, erode these high-frequency regions. For the **MSRS** dataset, Dig2DIG again delivers clearer human details and background textures, whereas Text-IF overwhelms visible structures with infrared intensity and CCF sacrifices visible-light detail. Our fusion strikes a superior balance between infrared saliency and visible clarity, yielding images that are both informative and visually natural. These qualitative findings corroborate the quantitative gains in Table 1, underscoring DIG's ability to safeguard critical texture during the reverse-diffusion process.

### 4.3 EVALUATION ON MULTI-FOCUS FUSION

For multi-focus image fusion, we compare our method with the state-of-the-art methods: FusionDN (Xu et al., 2020b), U2Fusion (Xu et al., 2020a), DeFusion (Liang et al., 2022), DDFM (Zhao et al., 2023b), Text-IF (Yi et al., 2024), TC-MoA (Zhu et al., 2024), TTD (Cao et al., 2025a) and CCF (Cao et al., 2025b).

**Quantitative Comparisons.** We evaluate our approach on the **MFFW** dataset using six metrics (SD, EI, EN, AG, SF, and MI). As shown in Table 2 (left), Dig2DIG outperforms competing methods on five of these six indicators by notable margins. In particular, our method achieves the highest SD (72.95), which is 5.65 above the second-best (67.30), reflecting enhanced contrast and clarity. We also secure top positions in EN (7.87), AG (6.75), SF (22.60), and MI (5.97), suggesting superior retention of details and overall information. Although FusionDN slightly outperforms Dig2DIG in EI, our model still ranks second. These results validate the efficacy of our dynamic fusion framework in handling multi-focus imagery. This result demonstrates the effectiveness of our method.

**Qualitative Comparisons.** On the multi-focus **MFFW** dataset (Fig. 4), Dig2DIG preserves fine textures and true chromatic tones across both focused and defocused regions. By contrast, TTD and TC-MoA introduce noticeable colour shifts, while FusionDN produces softer, less distinct edges. The consistently sharper structures and faithful colours highlight Dig2DIG's ability to fuse multi-focus inputs without sacrificing either structural or colour information.

### 4.4 EVALUATION ON MULTI-EXPOSURE FUSION

For multi-exposure image fusion, we compare our method with the state-of-the-art methods: FusionDN (Xu et al., 2020b), U2Fusion (Xu et al., 2020a), DeFusion (Liang et al., 2022), DDFM (Zhao et al., 2023b), Text-IF (Yi et al., 2024), TC-MoA (Zhu et al., 2024), and TTD (Cao et al., 2025a).

**Quantitative Comparisons.** As shown in Table 2 (right), we evaluate our method on the **MEFB** dataset using SD, EI, EN, AG, SF, and MI. Dig2DIG obtains the best performance on four of these

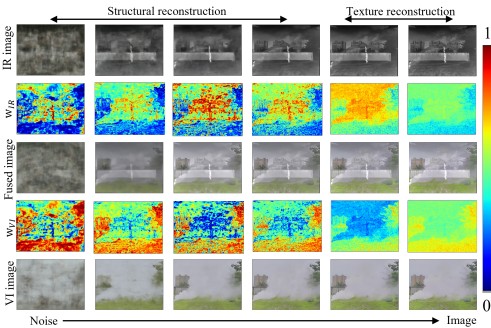

Figure 5: $w_k$ visualization.

Table 7: Efficiency on the M3FD dataset compared with diffusion-based methods.

| Method | SSIM | MSE | CC | LPIPS | t (s) | TFLOPS |
|--------|------|------|------|-------|-------|--------|
| DIG-15 | 1.30 | 2771 | 0.501 | 0.312 | 31 | 479 |
| DIG-20 | 1.38 | 2321 | 0.551 | 0.294 | 43 | 705 |
| DIG-25 | **1.41** | **2215** | **0.573** | **0.287** | 52 | 819 |
| DIG-50 | 1.41 | 2219 | 0.573 | 0.287 | 109 | 3327 |
| CCF | 1.40 | 2271 | 0.572 | 0.291 | 633 | 8505 |
| DDFM | 1.40 | 2221 | 0.568 | 0.303 | 180 | 2820 |

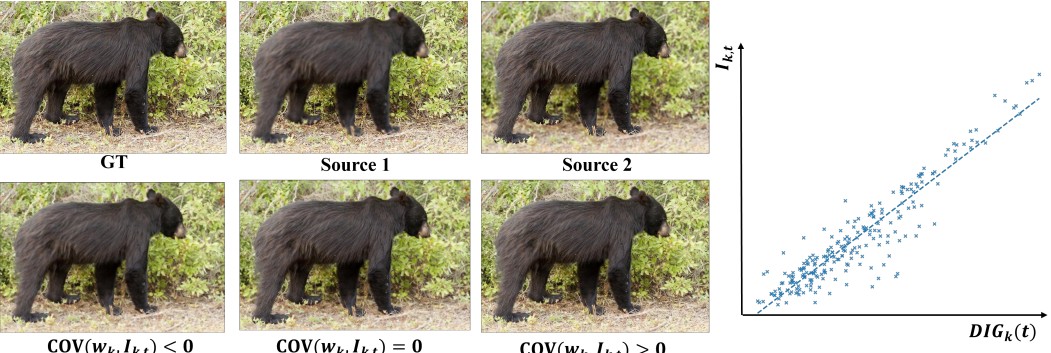

Figure 6: Influence of $\mathrm{COV}(w_k, I_{k,t})$ on quality and the positive correlation between DIG and $I_{k,t}$.

metrics (SD, EI, AG, SF), While TTD achieves a slightly higher EN and DDFM outperforms us in MI , our method still ranks second in both metrics.

**Qualitative Comparisons.** On the multi-exposure dataset (Fig. 4), TEXT-IF frequently overexposes high-luminance regions, erasing fine detail, while FusionDN, TC-MoA, and TTD introduce unnatural transitions between the desk lamp and its background. By contrast, Dig2DIG integrates information from all exposure levels, preserving highlight texture, shadow gradation, and smooth spatial transitions. The resulting images achieve a well-balanced combination of saturation and clarity, delivering the highest visual fidelity and detail retention among the compared methods.

## 4.5 DISCUSSION

**Discussion of Efficiency.** To reduce the overhead of computing DIG at each reverse diffusion step, we introduce a hyperparameter $S$ that specifies the interval at which DIG is calculated. In other words, instead of computing DIG at every step, it is updated every $S$ steps. Table 3 shows that setting the update interval to $S = 10$ offers the best trade-off between fusion quality and computational cost. When $S = 1$ or $S = 5$, DIG is refreshed at every (or nearly every) denoising step; near the *late* stages of denoising, the partially recovered image already resembles the clean target, so the residual difference becomes too small to provide reliable information-gain estimates, leading to a slight drop in performance. Conversely, with $S = 20$ the update is so sparse that finer-grained changes in the dynamic guidance can no longer be tracked, again causing a marginal decline. We therefore adopt $S = 10$, which keeps the computational overhead low while retaining high-quality fusion results.

"DIG-$N$" denotes our method with a total of $N$ reverse diffusion steps. Based on the results in Table 7, increasing the total number of reverse diffusion steps generally improves performance but also significantly increases runtime. We find that "DIG-25" effectively strikes a balance between runtime and fusion quality. Note that in the early stages of the reverse diffusion process, the noise level is high and the variance of DIG is large, which often makes the information gain inaccurate or ineffective. Based on this, and in order to fuse information more efficiently, Dig2DIG employs larger denoising steps at higher noise levels and smaller denoising steps at lower noise levels. This

approach ensures that, when the noise is sufficiently reduced, the valuable features of each modality can be more deeply integrated, thus effectively achieving more efficient information fusion.

**Discussion of DIG and** $I_{k,t}$**.** In most image fusion tasks, the ground truth $x^*(c)$ is unavailable, so the residual (yet-unfused) modality information $I_{k,t}$ in our theory cannot be directly observed. To provide a more direct quantitative support for the theoretical claims, we additionally conduct experiments on a GT-available multi-focus fusion benchmark, MFI-WHU (Zhang et al., 2021). On this dataset, since the all-in-focus GT is provided , we construct a GT-based residual measurement $I_{k,t}$ by first computing the pixel-wise $\ell_2$ distance between the current fused image $x_t$ and the GT, and then averaging it within the focused regions of each source modality. With $I_{k,t}$ available, we can validate two key implications of our bound within a unified experiment. As shown in Fig. 6, the scatter plot between $\mathrm{DIG}_k(t)$ and $I_{k,t}$ across all image exhibits a strong positive correlation (Pearson $r = 0.9345$), supporting DIG as a reliable monotone surrogate of the GT-based residual. Moreover, when using $\mathrm{softmax}(I_{k,t})$ to form fusion weights, positively aligning weights with residual magnitudes yields the lowest error (MSE 597.9), whereas uniform weighting and negatively correlated weighting lead to substantially higher errors (MSE 1060.8 and 1883.7, respectively). Together, these GT-based results quantitatively corroborate that DIG faithfully reflects residual information and that residual-aligned dynamic weighting tightens the generalization bound in practice.

$w_k$ **Visualization Results.** Fig. 5 visualizes $w_k$, in early denoising, prominent infrared structures exhibit larger $w_k$; as denoising proceeds and the overall image structure is rapidly reconstructed, regions with rich fine-grained textures then exhibit larger $w_k$. The results indicate that DIG accurately captures the relative magnitudes of information gain contributed by each modality across different regions of the image. The magnitude of DIG directly reflects how much information from the corresponding modality remains unfused in that region, and can therefore be used to guide fusion in diffusion models. further examples are provided in Appendix D.

**Discussion of Region-wise and Time-wise DIG Weighting.** Table 5 studies how applying DIG-based weights across regions and timesteps affects fusion quality. Using both region-wise and time-wise dynamic weighting consistently yields the best performance, indicating that the two dimensions are complementary. Region-only weighting already brings a clear gain over no weighting, showing that spatially adaptive guidance is important for identifying modality-salient areas (e.g., thermal targets vs. visible textures). Time-only weighting also improves over the baseline but is weaker than region-only weighting, suggesting that temporal adaptivity alone cannot fully resolve local modality competition without spatial discrimination. Overall, the strongest results are achieved when Dig2DIG jointly accounts for where each modality is informative and when this residual information should be emphasized during denoising, validating the design of our weighting strategy.

**Discussion of Weighting Functions for Deriving** $w_k$**.** Table 6 compares different ways to map DIG to fusion weights $w_k$. Softmax, Sigmoid gating, and ReLU normalization all consistently outperform the no weighting baseline on all metrics, indicating that Dig2DIG is robust to the specific DIG to $w_k$ mapping. Among them, Softmax gives the best overall performance in our setting, while Sigmoid gating and ReLU remain competitive with only minor gaps.

**Discussion of the choice of** $l$**.** To determine a suitable function for computing $l$, we conduct experiments on the M3FD dataset using different metric functions, including $\ell_1$, SSIM, and $\ell_2$, while considering the case with fixed uniform weight as the "baseline," denoted by $\varnothing$. in the table 4, it is evident that the $\ell_2$ distance achieves the best performance. Therefore, we adopt $\ell_2$ distance as the evaluation function for subsequent experiments. The overall performance suggests that introducing a reasonable metric function consistently enhances the results to varying degrees. Compared to the baseline without any distance metric, these improvements indicate the strong applicability of our method to different metric functions in both theoretical and practical aspects.

## 5 CONCLUSION

In this paper, we introduced a novel dynamic denoising diffusion framework for image fusion, which explicitly addresses the spatio-temporal imbalance in denoising through the lens of Diffusion Information Gains. By quantifying DIG by each modality at different noise levels, our method adaptively weights the fusion guidance to preserve critical features while ensuring high-quality, reliable fusion results. Theoretically, we proved that aligning the modality fusion weight with the residual modality information reduces the upper bound of the generalization error, thus offering a rigorous explanation for the advantages of dynamic denoising fusion.

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

APPENDIX

## A    MORE DETAILS ABOUT MULTIMODAL GUIDANCE

This supplementary note re-derives, step by step, the gradient-based sampling rule that underlies our guided image-fusion framework. Starting from the standard DDPM forward–reverse processes, we show how additional conditional terms lead to the multimodal guidance formula in equation 18. Readers who are new to diffusion models can thus follow the main paper without consulting external references.

In DDPM (Denoising Diffusion Probabilistic Models), the forward diffusion process adds noise to a clean sample $x_0$ over multiple steps, eventually transforming it into nearly pure Gaussian noise. This procedure is linear, so one can sample $x_t$ in a single shot at step $t$ via the closed-form expression:

$$x_t = \sqrt{\bar{\alpha}_t}\, x_0 + \sqrt{1 - \bar{\alpha}_t}\, \epsilon, \quad \epsilon \sim \mathcal{N}(0, I), \tag{9}$$

where $\alpha_t = 1 - \beta_t$, $\bar{\alpha}_t = \prod_{i=1}^{t} \alpha_i$, and $\{\beta_t\}_{t=1}^{T}$ is a predefined variance schedule. As $t$ increases, $x_t$ approaches a nearly pure noise distribution.

To generate a sample during inference, one starts from pure noise $x_T$ and iteratively denoises down to $x_0$. Under a common parameterization, each reverse update step is given by:

$$x_{t-1} = \frac{1}{\sqrt{\alpha_t}} \left( x_t - \frac{1 - \alpha_t}{\sqrt{1 - \bar{\alpha}_t}} \epsilon_\theta(x_t, t) \right) + \sigma(t)\, z, \tag{10}$$

Here $\sigma(t)$ is the closed-form posterior standard deviation; it does not depend on network parameters, where $\sigma^2(t) = \frac{(1-\alpha_t)(1-\bar{\alpha}_{t-1})}{1-\bar{\alpha}_t}$, $\epsilon_\theta(\cdot)$ is the network's noise prediction, and $z \sim \mathcal{N}(0, I)$. By iterating from $t = T$ down to $t = 0$, one transforms pure noise into a nearly clean sample.

From the closed-form forward process equation 9, If the model $\epsilon_\theta(x_t, t)$ accurately predicts the noise $\epsilon$, one can approximate the "denoised" $\hat{x}_0$ as:

$$\hat{x}_0 \approx \frac{1}{\sqrt{\bar{\alpha}_t}} \left( x_t - \sqrt{1 - \bar{\alpha}_t}\, \epsilon_\theta(x_t, t) \right). \tag{11}$$

This highlights the reverse denoising mechanism: once the correct noise component of $x_t$ is identified, we retrieve a good approximation of the clean data.

From the perspective of stochastic differential equations or variational inference, and based on equation 9, the gradient of $\log p(x_t)$ with respect to $x_t$ (i.e., the score function) can be expressed as:

$$\nabla_{x_t} \log p(x_t) = - \frac{x_t - \sqrt{\bar{\alpha}_t} x_0}{1 - \bar{\alpha}_t}. \tag{12}$$

Using the estimated $\hat{x}_0$ to replace $x_0$, and substituting equation 11, into equation 12, we derive:

$$\nabla_{x_t} \log p(x_t) \approx - \frac{\epsilon_\theta(x_t, t)}{\sqrt{1 - \bar{\alpha}_t}}. \tag{13}$$

In certain applications, such as text-to-image generation and multimodal data fusion, we often wish to incorporate additional conditions during the sampling process. By Bayes' theorem, the gradient of the conditional log-probability with respect to the current sample $x_t$ can be written as

$$\nabla_{x_t} \log p(x_t \mid c) = \nabla_{x_t} \log p(x_t) + \nabla_{x_t} \log p(c \mid x_t). \tag{14}$$

Here, $c$ represents one or more conditions guiding the generation process.

For $K$ conditions $\{c_k\}_{k=1}^{K}$, a commonly used separable approximation in multi-guidance diffusion is to model a weighted joint conditional distribution as a product of experts:

$$p_w(c \mid x_t) \propto \prod_{k=1}^{K} p(c_k \mid x_t)^{w_k}, \qquad w_k \geq 0, \sum_{k=1}^{K} w_k = 1. \tag{15}$$

Under this explicit joint model, the conditional score is Bayes consistent and satisfies

$$\nabla_{x_t} \log p_w(c \mid x_t) = \sum_{k=1}^{K} w_k \, \nabla_{x_t} \log p(c_k \mid x_t), \tag{16}$$

which recovers the weighted guidance form used in our sampler. Such PoE-style score composition is standard in multi-condition diffusion guidance.

where $w_k$ is a user-defined weight indicating the relative importance of condition $c_k$. Substituting equation 16 into equation 14 then gives:

$$\nabla_{x_t} \log p(x_t \mid c) \approx \nabla_{x_t} \log p(x_t) + \nabla_{x_t} \log p_w(c \mid x_t). \tag{17}$$

By adjusting the weights $\{w_k\}$, one can modulate the strength of each condition's contribution to the gradient-based sampling step, thus allowing fine-grained control over the generated samples.

From equation 10, equation 13, and equation 17, we derive the following update equation for the diffusion model:

$$x_{t-1} \approx \frac{1}{\sqrt{\alpha_t}} x_t + \underbrace{\frac{1}{\sqrt{\alpha_t}} (1 - \alpha_t) \nabla_{x_t} \log p(x_t)}_{\text{Unconditional Guidance}} + \underbrace{\frac{1}{\sqrt{\alpha_t}} (1 - \alpha_t) \sum_{k=1}^{K} w_k \nabla_{x_t} \log p(c_k \mid x_t)}_{\text{Multimodal Guidance}} + \underbrace{\sigma(t) z}_{\text{Noise}}. \tag{18}$$

This equation demonstrates that the update step in the diffusion model can be decomposed into three key components: Unconditional Guidance, Multimodal Guidance, and Noise. This decomposition encourages further exploration of the role of Multimodal Guidance in reducing the model's generalization error and improving conditional generation quality.

## B  PROOF

Our goal is to quantify how much each guided gradient step reduces the distance between the current fused image and the ideal fusion $x^*(c)$. The proof shows that, after summing all $T$ reverse diffusion steps, the generalization error separates into (i) fixed constants that do not depend on how we weight the modalities, and (ii) a negative sum of covariance terms $\text{Cov}(w_k, \text{alignment})$. Therefore, the more a guidance direction aligns with the "correct move" toward $x^*(c)$, the larger (more negative) the covariance can be made by assigning a bigger weight $w_k$, directly tightening the upper bound.

Importantly, the above summation is taken along the reverse-diffusion fusion trajectory $\{x_t\}_{t=0}^{T}$ generated by our sampler in equation 18. Thus, each guided gradient step refers to the concrete update from $x_t$ to $x_{t-1}$ in this trajectory, and the following smoothness inequality is applied only to consecutive pairs $(x_t, x_{t-1})$ produced by the diffusion sampler.

Since $\zeta(\cdot, x^*(c))$ is $L$-smooth with respect to its first argument, Here "$L$-smooth" means gradient-Lipschitz: $\|\nabla\zeta(x) - \nabla\zeta(y)\| \leq L\|x - y\|$ for all $x, y \in \mathbb{R}^{H \times W \times N}$.

For any $x, y \in \mathbb{R}^{H \times W \times N}$ we have:

$$\zeta(y, x^*(c)) \leq \zeta(x, x^*(c)) + \nabla_{x_t}\zeta(x, x^*(c)) \cdot (y - x) + \frac{L}{2}\|y - x\|^2. \tag{19}$$

Letting $x = x_t$ and $y = x_{t-1}$ gives a one-step difference inequality:

$$\zeta(x_{t-1}, x^*(c)) - \zeta(x_t, x^*(c)) \leq \nabla_{x_t}\zeta(x_t, x^*(c)) \cdot (x_{t-1} - x_t) + \frac{L}{2}\|x_{t-1} - x_t\|^2. \tag{20}$$

In many cases, we are primarily interested in the first-order term (dot product) and regard the second-order term as a manageable constant. Specifically, if we assume $\|x_{t-1} - x_t\|^2 \leq \Delta_t^2$, so that equation 20 can be relaxed to:

$$\zeta(x_{t-1}, x^*(c)) - \zeta(x_t, x^*(c)) \leq \nabla_{x_t}\zeta(x_t, x^*(c)) \cdot (x_{t-1} - x_t) + \frac{L}{2}\Delta_t^2. \tag{21}$$

In practice $T$ (the number of reverse steps) is large and $\|x_{t-1} - x_t\|$ is dominated by the scheduler ; we upper-bound it by a deterministic constant $\Delta_t$ and absorb $\frac{L}{2}\Delta_t^2$ into later constants. We do *not* need an exact value of $\Delta_t$, only the existence of such a uniform bound. Thus, a simple upper bound $\frac{L}{2}\Delta_t^2$ can be carried along in subsequent summations. we decompose:

$$x_{t-1} - x_t = \underbrace{\left(\tfrac{1}{\sqrt{\alpha_t}} - 1\right)x_t}_{\text{(I) scaling difference}} + \underbrace{\frac{1}{\sqrt{\alpha_t}}(1 - \alpha_t)\nabla_{x_t}\log p(x_t)}_{\text{(II) unconditional gradient}}$$

$$+ \underbrace{\frac{1}{\sqrt{\alpha_t}}(1 - \alpha_t)\sum_{k=1}^{K} w_k \nabla_{x_t}\log p(c_k \mid x_t)}_{\text{(III) multimodal guidance}}$$

$$+ \underbrace{\sigma(t)\,z}_{\text{(IV) noise}} . \tag{22}$$

(I) is deterministic w.r.t. $(c, z)$ once $x_t$ is fixed. (II) uses the unconditional score; deterministic conditioned on $x_t$. (III) contains the only $w_k$-dependent part. (IV) is the only term that depends on the fresh noise $z_t$. Plugging this into equation 21, we have:

$$\zeta(x_{t-1},\, x^*(c)) - \zeta(x_t,\, x^*(c)) \leq \nabla_{x_t}\zeta(x_t,\, x^*(c)) \cdot \left[\left(\tfrac{1}{\sqrt{\alpha_t}} - 1\right)x_t\right]$$

$$+ \nabla_{x_t}\zeta(x_t,\, x^*(c)) \cdot \left[\tfrac{1}{\sqrt{\alpha_t}}(1 - \alpha_t)\nabla_{x_t}\log p(x_t)\right]$$

$$+ \nabla_{x_t}\zeta(x_t,\, x^*(c)) \cdot \left[\tfrac{1}{\sqrt{\alpha_t}}(1 - \alpha_t)\sum_{k=1}^{K} w_k \nabla_{x_t}\log p(c_k \mid x_t)\right]$$

$$+ \nabla_{x_t}\zeta(x_t,\, x^*(c)) \cdot \left[\sigma(t)\,z\right] + \frac{L}{2}\Delta_t^2. \tag{23}$$

Because the fresh noise $z_t$ is independent of $(c, x_t)$ given $x_t$ (standard DDPM sampling), all terms that do not contain either $w_k$ or $z_t$ are $F_t$-measurable deterministic functions. We collect them into:

$$G(x_t, c, t) = \nabla_{x_t}\zeta(x_t,\, x^*(c)) \cdot \left[\left(\tfrac{1}{\sqrt{\alpha_t}} - 1\right)x_t\right] \tag{24}$$

$$+ \nabla_{x_t}\zeta(x_t,\, x^*(c)) \cdot \left[\tfrac{1}{\sqrt{\alpha_t}}(1 - \alpha_t)\nabla_{x_t}\log p(x_t)\right].$$

Then we can rewrite equation 23 more compactly as:

$$\zeta(x_{t-1},\, x^*(c)) - \zeta(x_t,\, x^*(c)) \leq G(x_t, c, t)$$

$$+ \nabla_{x_t}\zeta(x_t,\, x^*(c)) \cdot \left[\tfrac{1}{\sqrt{\alpha_t}}(1 - \alpha_t)\sum_{k=1}^{K} w_k \nabla_{x_t}\log p(c_k \mid x_t)\right]$$

$$+ \nabla_{x_t}\zeta(x_t,\, x^*(c)) \cdot \left[\sigma(t)\,z\right] + \frac{L}{2}\Delta_t^2. \tag{25}$$

Summing from $t = 1$ to $T$ in a telescoping manner, we have:

$$\zeta(x_0,\, x^*(c)) = \zeta(x_T,\, x^*(c)) + \sum_{t=1}^{T}\left[\zeta(x_{t-1}, x^*(c)) - \zeta(x_t, x^*(c))\right]. \tag{26}$$

Applying equation 25 at each step, we obtain (summing over $t$):

$$\zeta(x_0,\, x^*(c)) \leq \zeta(x_T,\, x^*(c)) + \sum_{t=1}^{T} G(x_t, c, t) + \sum_{t=1}^{T}\nabla_{x_t}\zeta(x_t,\, x^*(c)) \cdot \left[\sigma(t)\,z_t\right] + \sum_{t=1}^{T}\tfrac{L}{2}\Delta_t^2$$

$$+ \sum_{t=1}^{T}\nabla_{x_t}\zeta(x_t,\, x^*(c)) \cdot \left[\tfrac{1}{\sqrt{\alpha_t}}(1 - \alpha_t)\sum_{k=1}^{K} w_k \nabla_{x_t}\log p(c_k \mid x_t)\right]. \tag{27}$$

Finally, recall $x_0 = F(c)$, so $\zeta(x_0, x^*(c)) = \zeta(F(c), x^*(c))$. Taking $\mathbb{E}_{c\sim D}$ on both sides of equation 27, we obtain

$$
\begin{aligned}
\mathrm{GError}(F) &= \mathbb{E}_{c,z}\big[\zeta(x_0, x^*(c))\big] \\
&\leq \mathbb{E}_{c,z}\big[\zeta(x_T, x^*(c))\big] + \mathbb{E}_{c,z}\Big[\sum_{t=1}^{T} G(x_t, c, t)\Big] + \mathbb{E}_{c,z}\Big[\sum_{t=1}^{T} \tfrac{L}{2}\Delta_t^2\Big] \\
&\quad - \mathbb{E}_{c,z}\Big[\sum_{t=1}^{T}\sum_{k=1}^{K} \tfrac{1}{\sqrt{\alpha_t}}(1-\alpha_t)w_k \cdot \big[-\nabla_{x_t}\zeta(x_t, x^*(c))\,\nabla_{x_t}\log p(c_k \mid x_t)\big]\Big] \\
&\quad + \mathbb{E}_{c,z}\Big[\sum_{t=1}^{T}\nabla_{x_t}\zeta(x_t, x^*(c)) \cdot \big[\sigma(t)\,z_t\big]\Big] \\
&= \underbrace{\mathbb{E}_{c,z}\big[\zeta(x_T, x^*(c))\big] + \mathbb{E}_{c,z}\Big[\sum_{t=1}^{T} G(x_t, c, t)\Big] + \mathbb{E}_{c,z}\Big[\sum_{t=1}^{T} \tfrac{L}{2}\Delta_t^2\Big]}_{\text{constant}} \\
&\quad - \sum_{t=1}^{T}\Big[\tfrac{1}{\sqrt{\alpha_t}}(1-\alpha_t)\underbrace{\sum_{k=1}^{K}\mathbb{E}_{c,z}\big[w_k\big]}_{\text{equal to 1}}\underbrace{\mathbb{E}_{c,z}\big[-\nabla_{x_t}\zeta(x_t, x^*(c))\,\nabla_{x_t}\log p(c_k \mid x_t)\big]}_{\text{constant}}\Big] \\
&\quad - \sum_{t=1}^{T}\Big[\tfrac{1}{\sqrt{\alpha_t}}(1-\alpha_t)\sum_{k=1}^{K}\mathrm{Cov}\Big(w_k, -\nabla_{x_t}\zeta(x_t, x^*(c))\,\nabla_{x_t}\log p(c_k \mid x_t)\Big)\Big] \\
&\quad + \sum_{t=1}^{T}\Big[\mathbb{E}_{c,z}\big[\nabla_{x_t}\zeta(x_t, x^*(c))\big]\underbrace{\mathbb{E}_{c,z}\big[\sigma(t)\,z_t\big]\}}_{\text{equal to 0}} + \underbrace{\mathrm{Cov}\Big(\nabla_{x_t}\zeta(x_t, x^*(c)),\ \sigma(t)\,z_t\Big)}_{\text{equal to 0 (by independence)}}\Big] \\
&= C - \sum_{t=1}^{T}\Big[\tfrac{1}{\sqrt{\alpha_t}}(1-\alpha_t)\sum_{k=1}^{K}\mathrm{Cov}\Big(w_k, \underbrace{-\nabla_{x_t}\zeta(x_t, x^*(c))\,\nabla_{x_t}\log p(c_k \mid x_t)}_{\text{alignment Measure}}\Big)\Big]
\end{aligned}
\tag{28}
$$

Here independence follows from the fact that $z_t$ is freshly sampled *after* $x_t$ has been computed, hence uncorrelated with any $\mathcal{F}_t$-measurable quantity. We revisit the alignment measure $-\nabla_{x_t}\zeta(x_t, x^*(c))\,\nabla_{x_t}\log p(c_k \mid x_t)$ to elucidate its geometric interpretation. Recall we set $v_t = -\nabla_{x_t}\zeta(x_t, x^*(c))$. By the $L$-smoothness of $\zeta$ and the bounded data domain $\|x_t - x^*(c)\| \leq B_t$, we have the deterministic bound

$$
\|v_t\| = \|\nabla_{x_t}\zeta(x_t, x^*(c))\| \leq LB_t =: \bar{V}.
\tag{29}
$$

We will use $\bar{V}$ in place of $\|v_t\|$ whenever the latter is factored outside a covariance. Intuitively, when $x_t$ is close to the ideal fused image $x^*(c)$, $-\nabla_{x_t}\zeta(x_t, x^*(c))$ should point from $x_t$ toward $x^*(c)$. Thus $v_t$ can be seen as a vector indicating the current direction from the generated image to the ideal fused image $x^*(c)$. Let $\|v_t\|$ denote the norm of $v_t$. Next, let $\theta_{t,k} \in [0, \pi]$ be the angle between $v_t$ and $\nabla_{x_t}\log p(c_k \mid x_t)$. By definition of the dot product in terms of norms and angles, we have

$$
-\nabla_{x_t}\zeta(x_t, x^*(c))\,\nabla_{x_t}\log p(c_k \mid x_t) = \|v_t\|\,\|\nabla_{x_t}\log p(c_k \mid x_t)\|\cos(\theta_{t,k}).
\tag{30}
$$

So, we have

$$
\mathrm{GError}(F) \leq C - \sum_{t=1}^{T}\Big[\tfrac{1}{\sqrt{\alpha_t}}(1-\alpha_t)\bar{V}\sum_{k=1}^{K}\mathrm{Cov}\Big(w_k,\ \|\nabla_{x_t}\log p(c_k \mid x_t)\|\cos(\theta_{t,k})\Big)\Big]
\tag{31}
$$

As a result, the "directional alignment" with respect to the ideal fusion direction $v_t$ boils down to the projection $\|\nabla_{x_t}\log p(c_k \mid x_t)\|\cos(\theta_{t,k})$. The bound shows that boosting a modality's weight $w_k$ will tighten (give a lower) error bound iff the modality's guidance gradient points more toward the ideal-fusion direction ($\cos\theta_{t,k} > 0$) than average. This matches the practical heuristic "assign larger weights to modalities that are currently more helpful for approaching $x^*(c)$."

The covariance term obtained earlier still contains the factor $\|\nabla_{x_t} \log p(c_k \mid x_t)\|$, mixing magnitude and direction. Before turning to the directional component, we first explain that the gradient-guidance magnitudes are, to an excellent approximation, identical across modalities at every diffusion step. In our implementation, we compute the gradients using a surrogate objective that is analytically equivalent to an $\ell_1$ loss (Appendix C); consequently the norms for any two modalities satisfy $\|\nabla_{x_t} \log p(c_{k_1} \mid x_t)\| \approx \|\nabla_{x_t} \log p(c_{k_2} \mid x_t)\|$. Experiments in Appendix D corroborate this observation. Let this common value be $s_t$ (step–dependent, modality–independent). Pulling $s_t$ outside the covariance gives

$$\text{GError}(F) \;\leq\; C - \sum_{t=1}^{T}\Big[D_t \sum_{k=1}^{K} \text{Cov}\big(w_k, \cos\theta_{t,k}\big)\Big], \qquad D_t := \tfrac{1}{\sqrt{\alpha_t}}(1 - \alpha_t)\,\bar{V}\,s_t. \tag{32}$$

The problem is therefore reduced to understanding how the angles $\theta_{t,k}$ depend on the still-unfused information in each modality. All geometric arguments below are carried out in the discrete pixel space: each image is vectorized as $x \in \mathbb{R}^{HWN}$ with the standard Euclidean inner product $\langle x, y \rangle = \sum_p x_p y_p$ and the induced norm $\|x\|$. The decomposition into $u_{\text{IR}}, u_{\text{RGB}}, u_s$ is an approximate orthogonal subspace split, a standard viewpoint in fusion analysis. To carry out a geometric analysis and make the dependence of $\theta_{t,k}$ on $i_t^k$ explicit, we restrict attention to a simplified two-modality setting with infrared (IR) and visible light (RGB). Fix a reverse step $t$ (index suppressed). For clarity, the notational conventions are collected once in the table below, and all symbols retain their original meanings throughout the derivation.

| | | |
|---|---|---|
| $u_{\text{IR}}, u_{\text{RGB}}$ | – | orthogonal high-frequency bases, |
| $u_s$ | – | shared low-frequency basis, $|\langle u_s, u_* \rangle| = \delta \ll 1$, |
| $x_F,\; x_t$ | – | ideal and current fused images, |
| $c_{\text{IR}}, c_{\text{RGB}}, c_s$ | – | projections of $x_F$, |
| $\alpha, \beta, \gamma \in [0,1]$ | – | unrecovered fractions, |
| $a := \alpha c_{\text{IR}},\; b := \beta c_{\text{RGB}},\; c := \gamma c_s$ | – | residual information, |
| $\kappa_{\text{IR}}, \kappa_{\text{RGB}} \geq 0$ | – | score projections on $u_s$. |

**Link to $I_{k,t}$.** In this two-modality geometric instantiation, $a$ and $b$ are exactly the modality-specific residual components of $x^*(c) - x_t$ (the main-text definition $I_{k,t} = \|\Pi_k(x^*(c) - x_t)\|^2$); hence we identify $I_{\text{IR},t} \propto a^2$ and $I_{\text{RGB},t} \propto b^2$, so the following angle analysis directly establishes the monotone relation between $\cos\theta_{t,k}$ and $I_{k,t}$.

Define three explicit vectors

$$v = a\,u_{\text{IR}} + b\,u_{\text{RGB}} + c\,u_s, \qquad g_{\text{IR}} = a\,u_{\text{IR}} + \kappa_{\text{IR}}u_s, \qquad g_{\text{RGB}} = b\,u_{\text{RGB}} + \kappa_{\text{RGB}}u_s.$$

With $O(\delta)$ terms discarded,

$$|v|^2 = a^2 + b^2 + c^2, \quad |g_{\text{IR}}|^2 = a^2 + \kappa_{\text{IR}}^2, \quad |g_{\text{RGB}}|^2 = b^2 + \kappa_{\text{RGB}}^2,$$

$$v\cdot g_{\text{IR}} = a^2 + c\kappa_{\text{IR}}, \qquad v\cdot g_{\text{RGB}} = b^2 + c\kappa_{\text{RGB}}.$$

Hence

$$\cos\theta_{\text{IR}} = \frac{a^2 + c\,\kappa_{\text{IR}}}{\sqrt{a^2 + b^2 + c^2}\,\sqrt{a^2 + \kappa_{\text{IR}}^2}}, \qquad \cos\theta_{\text{RGB}} = \frac{b^2 + c\,\kappa_{\text{RGB}}}{\sqrt{a^2 + b^2 + c^2}\,\sqrt{b^2 + \kappa_{\text{RGB}}^2}}. \tag{33}$$

Introduce the log ratio $h(a,b) := \ln\big(\cos\theta_{\text{IR}} / \cos\theta_{\text{RGB}}\big)$; substituting equation 33 and differentiating *without abbreviation* yields

$$\frac{\partial h}{\partial a} = \frac{a\big[a^2 + 2\kappa_{\text{IR}}^2 - c\kappa_{\text{IR}}\big]}{(a^2 + c\kappa_{\text{IR}})(a^2 + \kappa_{\text{IR}}^2)}, \qquad \frac{\partial h}{\partial b} = -\frac{b\big[b^2 + 2\kappa_{\text{RGB}}^2 - c\kappa_{\text{RGB}}\big]}{(b^2 + c\kappa_{\text{RGB}})(b^2 + \kappa_{\text{RGB}}^2)}.$$

For fixed $a$ the quadratic $g(\kappa) = a^2 + 2\kappa^2 - c\kappa$ attains its minimum $a^2 - c^2/8$ at $\kappa = c/4$, so $\partial h/\partial a > 0$ for *all* $\kappa_{\text{IR}} \geq 0$ iff $a > \tau := c/\sqrt{8}$. The same threshold holds for $b$.

Main region $a > \tau,\; b > \tau$. Because $a, b > \tau := c/\sqrt{8}$, the numerators in the partial-derivative expressions are strictly positive for every $\kappa_{\text{IR}}, \kappa_{\text{RGB}} \geq 0$, hence $\partial_a h > 0$ and $\partial_b h < 0$. Thus

$$a > b \iff h(a,b) > 0 \iff \cos\theta_{\text{IR}} > \cos\theta_{\text{RGB}}. \tag{34}$$

Within this work zone, the modality that still carries *more* residual energy ($a$ or $b$) always owns the larger cosine.

Boundary case $a \leq \tau$ (IR almost exhausted). Define

$$\phi(\kappa) := \frac{a^2 + c\kappa}{\sqrt{a^2 + \kappa^2}}, \qquad \phi'(\kappa) = \frac{a^2(c - \kappa)}{(a^2 + \kappa^2)^{3/2}}.$$

Hence $\phi$ increases on $[0, c]$ and decreases on $(c, \infty)$, with maximum $\phi(c) = \sqrt{a^2 + c^2}$. Substituting $\kappa_{\mathrm{IR}} = c$ into equation 33 gives the universal upper bound

$$\cos\theta_{\mathrm{IR}} \ \leq \ \frac{\sqrt{a^2 + c^2}}{\sqrt{a^2 + b^2 + c^2}}. \tag{35}$$

For the RGB side consider

$$\psi(\kappa) := \frac{b^2 + c\kappa}{\sqrt{b^2 + \kappa^2}}, \qquad \psi'(\kappa) = \frac{b^2(c - \kappa)}{(b^2 + \kappa^2)^{3/2}},$$

so $\psi$ attains its minimum $\psi(0) = b$. Taking $\kappa_{\mathrm{RGB}} = 0$ yields a lower bound

$$\cos\theta_{\mathrm{RGB}} \ \geq \ \frac{b}{\sqrt{a^2 + b^2 + c^2}}. \tag{36}$$

Whenever $b^2 > a^2 + c^2$—in particular for the convenient sufficient condition $b \geq \sqrt{9/8}\, c$ (recall $a \leq \tau$ implies $a^2 \leq c^2/8$)—the numerators in equation 35–equation 36 satisfy the strict inequality $\sqrt{a^2 + c^2} < b$, which forces $\cos\theta_{\mathrm{IR}} < \cos\theta_{\mathrm{RGB}}$.

Boundary case $b \leq \tau$ (RGB almost exhausted). The argument is perfectly symmetric: replace $(a, \kappa_{\mathrm{IR}})$ with $(b, \kappa_{\mathrm{RGB}})$ and interchange the roles of IR/RGB. One obtains $\cos\theta_{\mathrm{IR}} > \cos\theta_{\mathrm{RGB}}$ whenever $a^2 > b^2 + c^2$ (sufficient condition $a \geq \sqrt{9/8}\, c$).

Corner case $a \leq \tau$ and $b \leq \tau$. Both numerators in equation 35–equation 36 are then bounded by $\sqrt{\tau^2 + c^2}$ while the common denominator exceeds $\sqrt{2c^2 + \tau^2}$, so each cosine is

$$\cos\theta_{\mathrm{IR}}, \ \cos\theta_{\mathrm{RGB}} \ \leq \ \frac{\sqrt{\tau^2 + c^2}}{\sqrt{2c^2 + \tau^2}} < \frac{1}{\sqrt{2}}.$$

Both guidance directions are therefore weak; any ordering between the two becomes immaterial for fusion.

Combining equation 34 with the two boundary analyses we obtain: outside a negligible corner region, the modality with larger $a$ or $b$ has the larger cosine. Because the $a$ and $b$ is exactly what the residual-information measure $I_{k,t}$ counts, there exists a positive scale $R_t$ such that

$$I_{k,t} = R_t \cos\theta_{t,k}, \qquad \mathrm{Cov}(w_k, \cos\theta_{t,k}) = \frac{1}{R_t}\mathrm{Cov}(w_k, I_{k,t}). \tag{37}$$

The geometric derivation above was carried out for two modalities solely to keep every intermediate quantity visible. The heart of the argument is the pairwise link between (i) the unrecovered residual information of a modality and (ii) the cosine it forms with the ideal-fusion direction. For a fusion task with $K > 2$ modalities $\{c_1, \ldots, c_K\}$ one decomposes the signal space into $K$ orthogonal high-frequency axes $\{u_1, \ldots, u_K\}$, plus the shared low-frequency axis $u_s$. Fixing any pair $(i, j)$ and repeating the foregoing two-dimensional projection immediately yields the same monotone relationship $I_{i,t} \propto \cos\theta_{i,t}$ and $I_{j,t} \propto \cos\theta_{j,t}$. Therefore the covariance structure $\mathrm{Cov}(w_k, I_{k,t})$ derived for two modalities extends component-wise to all $k \in \{1, \ldots, K\}$ without algebraic changes.

Substituting equation 37 into equation 32 and absorbing the positive factor $B_t/R_t$ into a new step-wise constant finally delivers:

$$\mathrm{GError}(F) \ \leq \ C - \sum_{t=1}^{T}\sum_{k=1}^{K} A_{k,t}\, \mathrm{Cov}(w_k, I_{k,t}) \ . \tag{38}$$

# C MORE DETAILS

## C.1 MORE DETAILS ABOUT CONDITIONAL SCORE

During reverse diffusion we require the conditional score $\nabla_{f_t} \log p(i, v \mid f_t)$ to steer the stochastic differential equation. We follow DDFM's EM routine and reproduce every algebraic step in this appendix so that the full computation is visible in one place. The presentation starts with the necessary notation, then walks through the EM loop, and finally collects the formulas that are fed into the diffusion sampler.

Let $i, v \in \mathbb{R}^{H \times W \times N}$ be the infrared and visible images. At diffusion timestep $t$ the current estimate is $f_t$. The classical fusion loss is

$$\mathcal{L}(f) = \|f - i\|_1 + \varphi \|f - v\|_1. \tag{39}$$

We transform this $\ell_1$ objective into a quadratic surrogate whose gradient is available in closed form.

Set $x = f - v$ and $y = i - v$. Then equation 39 becomes $\|y - x\|_1 + \varphi \|x\|_1$, which can be interpreted as the maximum-likelihood problem of a Laplace model

$$x_{ij} \sim \mathrm{Lap}(0, \rho), \qquad y_{ij} \mid x_{ij} \sim \mathrm{Lap}(x_{ij}, \gamma). \tag{40}$$

Using the Gaussian–exponential mixture representation of the Laplace distribution, each absolute term introduces an auxiliary precision variable. The resulting hierarchical graph is

$$\begin{cases} y_{ij} \mid x_{ij}, m_{ij} \sim \mathcal{N}(y_{ij}; x_{ij}, m_{ij}), \\ m_{ij} \sim \mathrm{Exp}(m_{ij}; \gamma), \\ x_{ij} \mid n_{ij} \sim \mathcal{N}(x_{ij}; 0, n_{ij}), \\ n_{ij} \sim \mathrm{Exp}(n_{ij}; \rho). \end{cases} \tag{41}$$

Adding a total-variation term $r(x) = \frac{\psi}{2} \|\nabla x\|_2^2$ leads to the log-likelihood

$$\mathcal{L}(x) = -\sum_{i,j} \left[ \frac{(x_{ij} - y_{ij})^2}{2m_{ij}} + \frac{x_{ij}^2}{2n_{ij}} \right] - \frac{\psi}{2} \|\nabla x\|_2^2. \tag{42}$$

For the current latent image $x^{(t)}$ compute

$$\bar{m}_{ij} = \sqrt{\frac{2 (y_{ij} - x_{ij}^{(t)})^2}{\gamma}}, \qquad \bar{n}_{ij} = \sqrt{\frac{2 x_{ij}^{(t) \, 2}}{\rho}}. \tag{43}$$

Define weights $m_{ij} = \sqrt{\bar{m}_{ij}}$ and $n_{ij} = \sqrt{\bar{n}_{ij}}$.

The conditional expectation of equation 42 becomes

$$E(x) = \|m \odot (x - y)\|_2^2 + \|n \odot x\|_2^2 + \psi \|\nabla x\|_2^2, \tag{44}$$

where $\odot$ is element-wise multiplication.

Introduce auxiliary variables $u, k$ and minimise

$$\begin{aligned} &\|m \odot (x - y)\|_2^2 + \|n \odot x\|_2^2 + \psi \|u\|_2^2 \\ &+ \frac{\eta}{2} \left( \|u - \nabla k\|_2^2 + \|k - x\|_2^2 \right). \end{aligned} \tag{45}$$

With the Fourier transform $\mathcal{F}$ (and complex conjugate $\bar{\cdot}$) the coordinate updates are

$$k = \mathcal{F}^{-1} \left( \frac{\mathcal{F}(x) + \mathcal{F}(\nabla) \overline{\mathcal{F}(u)}}{1 + \mathcal{F}(\nabla) \overline{\mathcal{F}(\nabla)}} \right),$$

$$u = \frac{\eta}{2\psi + \eta} \nabla k, \tag{46}$$

$$x = \frac{2m^2 \odot y + \eta k}{2m^2 + 2n^2 + \eta}.$$

The fused image estimate for this diffusion step is

$$\hat{f}_{0|t} = x + v. \tag{47}$$

The required gradient is the negative derivative of $E(x)$ at $x = \hat{f}_{0|t} - v$:

$$\nabla_{f_t} \log p(i, v \mid f_t) = - \nabla_x E(x)\Big|_{x = \hat{f}_{0|t} - v}, \tag{48}$$

where

$$\nabla_x E = 2m^2 \odot (x - y) + 2n^2 \odot x + \psi \nabla^\top \nabla x. \tag{49}$$

After updating $x$, refresh the Laplace scales as in:

$$\gamma = \frac{1}{HWN} \sum_{i,j} \mathbb{E}[m_{ij}], \qquad \rho = \frac{1}{HWN} \sum_{i,j} \mathbb{E}[n_{ij}]. \tag{50}$$

The new $\gamma, \rho$ enter the next E-step.

Practical recap:

1. Initialise $x^{(0)} = 0$ (or any prior guess) together with $\gamma, \rho, \psi, \eta$.

2. At every reverse-diffusion timestep run Eqs. equation 43–equation 48 to obtain $\hat{f}_{0|t}$ and the guidance gradient.

3. Use the gradient in the SDE integrator and proceed to the next timestep.

The $\ell_1$ fusion loss is converted to a quadratic surrogate through an EM iteration. Because the surrogate is quadratic, its gradient— given explicitly in equation 48—is available per pixel and can be inserted into the diffusion sampler without additional neural networks.

### C.2 MORE DETAILS ABOUT GRADIENT-GUIDANCE MAGNITUDES

The generalisation-error bound in equation 31 contains the term $\mathrm{Cov}\big(w_k, \ \|\nabla_{x_t} \log p(c_k \mid x_t)\| \cos \theta_{t,k}\big)$, whose interpretation hinges on the directional alignment $\cos \theta_{t,k}$. To isolate this angular factor we must show rigorously that, at every diffusion step $t$, the magnitudes $\|\nabla_{x_t} \log p(c_k \mid x_t)\|$ are (almost) the same for all modalities $k$. Below we provide two complementary arguments that justify this claim without introducing extra approximations.

$f$ denotes the fused image, $i$ and $v$ are the infrared and visible reference images, and $\varphi_t > 0$ is the step-dependent fusion ratio chosen by the scheduler. All norms on gradients are ordinary $\ell_2$ norms taken over the $N$ pixels, while the loss is measured in $\ell_1$.

The data-consistency term used by DDFM is the classic two-branch $\ell_1$ objective

$$L_t(f) = \|f - i\|_1 \ + \ \varphi_t \|f - v\|_1, \qquad \varphi_t > 0. \tag{51}$$

Except on a measure-zero set where $\ell_1$ is not differentiable,

$$\big|\partial_{f_p} |f_p - i_p|\big| = 1, \qquad \big|\partial_{f_p} \varphi_t |f_p - v_p|\big| = \varphi_t. \tag{52}$$

Hence every pixel gradient of the infrared branch has magnitude 1, while the visible branch has magnitude $\varphi_t$, independent of image content.

Summing the squared pixel-wise magnitudes over $N$ pixels gives

$$\big\|\nabla_f \|f - i\|_1\big\|_2 = \sqrt{N}, \qquad \big\|\nabla_f \varphi_t \|f - v\|_1\big\|_2 = \varphi_t \sqrt{N}. \tag{53}$$

Thus any discrepancy in the branch norms is a known scalar factor of $\varphi_t$.

Define re-scaled, non-negative weights

$$\tilde{w}_{IR} = \frac{w_{IR}}{S_t}, \quad \tilde{w}_{VI} = \frac{\varphi_t \, w_{VI}}{S_t}, \quad S_t = w_{IR} + \varphi_t \, w_{VI}. \tag{54}$$

Because $\tilde{w}_{IR} + \tilde{w}_{VI} = 1$, the combined gradient becomes

$$\tilde{w}_{IR} \, g_{\mathrm{IR}} + \tilde{w}_{VI} \, g_{\mathrm{VIS}} = \tfrac{1}{S_t}\Big( w_{IR} \, g_{\mathrm{IR}} + w_{VI} \, g_{\mathrm{VIS}} \Big), \tag{55}$$

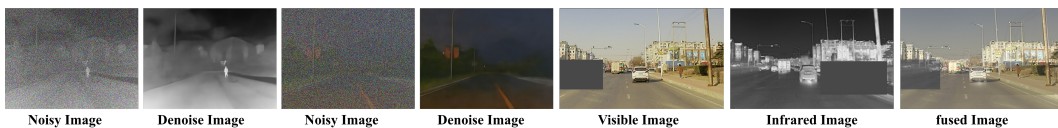

Noisy Image  Denoise Image  Noisy Image  Denoise Image  Visible Image  Infrared Image  fused Image

Figure 7: The robustness of single pre-trained diffusion model.

i.e. it is altered only by a scalar common to all modalities. Such a scalar vanishes in the cosine $\cos\theta_{t,k}$, hence the covariance in Eq. (21) depends only on the directional component.

Because the gradients of an $\ell_1$ loss have constant pixel magnitudes, their global $\ell_2$ norms differ by at most the known scalar $\varphi_t$. Re-scaling the branch weights therefore yields magnitudes that are effectively identical, justifying the step-independent constant $s_t$.

Many diffusion works replace Eq. equation 51 by a surrogate solved with iteratively reweighted least-squares (Charbonnier et al., 1997; Daubechies et al., 2004). The IRLS surrogate is analytically equivalent to the original $\ell_1$ problem. Classifier-free guidance (Ho & Salimans, 2022) and its descendants (Chung et al., 2022; Kawar et al., 2022; Gao et al., 2025) exploit exactly this observation: they balance data and prior gradients with a single scalar at each step. Whether one solves the exact $\ell_1$ problem or its IRLS surrogate , the guiding gradients entering Eq. (21) are provably or empirically equal in magnitude. The analysis of directional alignment $\cos\theta_{t,k}$ is therefore well-founded.

### C.3 More details about Pseudocode flow

---

**Algorithm 1** dig2DIG

---

**Require:** Multimodal sources $\{c_k\}_{k=1}^{K}$; diffusion model $\epsilon_\theta$; total steps $T$; DIG update interval $S$; discrepancy metric $l(\cdot,\cdot)$
**Ensure:** Fused image $x_0$
1: Pre-compute variance schedule $\{\alpha_t, \bar{\alpha}_t\}_{t=1}^{T}$
2: $x_T \sim \mathcal{N}(0, I)$ ▷ start from pure noise
3: $w_k \leftarrow \frac{1}{K}, \ k = 1, \ldots, K$ ▷ uniform init
4: **for** $t = T, T-1, \ldots, 1$ **do**
5:     **if** $t \bmod S = 0$ **then** ▷ update DIG every $S$ steps
6:         **for** $k = 1$ **to** $K$ **do** ▷ single-modal reconstructions
7:             $c_k^t \leftarrow \sqrt{\bar{\alpha}_t}\, c_k + \sqrt{1-\bar{\alpha}_t}\, \epsilon, \ \epsilon \sim \mathcal{N}(0, I)$
8:             $\hat{c}_k^t \leftarrow \frac{1}{\sqrt{\bar{\alpha}_t}}\big(c_k^t - \sqrt{1-\bar{\alpha}_t}\, \epsilon_\theta(c_k^t, t)\big)$
9:             $\mathrm{DIG}_k \leftarrow l(\hat{c}_k^t, c_k)$
10:         **end for**
11:         $w_k \leftarrow \dfrac{\exp(\mathrm{DIG}_k)}{\sum_{j=1}^{K} \exp(\mathrm{DIG}_j)}, \ \forall k$ ▷ softmax normalisation
12:     **end if**
13:      ▷ one reverse-diffusion step with guidance
14:     $\mathrm{SCORE}_{\text{uncond}} \leftarrow -\dfrac{\epsilon_\theta(x_t, t)}{\sqrt{1-\bar{\alpha}_t}}$
15:     $\mathrm{SCORE}_{\text{multi}} \leftarrow \sum_{k=1}^{K} w_k\, \mathrm{CONDSCORE}(x_t, c_k)$ ▷ EM-based
16:     $z \sim \mathcal{N}(0, I)$
17:     $x_{t-1} \leftarrow \dfrac{1}{\sqrt{\alpha_t}}\, x_t + \sigma(t)\, z + \dfrac{1-\alpha_t}{\sqrt{\alpha_t}}\big(\mathrm{SCORE}_{\text{uncond}} + \mathrm{SCORE}_{\text{multi}}\big)$
18: **end for**
19: **return** $x_0$

---

### C.4 MORE DETAILS ABOUT SINGLE PRE-TRAINED DIFFUSION MODEL

Figure 7 demonstrates the strong restoration capability obtained with a single diffusion model, underscoring the robustness of our method. For both infrared and visible images, even after noise is added, the diffusion model is able to reconstruct them faithfully. We also present scenarios in which portions of one modality are missing; the high-quality reconstructions further verify the robustness of our approach.

Although we employ only one pre-trained diffusion model, its training on a large and diverse dataset enables it to model a broad image manifold and thus recover information across multiple modalities. Because a single diffusion model already possesses this capability, we adopt one pre-trained diffusion model to keep the overall pipeline concise.

### C.5 MORE DETAILS ABOUT EXPERIMENTS

**Hardware and Software.** All experiments were conducted on a single NVIDIA RTX A6000 GPU. We used PyTorch 2.4.1 built with CUDA 12.1 and cuDNN 9.1. Unless otherwise noted, inference ran in FP32 with batch size 1.

**Backbone, Sampler, and Step Budgets.** For a fair comparison across diffusion-based methods (DDFM, CCF, and our Dig2DIG), we reuse the same unconditional pretrained diffusion checkpoint `256x256_diffusion_uncond.pt` without any fine-tuning, and we adopt DDIM sampling for all methods. DDFM is run with its default 100 reverse steps, CCF with its default 300 steps, and Dig2DIG with a total of 25 reverse steps ("DIG-25"). DIG is refreshed every $S=10$ steps, which we found to offer the best quality–cost trade-off; early, high-noise stages use larger denoising steps while later stages use smaller steps to capture fine details.

**I/O Resolution and Pre-/Post-Processing.** For every dataset and method, the model operates at native image resolution: inputs are fed at their original size and the fused outputs have exactly the same height and width. Images are read as `float32`, normalized to $[-1, 1]$ before diffusion inference, and de-normalized back to the original scale when writing results. Unless otherwise specified, we fix the random seed to 42 for reproducibility.

**Fairness Controls.** To isolate guidance effects, we keep the checkpoint, sampler, and I/O resolution identical across DDFM, CCF, and Dig2DIG; only the guidance terms differ. This ensures that observed performance gaps are not attributable to backbone capacity or training.

**Evaluation Protocol and Timing.** For visible–infrared fusion (VIF) we report PSNR, SSIM, MSE, Nabf, CC, and LPIPS; for multi-focus (MFF) and multi-exposure (MEF) we report SD, EI, EN, AG, SF, and MI. Following DDFM and CCF, SSIM is computed as the *sum over two references* to maintain protocol consistency, which can yield values greater than 1. Wall-clock runtime is measured per image on the RTX A6000 at batch size 1 after a brief warm-up and excludes data I/O; where applicable we also report TFLOPs alongside wall-clock time.

## D MORE RESULTS

### D.1 MORE QUALITATIVE COMPARISONS

As shown in 8 on the M3FD dataset, our method is the only one that simultaneously preserves sky textures, produces clear building structures, and retains fine infrared details in smoky regions. In contrast, CCF and related methods blur the buildings, while DCEvo, LFDT-Fusion, and MMDRFuse lose infrared details inside the smoke; CDDFuse and SwinFusion fail to maintain the texture in the sky.

As illustrated in 9 on the LLVIP dataset, our method is the only one that keeps both vehicle details and pedestrian details intact. MUFusion, LFDT-Fusion, and TC-MoA over-emphasize the infrared

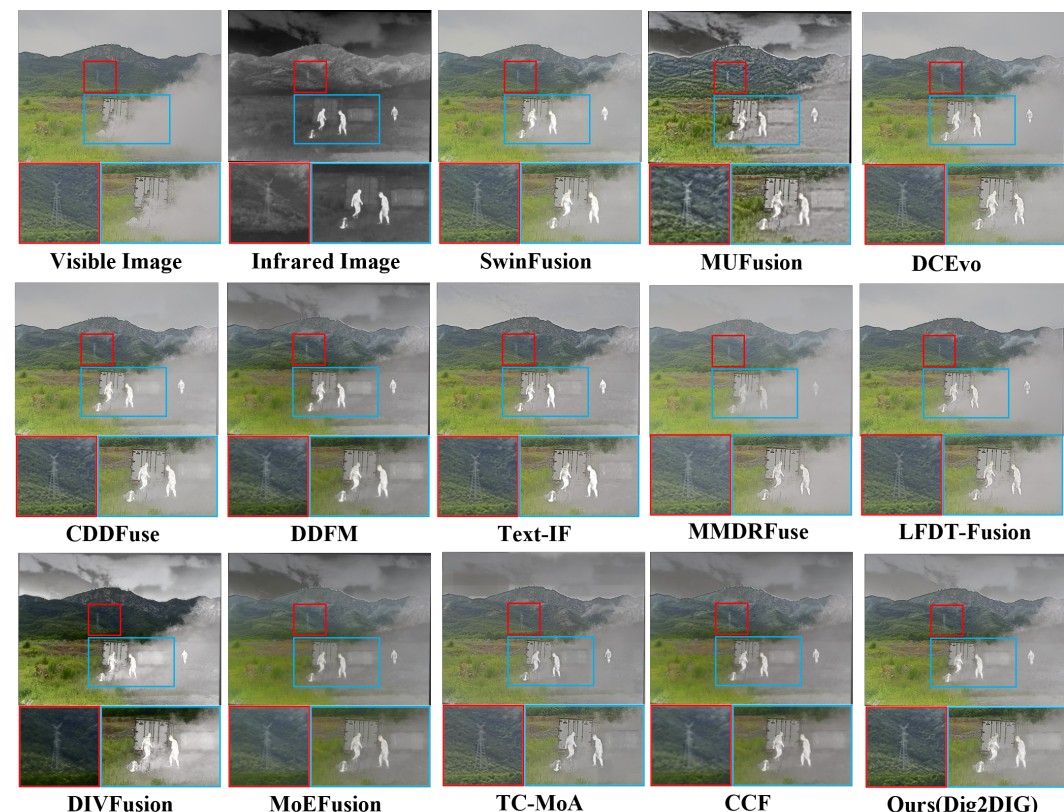

Figure 8: More qualitative comparisons of our method on M3FD datasets.

modality, which suppresses facial details of pedestrians, whereas DCEvo, CCF, and DDFM lose critical vehicle structures.

As shown in 10 On the MARS dataset, our method preserves both plant textures and human facial details. Most competing approaches—such as DCEvo, LFDT-Fusion, MMDRFuse, DDFM, DIV-Fusion, and CCF—are overly influenced by the infrared modality and therefore lose facial details. Compared with MoEFusion, our method also produces plant colors that are closer to those in the original visible modality.

As illustrated in 11 On the MFFW multi-focus dataset, our method maintains sharp textures and faithful colors, while CCF exhibits noticeable color distortion.

As shown in 12 On the MEFB multi-exposure dataset, our method achieves the most natural illumination transitions. Methods including CCF, DDFM, TTD, FusionDN, and U2Fusion show unnatural lamp-light transitions near the desk lamp region; compared with DeFusion and TC-MoA, our results contain fewer noise artifacts on the sofa. Overall, these visual comparisons further demonstrate the effectiveness and robustness of our approach across diverse fusion tasks.

## D.2 EXPERIMENTS ON DATASETS WITH MORE THAN TWO MODALITIES

To verify the scalability of our method beyond bimodal settings, we further conduct experiments on a multi-exposure benchmark with more than two modalities (Cai et al., 2018). As shown in 13 the qualitative results demonstrate that Dig2DIG can effectively fuse complementary information from multiple exposures, confirming its extensibility to $K > 2$ modality fusion scenarios.

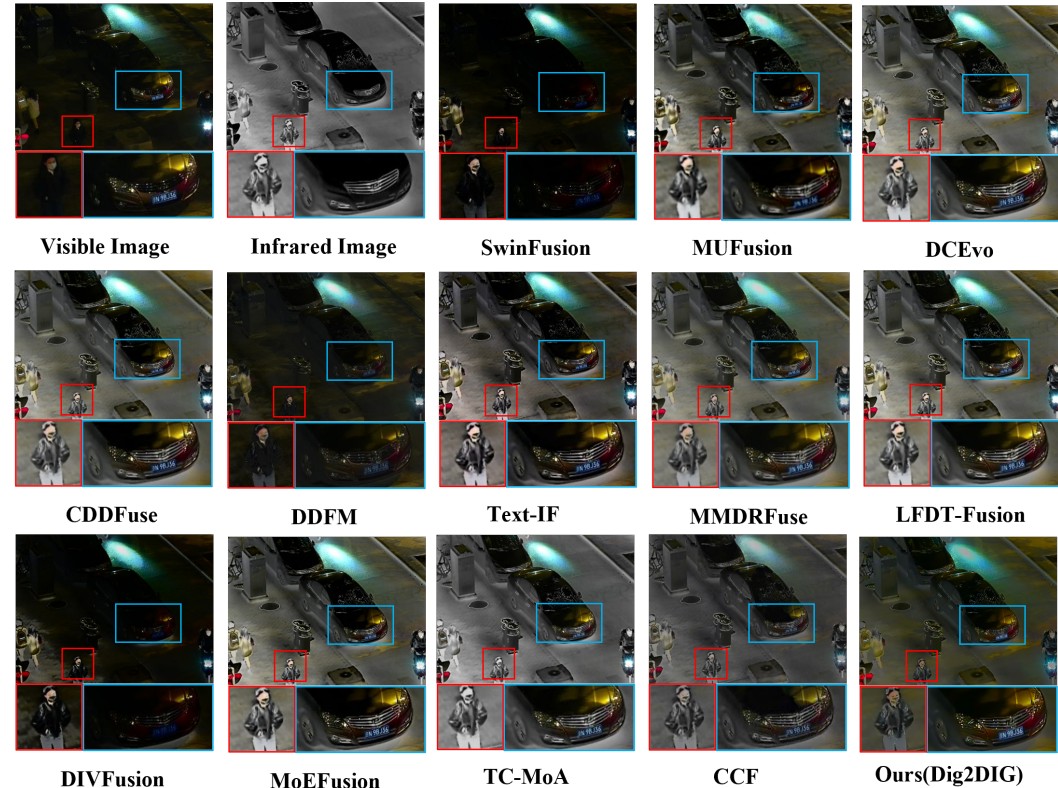

**Figure 9:** More qualitative comparisons of our method on LLVIP datasets.

| Method | LLVIP Dataset | | | | | | M3FD Dataset | | | | | | MSRS Dataset | | | | | |
|---|---|---|---|---|---|---|---|---|---|---|---|---|---|---|---|---|---|---|
| | PSNR↑ | SSIM↑ | MSE↓ | Nabf↓ | CC↑ | LPIPS↓ | PSNR↑ | SSIM↑ | MSE↓ | Nabf↓ | CC↑ | LPIPS↓ | PSNR↑ | SSIM↑ | MSE↓ | Nabf↓ | CC↑ | LPIPS↓ |
| $\mathrm{Cov}(w_k, I_{k,t}) < 0$ | 31.63 | 1.14 | 2331 | 0.015 | 0.65 | 0.380 | 30.24 | 1.35 | 2482 | 0.029 | 0.51 | 0.339 | 36.84 | 1.37 | 1586 | 0.010 | 0.58 | 0.301 |
| $\mathrm{Cov}(w_k, I_{k,t}) = 0$ | 36.10 | 1.18 | 2056 | 0.004 | 0.67 | 0.310 | 30.87 | 1.40 | 2221 | 0.007 | 0.56 | 0.303 | 38.19 | 1.39 | 1367 | 0.004 | 0.66 | 0.287 |
| $\mathrm{Cov}(w_k, I_{k,t}) > 0$ | 33.74 | 1.23 | 1464 | 0.001 | 0.73 | 0.298 | 31.83 | 1.41 | 2216 | 0.009 | 0.57 | 0.287 | 39.07 | 1.42 | 1366 | 0.001 | 0.63 | 0.282 |

Table 8: Effect of weight–DIG covariance on fusion performance on LLVIP, M3FD, and MSRS.

### D.3 EFFECT OF WEIGHT–DIG COVARIANCE ON FUSION PERFORMANCE

To further corroborate Theorem 1, we perform an ablation study under three covariance regimes between the guidance weight $w_k$ and the residual information proxy $\mathrm{DIG}_k(t)$:

(i) Positive covariance $(\mathrm{Cov}(w_k, I_{k,t}) > 0)$: the proposed DIG-based softmax;

(ii) Zero covariance $(\mathrm{Cov}(w_k, I_{k,t}) = 0)$: fixed, uniform weights $w_k = 1/K, \ \forall k$;

(iii) Negative covariance $(\mathrm{Cov}(w_k, I_{k,t}) < 0)$: an inverse-DIG softmax

$$w_k^{\mathrm{neg}}(t) = \frac{\exp\big(-\mathrm{DIG}_k(t)\big)}{\sum_{j=1}^{K} \exp\big(-\mathrm{DIG}_j(t)\big)}. \tag{56}$$

All other hyper-parameters are kept identical across settings. Experiments are conducted on the **LLVIP**, **M3FD**, and **MSRS** datasets, and the quantitative results are summarised in Table 8.

Across all three benchmarks, the positive-covariance scheme attains the best fusion accuracy, the uniform scheme ranks second, and the negative-covariance scheme performs worst. This ordering aligns with Theorem 1: a larger (positive) $\mathrm{Cov}(w_k, I_{k,t})$ lowers the generalisation-error upper bound, whereas a negative covariance increases it, thereby empirically validating our theoretical findings.

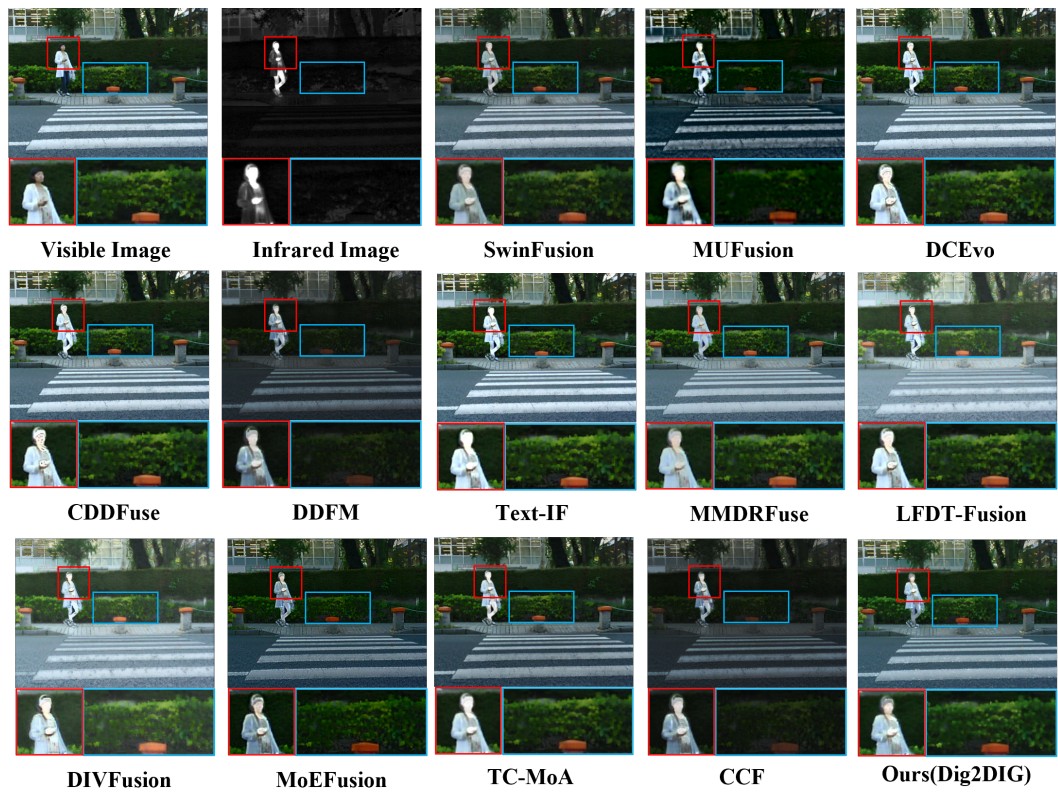

Figure 10: More qualitative comparisons of our method on MSRS datasets.

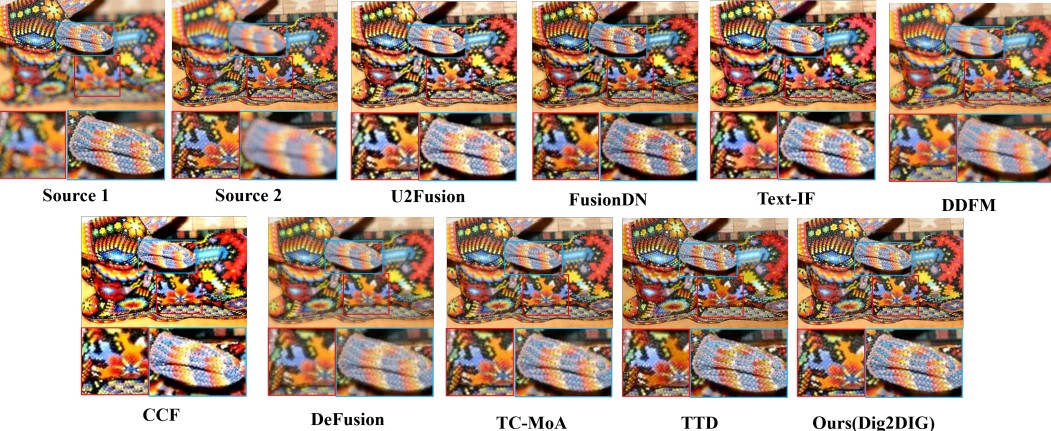

Figure 11: More qualitative comparisons of our method on MFFW datasets.

### D.4 MORE RESULTS ABOUT DIG VISUALIZATION

Under identical experimental settings, we compute at each reverse step $t$ the ratio of guidance-gradient norms between the infrared and visible modalities, $\rho_t = \frac{\|\nabla_{x_t} \log p(c_{\mathrm{ir}}|x_t)\|}{\|\nabla_{x_t} \log p(c_{\mathrm{vi}}|x_t)\|}$, and report its mean over all steps for each dataset. The dataset-wise averages are

$$\left(\overline{\rho}^{\mathrm{LLVIP}}, \overline{\rho}^{\mathrm{M3FD}}, \overline{\rho}^{\mathrm{MSRS}}\right) = (1.06, 0.98, 1.04).$$

Values close to unity indicate that the gradient-norm approximation used in our derivations is reasonable.

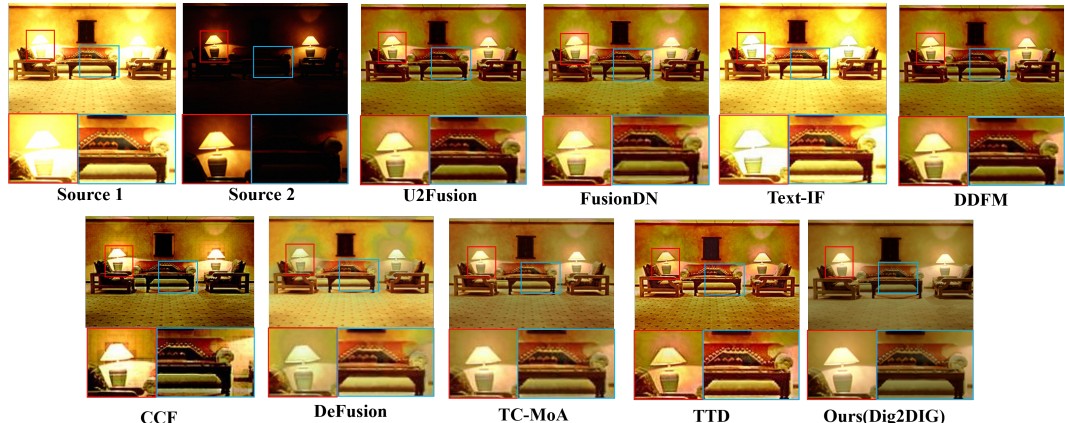

Figure 12: More qualitative comparisons of our method on MEFB datasets.

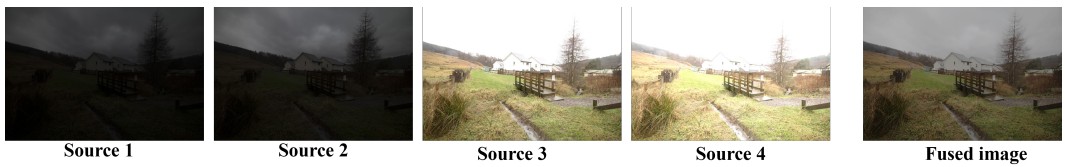

Figure 13: Qualitative results on datasets with more than Two modalities

## D.5 MORE RESULTS ABOUT DIG AND $w_k$ VISUALIZATION

As shown in 15, we present additional visualization results for $w_k$. In the early denoising stage, $w_k$ accurately highlights the salient pedestrians in the infrared modality while capturing the global scene structure in the visible modality. Because the visible images in this example are low-light and lack fine details, $w_k$ increasingly focuses on infrared details as denoising proceeds and the coarse structure is reconstructed. Fig. 16 additionally provides the visualization of the raw DIG maps, where $w_k$ is obtained by applying a pixel-wise Softmax to DIG across modalities.

## E OBJECT DETECTION EXPERIMENT

To evaluate the usability of the fusion results, we use a pre-trained YOLOv5 model to perform pedestrian detection on the LLVIP dataset. The results are shown in 14, demonstrating the usability of the fusion results.

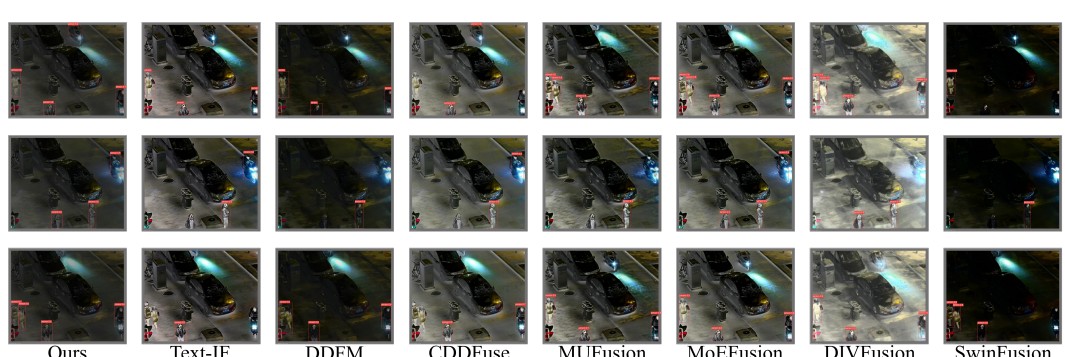

| Ours | Text-IF | DDFM | CDDFuse | MUFusion | MoEFusion | DIVFusion | SwinFusion |

Figure 14: Object detection comparison of our method and the recent proposed competing approaches on LLVIP dataset.

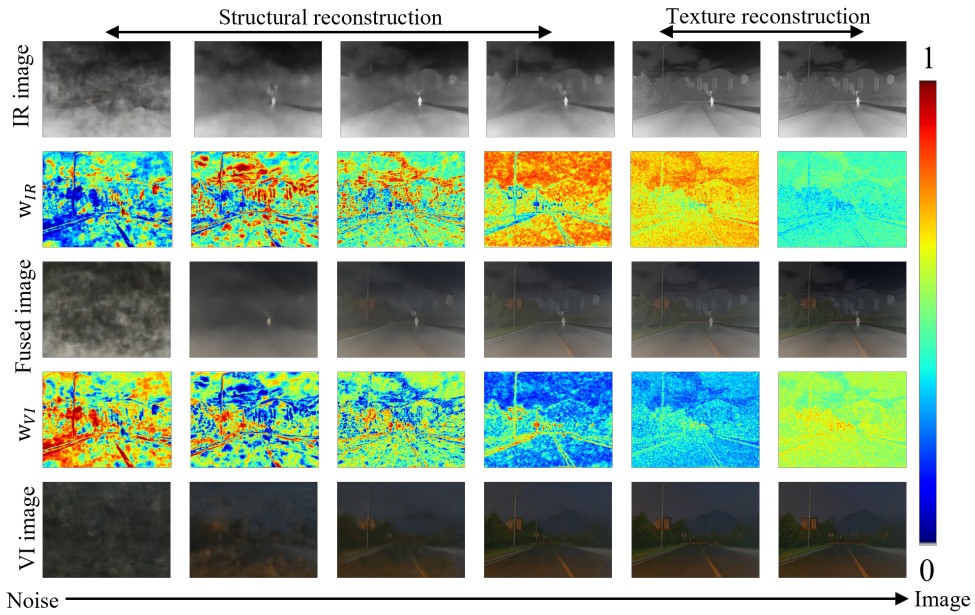

Figure 15: $w_k$ visualization during denoising steps.

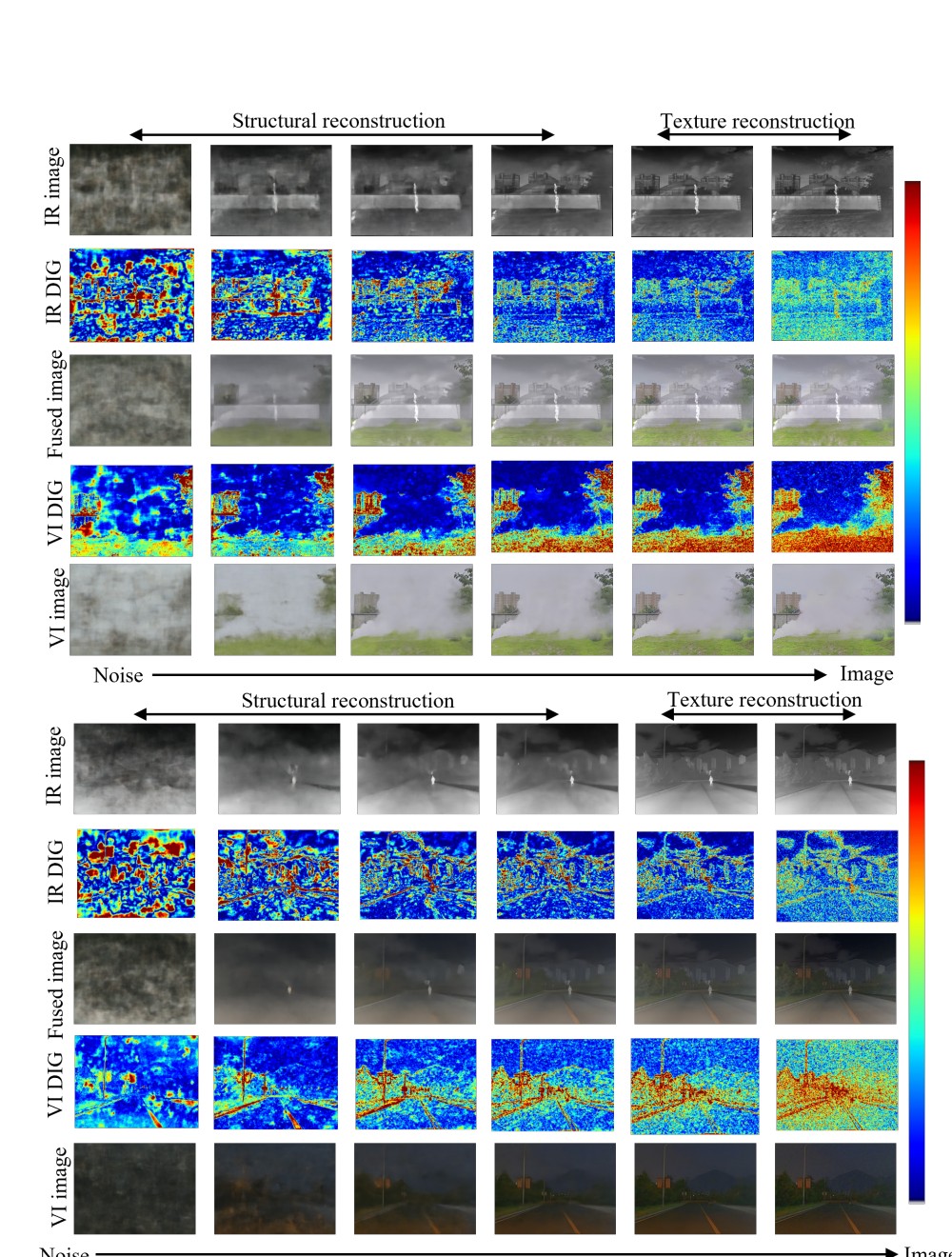

Figure 16: DIG visualization during denoising steps.

