# OpenReview forum: "Dig2DIG: Dig into Diffusion Information Gains for Image Fusion"
_ICLR.cc/2026/Conference — Submitted to ICLR 2026_

### Official Review · Reviewer_UcWT · 2025-10-30

**Soundness:** 2
**Presentation:** 3
**Contribution:** 2
**Rating:** 4
**Confidence:** 5

**Summary:**

Due to the spatiotemporal imbalance of information gain contributed by different image modalities during the denoising diffusion process, this paper proposes a diffusion-model-based dynamic fusion method that computes a distinct set of fusion coefficients at each diffusion time step, thereby optimizing modality-specific information gain across the different stages of the diffusion process.

**Strengths:**

- Overall, the discussion on dynamic fusion during the diffusion denoising phase is meaningful, as it aims to enhance the information gain achieved through fusion.
- The key idea of the proposed method for tightening the upper bound of fusion generalization is to ensure that the guidance weight assigned to each modality is positively correlated with the residual information of that modality that has not yet been integrated into the current image, and theoretical support for this is provided.
- Modality residual information is proxied by the information gain obtained during the diffusion process.

**Weaknesses:**

- Using DIG as a proxy for dynamic weighting is not entirely reasonable, since DIG only measures the information loss during the forward noising process. Intuitively, it is more correlated with the intrinsic information richness of the original image. Consequently, modality images containing richer information regions would consistently receive higher fusion weights at any diffusion timestep. This contradicts the methodological insight that different diffusion stages exhibit distinct generation dynamics, where low-frequency structures and high-frequency textures evolve at different rates.
- The illustration in Figure 1 may cause misunderstanding, as the information gain shown in the figure is different from the Diffusion Information Gain (DIG). It would be helpful for the authors to additionally indicate the computation method of the information gain depicted in the figure, so as to facilitate clearer understanding and distinction between the two concepts.
- An important question to consider is how the authors construct the loss when computing DIG, and whether using only the ℓ2 loss is reasonable. The dynamic weights derived solely from the ℓ2 loss merely reflect pixel-level differences, and it remains questionable whether directly employing such differences as dynamic weights to modulate other types of losses, such as gradient losses, is theoretically and practically sound.
- The authors have considered the dynamic weights in isolation, whereas in practice, the fusion weights are largely dependent on the formulation of the fusion loss constraints. However, the authors did not design targeted conditional constraints; instead, they directly adopted the conditional constraint scheme from DDFM
- The selection of comparison methods in the experiments appears somewhat inconsistent. For example, Text-IF is used as a comparison method for multi-focus and multi-exposure image fusion, even though it is not a unified fusion framework. In contrast, CCF, which is indeed a unified approach, is only compared in the task of infrared–visible image fusion.

**Questions:**

- Is it reasonable to use DIG as a substitute for the dynamic weights?
- Is it reasonable to compute DIG using only the ℓ2 loss? For instance, can it effectively measure the information loss in texture gradients?
- It is not clearly explained how the dynamic weights are incorporated into the EM conditions; this part requires further clarification. In addition, it remains unclear whether such dynamic weights are compatible with the conditional constraint process—for example, with the gradient penalty term in the EM algorithm.
- Is the selection of comparison algorithms fair?

---

> ### Author Response · Authors · 2025-11-23
> **Authors' Rebuttal to Reviewer UcWT (1/7)**
>
> We thank the reviewer for the careful and detailed review. We are pleased that you find **our discussion of dynamic fusion during reverse diffusion meaningful**, and that you recognize **the key theoretical idea of tightening the generalization upper bound by aligning modality weights with unfused residual information**, as well as **the use of diffusion information gain to operationalize this principle**. Meanwhile, we also appreciate the questions and suggestions you raised. We take them seriously and respond point by point below, with corresponding clarifications and additions in the revision.
>
>
> >**W1 DIG is not a static “richness” score.**
>
> **Response W1.** We appreciate the reviewer’s insightful comment and would like to clarify, with supporting evidence, why DIG does *not* reduce to a static measure of modality richness.
>
> **(1) DIG is a reverse-time, one-step residual—not a forward noising loss.**
>
> By definition, $\mathrm{DIG}_k(t)=l(\hat c\_k^{t},c\_k)$ depends on the reverse one-step reconstruction$\hat c\_k^{t}$
> produced by $\epsilon\_\theta(\cdot,t)$ at the current timestep $t$. Therefore, $\mathrm{DIG}_k(t)$ directly measures **how much modality-$k$ information remains unfused at step $t$ along the actual reverse trajectory**. It is explicitly **time- and model-dependent**, rather than a time-invariant proxy of “intrinsic richness.” This matches our theory: Eq. (4) only requires the guidance weight $w\_k(t)$ to be positively correlated with the unfused residual information $I\_{k,t}$, and DIG serves as an observable, monotone surrogate of $I\_{k,t}$.
>
> **(2) DIG naturally follows stage-wise diffusion dynamics and spatial heterogeneity.**
> Reverse diffusion recovers low-frequency structure earlier and high-frequency details later. Consequently, once a modality’s structural (low-frequency) content has been largely absorbed at early steps, its $\mathrm{DIG}_k(t)$ decreases; at later steps, a modality that contributes more fine textures will exhibit a higher residual, raising its $\mathrm{DIG}_k(t)$. Mapping DIG to weights via softmax  thus yields time-dependent weight reallocation.
> Moreover, because we compute DIG region-wise, the dominant modality can vary across spatial regions and timesteps. This is visualized in Fig. 5 , Fig. 15 (weight maps) and Fig. 16 (raw DIG): early steps emphasize salient IR structures, while later steps shift attention toward VI textures.
>
> **(3) Persistent preference for one modality can be correct in extreme cases.**
> When one modality contains little usable information (e.g., VI under extremely low illumination), it is desirable for the other modality to dominate throughout denoising. This does not contradict “dynamics”: dynamic guidance does not require frequent rank flips. It can also manifest as magnitude adaptation over $t$ and space, even if the leading modality remains the same. Theoretically, Eq. (4) only asks for $\mathrm{Cov}(w_k,I_{k,t})>0$ to tighten the bound; if one modality consistently carries more unfused information, maintaining a larger weight is exactly the right behavior.
>
> **(4) Empirical evidence and ablations.**
>
> * **Spatio-temporal visualizations:** Fig. 5 , Fig. 15 and Fig. 16 show clear temporal shifts and spatial complementarity in $w_k(t)$ or DIG.
> * **High dynamicity helps most:** Tab. 5 shows that combining region-wise and time-wise weighting outperforms either dimension alone, confirming the importance of both temporal and spatial dynamics.
> * **Covariance-controlled ablation:** Tab. 8 and Fig. 6 demonstrates a strict monotonic order—positive-covariance (DIG-Softmax or $I\_{k,t}$-Softmax) $>$ zero-covariance (uniform) $>$ negative-covariance (inverse-DIG or inverse $I\_{k,t}$)—exactly as predicted by Eq. (4).
> * **Refresh interval:** Tab. 3 shows that a moderate DIG refresh interval best tracks late-stage detail recovery while avoiding unstable early-stage estimates.
>
> **(5) Relation to “intrinsic richness.”**
>
> At any fixed $t$, $\mathrm{DIG}_k(t)$ indeed correlates with **how much modality information remains to be injected at that time**. But it is **not** a static property of the clean input independent of the reverse process. Additional results in Tab. 6 and Tab. 4 confirm that, across different monotone mappings and residual metrics, positively aligning weights with residual magnitude consistently improves fusion quality.
>
> In summary, DIG does **not** collapse to “static richness.” It is a time- and space-dependent estimate of **currently unfused information**, inherently aligned with diffusion’s stage-wise dynamics and our covariance-based theory, and supported by targeted visualizations and ablations.

---

> ### Author Response · Authors · 2025-11-23
> **Authors' Rebuttal to Reviewer UcWT (2/7)**
>
> >**W2 Definition of “Information Gain” in Fig. 1**
>
> **Response W2.** Thank you for the careful observation. We agree that the “information gain” visualized in Fig. 1 is not exactly the same as the Diffusion Information Gain (DIG) defined later. For clarity, the “information gain” in this figure is computed as the $\ell_2$ difference between the denoised estimate at timestep $t$ and a reference denoised estimate at $t_0=T/2$ during the reverse process, i.e., $\Delta I_k(t)=|\hat c_k^{,t}-\hat c_k^{,t_0}|_2$. We use this quantity only for an intuitive illustration: it makes the spatio-temporal imbalance of modality contributions along denoising easier to perceive, whereas DIG measures residual (yet-unabsorbed) modality information and is theoretically grounded but less immediately interpretable as a visualization. We have revised the caption around Fig. 1 to explicitly distinguish this illustrative “information gain” from DIG. Raw DIG visualizations are provided separately (see Fig. 16) for readers who want to inspect the absolute residual-gain behavior.

---

> ### Author Response · Authors · 2025-11-23
> **Authors' Rebuttal to Reviewer UcWT (3/7)**
>
> > **W3 On the loss used for DIG and whether ℓ2-only is reasonable.**
>
> **Response W3.** Thank you for the thoughtful question. We agree that the loss used to compute DIG matters, because DIG is intended to estimate “how much useful modality information remains unfused at step (t).” In the original submission, we used an ℓ2 residual for DIG mainly for **simplicity and stability**: (i) ℓ2 provides a clean, low-variance proxy of one-step reconstruction error, and (ii) it allows us to isolate the effect of dynamic weighting without introducing extra hyper-parameters.
>
> That said, our EM-based guidance energy contains both a pixel fidelity term and a modality-wise TV term (edge-preserving smoothing). Following your suggestion, we implemented an **energy-aligned DIG** variant where the DIG loss matches the guidance energy, i.e., using an ℓ2+TV residual (with the same TV weight as in EM). Intuitively, this encourages DIG to respond not only to pixel discrepancies but also to structural/edge cues that are important for fusion guidance.
>
> We report the averaged results on LLVIP below:
>
> | Metric | DIG with ℓ2 only | DIG with ℓ2+TV (aligned) |       Δ |
> | ------ | ---------------: | -----------------------: | ------: |
> | PSNR↑  |          33.7490 |              **33.8461** | +0.0971 |
> | SSIM↑  |           1.2328 |               **1.2361** | +0.0033 |
> | MSE↓   |        1464.3591 |            **1460.2319** | −4.1272 |
> | Nabf↓  |           0.0001 |                   0.0001 |  0.0000 |
> | CC↑    |           0.7309 |               **0.7356** | +0.0047 |
> | LPIPS↓ |           0.2981 |               **0.2961** | −0.0020 |
>
> These gains are small but consistent across metrics, indicating that aligning DIG with the guidance energy can better capture structural residuals and provide modest additional benefits. Meanwhile, ℓ2-only DIG remains a simple and reliable residual proxy, showing that DIG-driven dynamic guidance is robust to reasonable loss choices rather than being tied to a specific form. We sincerely thank the reviewer for this insightful and experience-based suggestion, which helped us make this point clearer and better supported.

---

> ### Author Response · Authors · 2025-11-23
> **Authors' Rebuttal to Reviewer UcWT (4/7)**
>
> > **W4 On Reusing DDFM Constraints**
>
> **Response W4.** Thank you for this important point. We agree that fusion weights should be interpreted jointly with the conditional constraints they modulate. In our method, DIG does not introduce a separate objective; instead, it operates within the same conditional objective and guidance formulation to dynamically rebalance each modality’s contribution per step and per region. Thus, DIG-based weighting is a principled reallocation of guidance strength under a fixed fusion constraint set.
>
> We intentionally reuse the DDFM conditional constraints as a strong and widely adopted baseline to isolate the effect of dynamic weighting in a strictly controlled setting. With the sampler and conditional formulation kept identical, the only change is static versus DIG-driven weights, so the consistent improvements can be attributed to the proposed dynamic principle.
>
> Moreover, our theory is developed for a generic fusion objective and is not tied to any DDFM-specific form. The DIG weighting mechanism is therefore constraint-agnostic and plug-and-play: as long as modality-specific conditional terms are available, it can be used to dynamically reweight other reasonable fusion constraints as well. DDFM is used here mainly as a clean and fair experimental anchor.

---

> ### Author Response · Authors · 2025-11-23
> **Authors' Rebuttal to Reviewer UcWT (5/7)**
>
> >**W5 Baseline Consistency Across Tasks**
>
> **Response W5.** Thank you for pointing out this inconsistency. We agree that, when evaluating a unified fusion framework, the comparison set should be as consistent as possible across tasks. Following your suggestion, we have **added CCF**, a unified diffusion-based fusion method, as an additional baseline in our **multi-focus** and **multi-exposure** experiments. The updated quantitative results are:
>
> **MFFW Dataset**
>
> | Method      |       SD↑ |       EI↑ |      EN↑ |      AG↑ |       SF↑ |      MI↑ |
> | ----------- | --------: | --------: | -------: | -------: | --------: | -------: |
> | CCF         |     69.71 |     14.85 |     7.75 |     6.70 |     21.49 |     4.23 |
> | **Dig2DIG** | **72.95** | **16.64** | **7.87** | **6.75** | **22.60** | **5.97** |
>
> **MEFB Dataset**
>
> | Method      |       SD↑ |       EI↑ |      EN↑ |      AG↑ |       SF↑ |      MI↑ |
> | ----------- | --------: | --------: | -------: | -------: | --------: | -------: |
> | CCF         |     71.01 |     19.99 |     7.35 |     8.03 |     22.71 |     3.93 |
> | **Dig2DIG** | **75.05** | **20.21** | **7.38** | **8.10** | **23.60** | **6.87** |
>
> As shown, Dig2DIG consistently outperforms CCF across all metrics on both datasets, confirming the advantage of our DIG-driven dynamic weighting under a stronger unified baseline.
>
> Regarding Text-IF, we included it because we trained it on the corresponding multi-focus and multi-exposure datasets, strictly following its intended per-task training setting. Although Text-IF is not a unified framework, this task-matched retraining ensures it serves as a strong baseline and keeps the comparison fair.

---

> ### Author Response · Authors · 2025-11-23
> **Authors' Rebuttal to Reviewer UcWT (6/7)**
>
> >**Q1 Reasonableness of using DIG for dynamic weights.**
>
> **Response Q1**. Yes. DIG is defined on the reverse trajectory as a one-step residual, $\mathrm{DIG}\_k(t)=l(\hat c\_k^{,t},c\_k)$, so it measures how much modality-$k$ information remains unfused at step $t$ and region $p$, rather than a static “intrinsic richness” score. Therefore mapping DIG to $w\_k(t)$ yields time- and region-adaptive guidance consistent with stage-wise diffusion dynamics. A detailed clarification and supporting evidence are provided in **W1**.
>
> >**Q2 Reasonableness of ℓ2-only DIG.**
>
> **Response Q2**. We agree the DIG should reflect the guidance energy. ℓ2-only DIG was used originally for simplicity and stability, serving as a low-variance residual proxy. Following the reviewer’s suggestion, we also tested an energy-aligned DIG that uses the same ℓ2+TV form as our EM guidance; it yields small but consistent gains while preserving the same trend, indicating robustness to reasonable DIG computation forms. A detailed clarification are reported in **W3**.

---

> ### Author Response · Authors · 2025-11-23
> **Authors' Rebuttal to Reviewer UcWT (7/7)**
>
> > **Q3 How dynamic weights are incorporated into EM, and compatibility with conditional constraints.**
>
> **Response Q3.**  In our method, within each EM-guided reverse step, after computing per-modality conditional gradients, we reweight them by DIG to form the final conditional guidance. At each reverse step, the sampler computes, for every modality $k$, the gradient of its own regularized conditional energy (the same modality-separable objective as in DDFM, i.e., a data term plus the gradient regularizer). We then evaluate region-wise DIG at the current step, map it to non-negative, normalized weights $w_k(t)$, and linearly combine the modality gradients to form the final conditional guidance used in the reverse update.
>
> This is compatible with the conditional-constraint process because (i) the gradient penalty is an intrinsic regularizer inside each single-modality objective, so it has already shaped each modality’s gradient before weighting; our DIG weights therefore do not change the form of the constraints/objectives, and only rescale their post-regularization contributions. (ii) Diffusion guidance naturally supports linear combinations of multiple conditional gradients with coefficient-controlled strengths (e.g., CFG and multi-condition guidance). Our weighting follows this standard form, with coefficients determined by DIG. Hence, as long as the conditional constraints are defined in a modality-wise manner, DIG-based weighting can be directly applied to dynamically rebalance their guidance strengths.
>
> Finally, this also explains why energy-aligned DIG can yield further gains in practice: when DIG uses a residual consistent with the same regularized guidance energy, the resulting weights better reflect which modality still carries useful constraint-aware residual information to inject.
>
> > **Q4 Is the selection of comparison algorithms fair?**
>
> **Response Q4.** Yes. All baselines and our method are evaluated under the same hardware environment and identical testing pipeline, using the same metrics and evaluation protocol. For methods that require training, we follow their official settings and retrain them with the same dataset splits as ours to ensure a fair, task-matched comparison. We also aim for consistent baseline coverage across tasks, and have strengthened this further following the reviewer’s suggestions. Detailed experimental settings are provided in Appendix C.5 and discussed in W5.

---

> ### Author Response · Authors · 2025-11-27
> **Summary of responses to Reviewer UcWT**
>
> We sincerely thank Reviewer UcWT for the careful and insightful review. In our rebuttal and revision, we:
>
> • Clarify that DIG is defined as a reverse-time, one-step residual along the actual reverse trajectory, measuring unfused modality information rather than static “intrinsic richness”; we relate this explicitly to the residual term in our generalization bound and support it with spatio-temporal visualizations and covariance-controlled ablations **(W1, Q1)**.
>
> • Distinguish the illustrative “information gain” in Fig. 1 from the formally defined DIG: the former is an ℓ2 difference between denoised estimates at different timesteps used only for intuition, while DIG measures residual (yet-unabsorbed) modality information; we revise the caption accordingly and additionally provide raw DIG visualizations **(W2)**.
>
> • Analyze the influence of the loss used in DIG, explaining why ℓ2-only DIG was initially chosen for simplicity and stability, and add an energy-aligned ℓ2+TV variant that better matches the EM guidance energy; the ℓ2+TV DIG yields small but consistent gains while confirming that our conclusions are robust to reasonable DIG computation forms **(W3, Q2)**.
>
> • Clarify how DIG-based dynamic weights are incorporated within each EM-guided reverse step: modality-wise conditional gradients under the DDFM-style constraints are first computed and regularized, then reweighted by normalized DIG-based coefficients and linearly combined; this preserves the constraint form while rebalancing guidance strengths in a plug-and-play manner, and is fully compatible with gradient-penalty terms **(W4, Q3)**.
>
> • Strengthen the experimental protocol by adding CCF, a unified diffusion-based fusion framework, as a baseline on the multi-focus and multi-exposure tasks and showing that Dig2DIG consistently outperforms it on all metrics; we also clarify why including a task-wise trained Text-IF remains fair, and summarize the unified evaluation setting across baselines **(W5, Q4)**.

---

### Official Review · Reviewer_aQV2 · 2025-11-01

**Soundness:** 3
**Presentation:** 3
**Contribution:** 3
**Rating:** 4
**Confidence:** 3

**Summary:**

This paper proposes Dig2DIG, a dynamic image fusion framework built upon diffusion models, introducing Diffusion Information Gains (DIG) to dynamically guide the fusion process at each denoising step. The core idea is to weight modality contributions by quantifying how much information each modality can provide at each step, theoretically linking this approach to a provable reduction of the generalization error bound in multimodal fusion. Theoretical analysis, practical algorithmic realization, and extensive experimental validation are provided.

**Strengths:**

1. This paper thoroughly analyzes the theoretical motivation behind dynamic fusion in denoising diffusion models.
2. This paper systematically compares Dig2DIG with strong baselines over multiple challenging datasets.

**Weaknesses:**

1. This paper miss some discussion and comparison with several recent work, e.g., [R1-R2]
2. While the paper’s theoretical and algorithmic contributions are clear, the system-level architecture largely repurposes standard DDPM sampling with softmax weighting for fusion guidance.
3. While the mathematics generalizes to $K>2$, the paper does not present empirical or even synthetic evidence for how the framework scales with larger numbers or more heterogeneous modalities, nor does it discuss failure points in such scenarios.
4. Ablations that could further clarify the incremental importance of each design are only briefly touched upon. e.g., what if weighting is region-wise but not dynamic in time? What if different normalization or activation functions are used on DIG?
References:

[R1] Deng Y, Xu T, Cheng C, et al. Mmdrfuse: Distilled mini-model with dynamic refresh for multi-modality image fusion[C]//Proceedings of the 32nd ACM International Conference on Multimedia. 2024: 7326-7335.

[R2] Yang B, Jiang Z, Pan D, et al. LFDT-Fusion: A latent feature-guided diffusion Transformer model for general image fusion[J]. Information Fusion, 2025, 113: 102639.

**Questions:**

See the weakness

---

> ### Author Response · Authors · 2025-11-23
> **Authors' Rebuttal to Reviewer aQV2 (1/3)**
>
> We thank the reviewer for the careful review and helpful feedback. We are glad that you find our theoretical and algorithmic contributions clear, and we appreciate your recognition of the value of DIG-based dynamic weighting. Your comments also highlight several places where the paper can be strengthened in discussion, system-level clarification, and experimental support. We take these suggestions seriously and respond to them point by point below, with corresponding clarifications and additions in the revision.
>
> > **W1 Adding MMDRFuse and LFDT-Fusion as a baseline**
>
> **Response W1.** We thank the reviewer for the valuable suggestion. In the revised version, we have added **MMDRFuse ** and **LFDT-Fusion** as additional recent SOTA baselines under the same experimental settings, and report both quantitative and qualitative results for a more comprehensive comparison. Both methods are competitive; meanwhile, our Dig2DIG achieves consistent improvements across all three datasets, further supporting the effectiveness of our dynamic DIG-guided fusion. The updated numbers are summarized below:
>
> **LLVIP dataset**
>
> | Method         |     PSNR↑ |    SSIM↑ |     MSE↓ |     Nabf↓ |      CC↑ |    LPIPS↓ |
> | -------------- | --------: | -------: | -------: | --------: | -------: | --------: |
> | MMDRFuse       |     33.28 |     1.20 |     2159 |     0.025 |     0.69 |     0.302 |
> | LFDT-Fusion    |     33.31 |     1.20 |     2534 |     0.019 |     0.66 |     0.302 |
> | Dig2DIG (Ours) | **33.74** | **1.23** | **1464** | **0.001** | **0.73** | **0.298** |
>
> **M3FD dataset**
>
> | Method         |     PSNR↑ |    SSIM↑ |     MSE↓ |     Nabf↓ |      CC↑ |    LPIPS↓ |
> | -------------- | --------: | -------: | -------: | --------: | -------: | --------: |
> | MMDRFuse       |     31.51 |     1.40 |     3508 |     0.014 |     0.54 |     0.301 |
> | LFDT-Fusion    |     30.54 |     1.39 |     3714 |     0.027 |     0.47 |     0.318 |
> | Dig2DIG (Ours) | **31.83** | **1.41** | **2216** | **0.009** | **0.57** | **0.287** |
>
> **MSRS dataset**
>
> | Method         |     PSNR↑ |    SSIM↑ |     MSE↓ |     Nabf↓ |      CC↑ |    LPIPS↓ |
> | -------------- | --------: | -------: | -------: | --------: | -------: | --------: |
> | MMDRFuse       |     39.01 |     1.40 |     2199 |     0.190 |     0.60 |     0.323 |
> | LFDT-Fusion    |     38.97 |     1.41 |     2525 |     0.031 |     0.60 |     0.306 |
> | Dig2DIG (Ours) | **39.07** | **1.42** | **1366** | **0.001** | **0.63** | **0.282** |
>
> These results confirm that, even against newly added strong baselines, our method provides stable gains across datasets.

---

> ### Author Response · Authors · 2025-11-23
> **Authors' Rebuttal to Reviewer aQV2 (2/3)**
>
> >**W2. The system largely repurposes standard DDPM sampling with a softmax weighting for fusion guidance**
>
> **Response W2.** We appreciate the reviewer’s positive assessment of our theoretical and algorithmic contributions. We intentionally keep the backbone sampler standard (DDPM/DDIM) so that the novelty is concentrated on *when, where, and which modality to emphasize*, i.e., a principled dynamic guidance rule. This design lets us isolate the gains brought by DIG under a strictly controlled apples-to-apples setting. Our system differs fundamentally from a “fixed guidance with a cosmetic softmax” reuse in three aspects:
>
> **1. Why fixed guidance/weights are suboptimal for diffusion fusion**
>
>    Most prior diffusion-based fusion methods apply fixed, step-invariant guidance strengths throughout the reverse process. This overlooks the spatio-temporal imbalance we reveal: different frequencies/regions are recovered at different rates, and at a given step the modalities contribute unequally to the current sample. This leads to two issues:
>
>    **Theory:** fixed weights are data-independent, yielding $\mathrm{Cov}(w_k, I_{k,t})=0$. By Eq.(4), the bound collapses to the constant $C$, preventing any tightening via positive correlation between weights and still-unfused information.
>
>    **Intuition:** a single static weighting cannot simultaneously match the needs of different reverse stages (earlier steps favor stable structure and salient targets, while later steps favor texture and fine details). As a result, static guidance may bias the fusion toward weaker modalities or uninformative regions at certain stages, causing artifacts such as over-exposure or detail loss. Our overall results and ablations show that dynamic weighting substantially mitigates such biases.
>
>
> **2. We prove that dynamic weighting is better**
>
>    Our Theorem 1 decomposes the generalization-error upper bound as
>
>    $$\mathrm{GError}(F)\le C-\sum_{t,k}A_{k,t},\mathrm{Cov}(w_k,I_{k,t})$$
>
>    This yields an explicit sufficient condition: if $w_k$ is positively correlated with the residual (yet-unfused) modality information $I_{k,t}$, the bound is lower. We operationalize $I_{k,t}$ with an observable proxy, the Diffusion Information Gain, $\mathrm{DIG}_k(t)=l(\hat c_t^k,c_k)$, and obtain $w_k(t)$ via $\mathrm{softmax}(\mathrm{DIG})$.
>
>    Here softmax is only a monotonic mapping; the key is that DIG is defined and estimated to reflect how much useful information modality $k$ can still supply at step $t$. Our covariance-controlled ablation (positive/zero/negative) matches Eq.(4) exactly, and on a GT-available multi-focus dataset we further show a strong correlation (Pearson $r=0.9345$) between DIG and the GT-based residual $I\_{k,t}$, quantitatively supporting the “dynamic over static” conclusion.
>
> **3. How we implement dynamic guidance (not “just softmax”)**
>
>    We provide a complete, reproducible dynamic mechanism rather than swapping a constant with softmax:
>
>    **Measuring residual information:** compute $\mathrm{DIG}_k(t)$ per modality, per region, per step from single-modal one-step reconstructions, robust across $\ell_2/\ell_1/\mathrm{SSIM}$.
>
>    **Theoretically grounded modality composition:** we adopt a solid, theory-backed weighted compositional guidance formulation that is fully compatible with diffusion sampling; see Appendix.
>
>    **Spatio-temporal dynamicity:** weights adapt across timesteps and across pixels/regions, yielding a highly dynamic guidance strategy in both time and space.
>
>    **Efficiency scheduling:** we use step sizing and a DIG refresh interval $S$ to reduce computation, achieving training-free gains over strong baselines using the same unconditional checkpoint.
>
> In summary, our contribution is not a new sampler, but a theoretically guaranteed dynamic guidance paradigm—**making weights positively correlated with residual information**—and a practical way to compute it via DIG. Under identical backbone and sampler, the improvements come from this dynamic principle rather than changing the diffusion core, which also makes the method easy to reuse and verify across VIF, MFF, and MEF.

---

> ### Author Response · Authors · 2025-11-23
> **Authors' Rebuttal to Reviewer aQV2 (3/3)**
>
> >**W3 Experiments on K>2 modalities.**
>
> **Response W3.** We thank the reviewer for pointing out the need to demonstrate scalability beyond bimodal fusion. In the revision, we add experiments in Appendix D.2 on a multi-exposure benchmark with more than two modalities. As shown in Fig. 13, the qualitative results demonstrate that Dig2DIG can effectively fuse complementary information from multiple exposures, confirming that our DIG-guided dynamic weighting extends naturally to (K>2) modality fusion scenarios.
>
>
> > **W4 Spatio-temporal weighting and DIG-to-weight mapping ablations.**
>
> **Response W4.** We agree that incremental ablations are important, and we now provide them explicitly. First, our full method already performs **spatio-temporal dynamic weighting** (region-wise *and* time-wise). We compare against spatial-only, temporal-only, and no-weighting variants on M3FD dataset:
>
> | Metric                         |     PSNR↑ |    SSIM↑ |     MSE↓ |     Nabf↓ |       CC↑ |    LPIPS↓ |
> | ------------------------------ | --------: | -------: | -------: | --------: | --------: | --------: |
> | Region + Time weighting (ours) | **31.83** | **1.41** | **2216** | **0.009** | **0.573** | **0.287** |
> | Region-only weighting          |     31.40 |     1.40 |     2269 |     0.012 |     0.570 |     0.290 |
> | Time-only weighting            |     30.67 |     1.37 |     2350 |     0.031 |     0.542 |     0.310 |
> | No weighting                   |     30.29 |     1.36 |     2381 |     0.040 |     0.535 |     0.322 |
>
> These results show that making the guidance highly dynamic in both space and time yields the best performance, while removing either dimension degrades quality.
>
> Second, we explore alternative monotonic mappings from DIG to weights, beyond softmax:
>
> | Metric            |     PSNR↑ |    SSIM↑ |     MSE↓ |     Nabf↓ |       CC↑ |    LPIPS↓ |
> | ----------------- | --------: | -------: | -------: | --------: | --------: | --------: |
> | Softmax (default) | **31.83** | **1.41** | **2216** | **0.009** | **0.573** | **0.287** |
> | Sigmoid gating    |     31.62 |     1.40 |     2239 |     0.022 |     0.556 |     0.292 |
> | ReLU mapping      |     31.67 |     1.41 |     2240 |     0.017 |     0.571 |     0.299 |
> | No weighting      |     30.29 |     1.36 |     2381 |     0.040 |     0.535 |     0.322 |
>
> We observe that different monotonic activations remain consistently better than no-weighting, indicating that the gain comes from DIG-based dynamic guidance itself rather than a particular normalization choice, while softmax performs best due to stable cross-modality relative weighting.
>
> These ablations further highlight the robustness and generality of our method: across different spatio-temporal weighting variants and multiple monotonic DIG-to-weight mappings, DIG-guided dynamic guidance consistently improves performance over no-weighting. Importantly, this aligns with our theory, which only requires a positive correlation between DIG and the fusion weights, and the observed gains provide effective empirical support for the correctness of our theoretical claims.

---

> ### Author Response · Authors · 2025-11-27
> **Summary of responses to Reviewer aQV2**
>
> We sincerely thank Reviewer aQV2 for the thoughtful comments and suggestions. In our rebuttal and revision, we:
>
> • Add MMDRFuse and LFDT-Fusion as strong recent baselines on all three VIF datasets (LLVIP, M3FD, MSRS) under identical training and evaluation protocols, and show that Dig2DIG consistently achieves better PSNR/SSIM/LPIPS and other metrics, strengthening the empirical comparison (W1).
>
> • Clarify the system-level novelty of Dig2DIG: we deliberately keep the DDPM/DDIM sampler standard and focus the contribution on a theoretically justified dynamic guidance rule that makes fusion weights positively correlated with residual modality information. We explain how static, step-invariant weights collapse our bound, how DIG operationalizes the residual term, and support this with covariance-controlled and GT-based correlation analyses, emphasizing that our method is more than a simple “softmax tweak” over fixed guidance (W2).
>
> • Provide new experiments for K>2 modalities on a multi-exposure benchmark with more than two exposures, showing that DIG-guided dynamic weighting naturally extends to multi-exposure fusion and produces visually compelling results (W3).
>
> • Add explicit ablations on the spatio-temporal structure of weighting (region+time vs region-only vs time-only vs no weighting) and on different monotone mappings from DIG to weights (softmax, sigmoid, ReLU). These ablations show that: (i) full spatio-temporal dynamics works best, and (ii) different monotone mappings remain clearly better than no weighting, with softmax performing best, confirming that the gains stem from DIG-based dynamic guidance itself rather than a particular activation choice (W4).
>
> These additions aim to better situate Dig2DIG among recent works, clarify where the system-level novelty lies, and empirically substantiate the scalability and robustness of our dynamic fusion framework.

---

### Official Review · Reviewer_beN9 · 2025-11-01

**Soundness:** 2
**Presentation:** 2
**Contribution:** 2
**Rating:** 4
**Confidence:** 5

**Summary:**

This paper introduces a novel dynamic image fusion framework for diffusion models, addressing the limitations of static fusion strategies. The authors first identify a "spatio-temporal imbalance" in the denoising process, where information from source modalities emerges at unequal rates across different steps and regions. To leverage this, they propose a metric called Diffusion Information Gains (DIG) to quantify the per-modality information contribution at each denoising step. These DIG values are then used as dynamic weights to guide the fusion, ensuring that only the most informative regions from each source are integrated at the appropriate time. The authors provide a theoretical proof that this dynamic weighting scheme provably reduces the upper bound of the generalization error, and experimental results confirm that the method achieves superior fusion quality and inference efficiency compared to existing diffusion-based approaches.

**Strengths:**

The paper is based on an interesting observation of spatio-temporal imbalance in the denoising process. The proposed dynamic weighting scheme is an intuitive response to this identified issue.

**Weaknesses:**

1. The qualitative comparisons appear incomplete and inconsistent with the quantitative evaluation. While numerous methods are benchmarked in the quantitative tables, the qualitative results in the figures only feature a select subset. For instance, in the visible-infrared fusion task (Figure 3), several strong baselines such as SwinFusion, DIVFusion, MoE-Fusion, CDDFuse, and DDFM are notably absent from the visual comparison. This omission is also observed for the MFFW and MEFB datasets. The authors should provide a rationale for this selective comparison or include qualitative results for all methods to ensure a fair and comprehensive evaluation.
1. Addtionally, DCEvo, as a SOTA fusion method in IV fusion, is also recommended for comparative analysis in experiments.
[1] DCEvo: Discriminative Cross-Dimensional Evolutionary Learning for Infrared and Visible Image Fusion. CVPR 2025.
2. The definition and application of the Generalization Error (GError) in Equation 3 are confusing in the context of image fusion. Generalization error is a concept fundamentally rooted in supervised learning, where a ground truth is available for evaluation. However, image fusion is an inherently unsupervised task that lacks a single, well-defined ground truth. Therefore, the direct application of this GError formulation seems unjustified. The authors must provide more substantial evidence or a more rigorous justification to demonstrate why this generalization bound is applicable and meaningful for an unsupervised task like image fusion.
3. The paper's central claim that the weights w_k derived from DIG guarantee a smaller generalization error bound—is not sufficiently proven. The theoretical connection between the proposed Diffusion Information Gain (DIG) and the reduction of the generalization error is tenuous. The manuscript lacks a formal proof demonstrating how optimizing for DIG directly leads to the tightening of this bound. Without this crucial link, the motivation for introducing DIG appears ad-hoc, and its relationship to the theoretical framework is not well-founded.
4. The visualization of the information gains in Figure 2 is perplexing and counterintuitive. The 'vi information gain' and 'ir information gain' maps appear to be almost perfectly complementary. The expectation is that the information gain for a modality should highlight unique information present in that modality. However, the 'vi information gain' map seems to highlight information behind the smoke, which is not visible in the original visible image and is instead a key feature of the infrared image. This is a significant contradiction. The authors must clarify the calculation and meaning of these gain maps and provide additional visual examples to resolve this apparent inconsistency.

**Questions:**

Please see Weaknesses

---

> ### Author Response · Authors · 2025-11-23
> **Authors' Rebuttal to Reviewer beN9 (1/4)**
>
> We thank the reviewer for the careful and constructive evaluation of our submission. We greatly appreciate your positive assessment, particularly your recognition of **(i) the interesting observation of spatio-temporal imbalance along the denoising trajectory** and **(ii) the proposed dynamic DIG-based weighting scheme as an intuitive response to this issue**. At the same time, we are grateful for the detailed concerns you raised, which help us further improve the fairness of our comparisons, the clarity of the generalization analysis, and the interpretation of DIG visualizations. Below, we respond to your comments point by point and summarize the corresponding clarifications and revisions.
>
> > **W1  Completing qualitative comparisons**
>
> **Response W1.** We thank the reviewer for the careful observation and apologize for the incomplete qualitative comparisons in the main paper. Due to space and layout constraints, the current draft visualizes only a subset of representative recent strong baselines. In the revision, we provide the **full qualitative results for all evaluated methods** in Appendix D (Figs. 8–12), together with detailed cross-dataset visual analysis, to ensure a fair and comprehensive evaluation.
>
> As shown in Fig. 8 on the M3FD dataset, our method is the only one that simultaneously preserves sky textures, produces clear building structures, and retains fine infrared details in smoky regions. In contrast, CCF and related methods blur the buildings, while DCEvo, LFDT-Fusion, and MMDRFuse lose infrared details inside the smoke; CDDFuse and SwinFusion fail to maintain the texture in the sky.
>
> As illustrated in Fig. 9 on the LLVIP dataset, our method is the only one that keeps both vehicle details and pedestrian details intact. MUFusion, LFDT-Fusion, and TC-MoA over-emphasize the infrared modality, which suppresses facial details of pedestrians, whereas DCEvo, CCF, and DDFM lose critical vehicle structures.
>
> As shown in Fig. 10 On the MARS dataset, our method preserves both plant textures and human facial details. Most competing approaches—such as DCEvo, LFDT-Fusion, MMDRFuse, DDFM, DIVFusion, and CCF—are overly influenced by the infrared modality and therefore lose facial details. Compared with MoEFusion, our method also produces plant colors that are closer to those in the original visible modality.
>
> As illustrated in Fig. 11 On the MFFW multi-focus dataset, our method maintains sharp textures and faithful colors, while CCF exhibits noticeable color distortion.
>
> As shown in in Fig. 12 On the MEFB multi-exposure dataset, our method achieves the most natural illumination transitions. Methods including CCF, DDFM, TTD, FusionDN, and U2Fusion show unnatural lamp-light transitions near the desk lamp region; compared with DeFusion and TC-MoA, our results contain fewer noise artifacts on the sofa. Overall, these visual comparisons further demonstrate the effectiveness and robustness of our approach across diverse fusion tasks.}
>
> > **W2  Adding DCEvo as a baseline**
>
> **Response W2.** We thank the reviewer for the valuable suggestion. In the revised version, we have added **DCEvo (CVPR 2025)** as an additional SOTA baseline under the same experimental settings, and report its quantitative and qualitative performance for a more comprehensive evaluation. DCEvo indeed shows competitive results; meanwhile, our Dig2DIG achieves consistent improvements across all three datasets, further supporting the effectiveness of our approach. The updated numbers are summarized below:
>
> **LLVIP dataset**
>
> | Method         | PSNR↑ | SSIM↑ | MSE↓ | Nabf↓ | CC↑  | LPIPS↓ |
> | -------------- | ----- | ----- | ---- | ----- | ---- | ------ |
> | DCEvo          | 32.42 | 1.15  | 2575 | 0.014 | 0.66 | 0.321  |
> | Dig2DIG (Ours) | 33.74 | 1.23  | 1464 | 0.001 | 0.73 | 0.298  |
>
> **M3FD dataset**
>
> | Method         | PSNR↑ | SSIM↑ | MSE↓ | Nabf↓ | CC↑  | LPIPS↓ |
> | -------------- | ----- | ----- | ---- | ----- | ---- | ------ |
> | DCEvo          | 31.45 | 1.40  | 3812 | 0.071 | 0.50 | 0.290  |
> | Dig2DIG (Ours) | 31.83 | 1.41  | 2216 | 0.009 | 0.57 | 0.287  |
>
> **MSRS dataset**
>
> | Method         | PSNR↑ | SSIM↑ | MSE↓ | Nabf↓ | CC↑  | LPIPS↓ |
> | -------------- | ----- | ----- | ---- | ----- | ---- | ------ |
> | DCEvo          | 38.00 | 1.41  | 2464 | 0.039 | 0.60 | 0.299  |
> | Dig2DIG (Ours) | 39.07 | 1.42  | 1366 | 0.001 | 0.63 | 0.282  |

---

> ### Author Response · Authors · 2025-11-23
> **Authors' Rebuttal to Reviewer beN9 (2/4)**
>
> > **W3 Clarifying GError under predefined fusion objectives**
>
> **Response W3.** We thank the reviewer for questioning the definition and applicability of GError in Eq.(3). We agree that image fusion typically lacks a single, objective ground-truth fused image at the data level. However, it is important to clarify that **fusion is not a goal-free unsupervised task**. Across the fusion literature, methods are formulated to approximate an “ideal fused image”  under a **predefined fusion objective**, which specifies that effective/salient information from multiple modalities should be preserved and complementarily integrated. This objective is then instantiated through fixed fusion rules, weights, or pseudo targets.
>
> This objective-driven paradigm is shared by both classical and learning-based approaches. For example, pyramid fusion determines weights via saliency and consistency measures to retain salient structures from the sources [r1]. Wavelet and multiresolution fusion uses max-activity / max-coefficient selection or weighted averaging rules to preserve the most salient detail coefficients, which is equivalent to optimizing a predefined information-preservation objective [r2]. Sparse-representation and multi-scale transform frameworks further write “preserve and complement effective features” as explicit energy/representation objectives and perform fusion inference under them [r3]. Likewise, learning-based fusion relies on targets: some tasks construct pseudo GT for supervision (e.g., IFCNN trains with constructed all-in-focus targets in multi-focus and multi-modality settings) [r4], while many others adopt self-supervised losses (intensity, gradient, structure, information measures) that serve as explicit optimization objectives, still centered on maximizing effective information retention [r5]. Existing surveys explicitly summarize that fusion methods either use constructed GT or predefined information-preservation losses as objectives [r6].
>
> Under this widely adopted “predefined fusion objective” formulation, the GError in Eq.(3) **does not rely on pixel-wise ground-truth fused images**. It captures how the algorithm’s ability to meet the fusion objective on the training distribution differs from its ability to meet the same objective on new scenes or under distribution shifts. The objective here is **to fuse as much effective multi-modal information as possible**, i.e., to maximally preserve  integrate informative content from all modalities, rather than following a method-specific fixed weighting rule, pseudo target, or narrow loss. Therefore, analyzing generalization under this objective is meaningful: compared with fixed weights or fixed, method-tied targets that introduce strong bias, an objective centered on comprehensive multi-modal information retention is inherently more robust and more likely to generalize across scenes and distributions. Our theory builds on this view to explain how DIG’s dynamic weighting reduces over-reliance on any single fixed objective and improves cross-scene generalization.
>
> [r1] Peter J. Burt, Robert J. Kolczynski. Enhanced Image Capture Through Fusion. In Proceedings of ICCV, 1993.
>
> [r2] Hui Li, Bharat S. Manjunath, Sanjit K. Mitra. Multisensor Image Fusion Using the Wavelet Transform. Graphical Models and Image Processing (GMIP), 57(3):235–245, 1995.
>
> [r3] Yu Liu, Shuping Liu, Zengfu Wang. A General Framework for Image Fusion Based on Multi-scale Transform and Sparse Representation. Information Fusion, 24:147–164, 2015.
>
> [r4] Yu Zhang, Yu Liu, Peng Sun, Han Yan, Xiaolin Zhao, Li Zhang. IFCNN: A General Image Fusion Framework Based on Convolutional Neural Network. Information Fusion, 54:99–118, 2020.
>
> [r5] Jiayi Ma, Yong Ma, Chang Li. Infrared and Visible Image Fusion Methods and Applications: A Survey. Information Fusion, 45:153–178, 2019.
>
> [r6] Hao Zhang, Han Xu, Xin Tian, Junjun Jiang, Jiayi Ma. Image Fusion Meets Deep Learning: A Survey and Perspective. Information Fusion, 76:323–336, 2021.

---

> ### Author Response · Authors · 2025-11-23
> **Authors' Rebuttal to Reviewer beN9 (3/4)**
>
> > **W4 Theoretical link between DIG and tightening the GError bound**
>
> **Response W4.** Thank you for raising this concern. We would like to clarify that our claim is **not** that DIG-based weights always make the *true* generalization error strictly smaller for every sample, but that **under our diffusion-guided fusion operator, dynamic weighting can provably tighten the upper bound of GError**, and DIG provides an observable proxy that makes this tightening operational rather than ad-hoc.
>
> **(1) Static vs. dynamic weighting follows directly from the bound structure.**
> From Theorem 1 ,
> $$
> \text{GError}(F)\le C-\sum_{t=1}^{T}\sum_{k=1}^{K}A_{k,t},\mathrm{Cov}(w_k, I_{k,t}),
> $$
> where $C$ and $A_{k,t}>0$ are constants independent of ${w_k}$.
> Hence, **the only weight-dependent part of the bound is the covariance term**.
>
> * A static scheme (e.g., $w_k=1/K$) is data-independent at every step, yielding $\mathrm{Cov}(w_k,I_{k,t})=0$, so the bound reduces to $C$.
> * Any dynamic scheme that makes $w_k$ positively correlated with the *unfused residual information* $I_{k,t}$ makes the covariance positive and therefore **strictly decreases the bound**.
>
> Thus the sufficient condition for tightening the bound is algebraically explicit:
> $$
> \mathrm{Cov}(w_k, I_{k,t})>0 \Rightarrow \text{a smaller GError upper bound}.
> $$
> **(2) Why the DIG form enforces this condition.**
> The latent residual term $I_{k,t}$ is unobservable. We define the single-modality one-step reconstruction discrepancy
> $$
> \mathrm{DIG}_k(t)=l(\hat c_k^t, c_k),
> $$
>
> which measures how far the current single-modal reconstruction is from its clean input at noise level $t$. A larger DIG indicates that modality $k$ still contains more information not yet absorbed by the current fused sample $x_t$. Based on the diffusion spatio-temporal imbalance and the known frequency-wise restoration behavior, $\mathrm{DIG}\_k(t)$ is a monotone, observable proxy of $I\_{k,t}$.We then set
> $$
> w_k(t)=\mathrm{softmax}(\mathrm{DIG}_k(t)).
> $$
>
> Softmax is strictly increasing with respect to $\mathrm{DIG}\_k(t)$: the modality with larger residual receives a larger weight. If $\mathrm{DIG}\_k(t)$ is positively correlated with $I\_{k,t}$ (as supported by our theoretical/spectral arguments), then
>
> $$
> \mathrm{Cov}(w_k, I_{k,t}) \propto \mathrm{Cov}(w_k,\mathrm{DIG}_k(t))>0,
> $$
>
> Thereby tightening the bound. In this sense, DIG is not an extra heuristic objective; it is a computable instantiation of the covariance-maximizing principle implied by the theorem.
>
> **(3) Empirical evidence matches the theoretical prediction.**
> In Appendix D.3 and Sec. 4.5 we perform covariance-controlled ablations: positive covariance (DIG softmax or $I\_{k,t}$ softmax), zero covariance (uniform weights), and negative covariance (inverse-DIG softmax or inverse-$I\_{k,t}$ softmax). We observe a consistent monotone performance ordering across datasets:
> $$
> \mathrm{Cov}(w_k,I_{k,t})>0      >\mathrm{Cov}(w_k,I_{k,t})=0         >       \mathrm{Cov}(w_k,I_{k,t})<0
> \quad\Rightarrow\quad
> \text{best} >\text{middle}>\text{worst},
> $$
>
> Overall, the formal chain is: **bound decomposition → need positive covariance → DIG as observable residual proxy → softmax DIG achieves positive covariance → bound is tightened**. This link is fully derived in the main proof and corroborated by our covariance ablation, so DIG is theoretically grounded rather than ad-hoc.

---

> ### Author Response · Authors · 2025-11-23
> **Authors' Rebuttal to Reviewer beN9 (4/4)**
>
> >**W5 Correcting Fig. 5 as weight (softmax-DIG) visualization**
>
> **Response W5.** Thank you for pointing this out. After revisiting our draft, we realized that the issue is not a reader misunderstanding but an imprecise description on our side. In Fig. 5 we actually visualize $\mathrm{softmax}(\mathrm{DIG}_k(t))$, i.e., the dynamic fusion weights $w_k(t)$, rather than the raw $\mathrm{DIG}_k(t)$ values themselves. Our intention was to highlight how the modality weights dynamically shift across regions and timesteps during denoising, indicating which modality should be emphasized at each step and location.
>
> We have corrected this wording in the main text by explicitly describing Fig. 5 as a weight visualization. Moreover, to address the reviewer’s expectation about “absolute” modality information, we now provide the raw (pre-softmax) DIG maps in Fig. 16. These DIG visualizations match the expected behavior: for the smoky regions, the VI-modality DIG does not reveal the human silhouette, while the IR-modality DIG retains the relevant structure. This supplementary figure clarifies the distinction between (i) raw $\mathrm{DIG}_k(t)$ as an absolute per-modality reconstruction discrepancy and (ii) $\mathrm{softmax}(\mathrm{DIG}_k(t))$ as the relative, complementary weights used for dynamic fusion.

---

> ### Author Response · Authors · 2025-11-27
> **Summary of responses to Reviewer beN9**
>
> We sincerely thank Reviewer beN9 for the constructive and detailed comments. In our rebuttal and revision, we:
>
> • Complete the qualitative comparisons by adding full visual results for all evaluated baselines (including SwinFusion, DIVFusion, MoE-Fusion, CDDFuse, DDFM, etc.) across VIF, MFFW, and MEFB in Appendix D (Figs. 8–12), and provide cross-dataset visual analyses showing where Dig2DIG better preserves structures, textures, and illumination **(W1)**.
>
> • Add DCEvo as an additional SOTA baseline under the same experimental settings, and report its quantitative and qualitative performance on LLVIP, M3FD, and MSRS. While DCEvo is competitive, Dig2DIG consistently achieves better PSNR/SSIM/LPIPS and related metrics, further supporting the effectiveness of our approach **(W2)**.
>
> • Clarify the role of the generalization error (GError) in Eq. (3) for image fusion: rather than relying on a pixel-wise ground-truth fused image, the bound is defined with respect to a predefined fusion objective (maximizing effective multi-modal information preservation), which is standard in both classical and deep fusion literature. We connect this to existing rule-based and self-/pseudo-supervised objectives and explain why studying generalization under this objective is meaningful for unsupervised fusion **(W3)**.
>
> • Strengthen the theoretical link between DIG and the tightening of the GError bound by explicitly showing how Theorem 1 decomposes the bound, why static (data-independent) weights collapse the covariance term, and how DIG—defined as a one-step reconstruction discrepancy and mapped through a monotone softmax—serves as an observable residual proxy that induces positive covariance and thus a smaller upper bound. Covariance-controlled ablations empirically match this prediction **(W4)**.
>
> • Correct the interpretation of Fig. 5, clarifying that it visualizes the DIG-based weights (softmax(DIG)) rather than raw DIG values, and add separate raw DIG maps in Fig. 16. These raw DIG visualizations behave as expected (e.g., IR DIG, not VI DIG, highlights the human silhouette behind smoke), resolving the apparent contradiction and clarifying the distinction between absolute residual information and relative fusion weights **(W5)**.

---

### Official Review · Reviewer_u7Jm · 2025-11-02

**Soundness:** 3
**Presentation:** 2
**Contribution:** 2
**Rating:** 4
**Confidence:** 5

**Summary:**

This paper proposes a novel dynamic image fusion framework called **Dig2DIG**, which leverages **Diffusion Information Gain (DIG)**. Traditional diffusion-based fusion methods employ fixed fusion weights, ignoring the varying rates of information contribution among different image regions and modalities during denoising.

Dig2DIG addresses this issue by quantifying each modality’s information contribution at every diffusion step and dynamically assigning fusion weights based on DIG. Theoretically, the authors claim that aligning fusion weights with residual modality information can reduce the generalization error of the fusion model.

Experiments on multiple datasets (e.g., LLVIP, M3FD, MSRS, MFFW, MEFB) demonstrate that Dig2DIG achieves state-of-the-art performance in both fusion quality and computational efficiency.

**Strengths:**

1. **Novel Theoretical Foundation** – The paper provides a formal analysis suggesting that dynamic weighting via DIG can theoretically improve generalization compared to static fusion. The proof is sufficiently detailed.

2. **No-Training Framework** – The method reuses pre-trained diffusion models without additional training or fine-tuning, enhancing generality and reproducibility across tasks.

3. **Efficiency Improvement** – The inference time is significantly reduced (up to 70%) by skipping low-information diffusion steps, while maintaining comparable accuracy, demonstrating potential for practical efficiency gains.

**Weaknesses:**

1. **Limited Applicability Beyond Diffusion Models** – The theoretical proof relies only on L-smoothness assumptions and does not strictly require a diffusion process. However, DIG itself is closely tied to diffusion-based sampling. Its generalization to non-diffusion or Transformer-based fusion frameworks remains unclear.

2. **Practical Interpretability** – Although theoretically well-founded, the practical estimation of “information gain” is not intuitively interpretable for non-expert users, and the DIG computation may be sensitive to design choices such as the metric function.

3. **Insufficient Experimental Validation** – Several theoretical or qualitative claims—such as the correlation between residual information and reduced generalization error, or the spatio-temporal imbalance illustrated in Figure 1—lack quantitative or ablation experiments for confirmation. Many statements remain conceptual rather than empirically supported.

4. **Heuristic Nature of Step-Skipping** – The claim that “steps with low information gain can be skipped” is based on empirical observation rather than theoretical proof. While steps with small DIG values contribute minimally and can be skipped with negligible quality loss, there is no formal guarantee that diffusion consistency is preserved. Thus, the approach should be viewed as a heuristic efficiency optimization, not a theoretically grounded one.

5. **Unclear Definition of the Residual Information Term $I_{k,t}$** – This term plays a central role in the theoretical derivation (Eq. 4), yet its formulation is only symbolic. The function $\Delta I(c_k, x_t, x^*(c))$ is introduced without specifying its exact mathematical form, dimensionality, or relationship to observable quantities.

6. **Unexplained Link Between DIG and $I_{k,t}$** – The paper states that since $I_{k,t}$ is unobservable, it is replaced by the empirical DIG metric in Section 3.3. However, the connection between this substitution and later equations (e.g., Eq. 37 in the appendix) is not clearly explained.

7. **Inconsistent Probabilistic Assumptions** – The derivation first assumes i.i.d. (independent) modalities for linear weighting of $x_t$, but later uses the DDFM framework, which is based on a joint hierarchical Bayesian model under EM optimization (keeps the joint distribution). These two assumptions contradict each other, and the latter is theoretically more appropriate. This inconsistency undermines the claimed theoretical rigor.

8. **Problematic Weighted Approximation (Eq. 16)** – The authors initially assume conditional independence (Eq. 15), then relax this by introducing arbitrary weights $w_k$ without redefining the joint distribution $p(c_1, \dots, c_K | x_t)$. This heuristic substitution lacks probabilistic justification. When modalities are correlated—which is usually true since all source images come from the same scene—the gradient of the joint log-likelihood should contain cross terms that cannot be represented by a simple weighted sum of marginal gradients. As a result, Eq. (16) violates Bayes’ theorem consistency and weakens the “provable” nature of subsequent derivations.

9. **Redundant Description in Section C.1** – The paper states, “We follow the EM algorithm of DDFM,” but the process described is nearly identical to DDFM’s. The section should either explicitly detail differences in E- and M-step formulations compared to DDFMs' or be removed for conciseness.

10. **Inconsistent Equation Formatting** – Equation references are inconsistently labeled (“eq.”, “Eq.”, “equation”, “eq. equation”), which detracts from clarity and professionalism.

11. **Problematic geometric analysis** – The analysis is not mathematically well-grounded starting fron  ( v = a,u_{IR} + b,u_{RGB} + c,u_S ) . The paper does not specify the underlying metric space, ignores the stochastic dependency of diffusion variables, and conflates statistical covariance with geometric cosine similarity. As a result, the claimed geometric intuition may be conceptually appealing but lacks rigorous justification or empirical verification.

**Questions:**

1. Can the proposed DIG mechanism be extended to **non-diffusion generative models**, such as Transformer- or GAN-based image fusion systems?
2. Regarding **step-skipping (weakness 4)**: does “skipping low-information steps” correspond to using $S = 10$ or the DIG-25 variant for inference? What is the precise relationship between these two settings?
3. Could the authors provide additional **ablation or sensitivity analyses** to confirm how DIG values correlate with residual information and generalization error reduction?
4. How does the choice of metric $l(\cdot, \cdot)$ in Eq. (5) affect the stability of DIG computation?
5. How could the DIG framework be modified to capture **nonlinear inter-modality interactions** beyond simple time-dependent linear weighting?
6. What space or norm is this ''angle'' defined in, and how is the stochastic nature of the diffusion variables handled?

---

> ### Author Response · Authors · 2025-11-23
> **Authors' Rebuttal to Reviewer u7Jm (1/10)**
>
> We thank Reviewer u7Jm for the careful and detailed review of our submission. We are grateful for your positive assessment of our work, in particular your recognition of **the theoretical contribution** that links dynamic weighting via DIG to generalization behavior, as well as the **training-free** nature of our framework that reuses a single pre-trained diffusion model across multiple fusion tasks. At the same time, we appreciate the thoughtful critical feedback and the important concerns you raise, which help us identify several aspects where the exposition and experiments can be further clarified and strengthened. Below, we respond to your comments one by one and describe the corresponding clarifications and revisions in the paper.
>
> > **W1 Clarification on applicability beyond diffusion models**
>
> **Response W1.** The reviewer points out that, in the current presentation, our generalization bound is stated under a generic $L$-smoothness assumption and therefore seems not to explicitly depend on the diffusion process, whereas DIG as a method is tightly coupled to diffusion-based sampling. We agree that, in the current proof, we have not articulated clearly how the theorem connects to the specific reverse-diffusion dynamics, which makes the presentation less intuitive. We thank the reviewer for this helpful reminder.
>
> In the revision, we clarify that the theorem is in fact proved along the specific reverse-diffusion trajectory induced by our fusion sampler, rather than for an arbitrary sequence of $L$-smooth updates. Concretely, we  have added a short paragraph at the beginning of Appendix B to (i) define the image sequence $\left\lbrace x_t \right\rbrace_{t=0}^T$
>  generated by the diffusion-based fusion sampler in Eq. (10), and (ii) state explicitly that the $L$-smoothness inequality is applied only to each consecutive pair $(x_t, x_{t-1})$ on this trajectory. Immediately afterwards, we substitute the concrete reverse-diffusion update to decompose $x_{t-1} - x_t$ into the unconditional score term, the multimodal guidance term, and the Gaussian noise term. This makes it explicit that the argument relies on the particular structure of the diffusion sampling procedure, and is not intended as a generic result for arbitrary $L$-smooth generative models.
>
> Regarding applicability beyond diffusion models, we will also make the scope of the current theory more explicit in the revised manuscript. Strictly speaking, the present theorem is established for diffusion models of the type used in our fusion sampler, that is, diffusion-based generative processes that produce a sequence of intermediate images during sampling. We do not claim that the same bound automatically extends to non-diffusion or Transformer-based fusion architectures. Extending DIG-style theoretical guarantees to such more general generative models would require a different analysis, and we will add a short discussion in the main text of the revision to state this limitation and to highlight it as an interesting direction for future work.

---

> ### Author Response · Authors · 2025-11-23
> **Authors' Rebuttal to Reviewer u7Jm (2/10)**
>
> > **W2 Clarifying practical interpretation and metric choice for DIG**
>
> **Response W2.** The reviewer notes that, although the notion of “information gain” in our framework is theoretically grounded, its practical meaning may not be immediately intuitive for non-expert readers, and that the DIG computation may be sensitive to design choices such as the metric function. We agree that these two aspects – how DIG should be understood in practice, and how strongly it depends on the choice of metric – should be stated more clearly, and we thank the reviewer for raising this point.
>
> In practical terms, DIG at step $t$ for modality $k$ is defined as $\mathrm{DIG}_k(t) = l(\hat{c}^t_k, c_k)$, that is, the discrepancy between the one-step reconstruction $\hat{c}^t_k$ of that modality and its own observed input $c^k$. Intuitively, a larger $\mathrm{DIG}_k(t)$ means that at step $t$ the current reconstruction of modality $k$ is still far from its input, so this modality still carries more residual information that has not yet been injected into the fused image. Conversely, a smaller $\mathrm{DIG}_k(t)$ means that the modality is already well reconstructed and is locally less informative at that step. From the theoretical perspective, Theorem 1 only requires $\mathrm{DIG}_k(t)$ to be a monotone proxy of the residual information term $I\_{k,t}$: modalities with larger residual information should receive larger weights. In particular, the bound does not rely on a specific analytic form of $l(\cdot,\cdot)$, but only on the assumption that $l(\hat{c}^t_k, c_k)$ increases when more residual information remains.
>
> The empirical results in Table 4 provide supporting evidence for this interpretation and offer an indication that the method is not overly sensitive to the choice of metric. On the M3FD dataset, we compare several choices of $l$: no metric (uniform weights), $\ell_1$, SSIM, and $\ell_2$. All three non-trivial metrics ($\ell_1$, SSIM, $\ell_2$) yield clear improvements over the baseline with uniform weights, while the performance differences among these three metrics are relatively modest, with $\ell_2$ being slightly better overall. This suggests that our framework is not overly sensitive to the specific form of $l$, as long as it provides a reasonable, monotone measure of reconstruction error.

---

> ### Author Response · Authors · 2025-11-23
> **Authors' Rebuttal to Reviewer u7Jm (3/10)**
>
> > **W3 Need for Stronger Experimental Verification**
>
> **Response W3**. We thank the reviewer for noting the insufficient experimental validation, and we agree that several theoretical and qualitative observations should be supported by more direct quantitative evidence or ablations. In the revision, we will strengthen the paper in two ways.
>
> First, regarding the “spatio-temporal imbalance” in Fig. 1, we will clarify its positioning and add appropriate references. Many diffusion-model studies have analyzed the non-uniform information contribution across reverse-diffusion timesteps, reporting temporal imbalance along the sampling trajectory and its implications such as phase-wise redundancy and timestep importance differences [r1-3]. Our contribution is not to claim the first discovery of this phenomenon, but to summarize and formalize, in the multimodal fusion setting, its coupled spatial (region-wise modality dominance) and temporal (step-wise information injection non-uniformity) manifestations into a unified “spatio-temporal imbalance” perspective, which then motivates our DIG-based dynamic weighting scheme. We will revise the corresponding text and include these citations.
>
> Second, to directly validate the theory (see Section 4.5), we add new experiments on a ground-truth multi-focus fusion task, namely MFI-WHU [r4]. In our theory, $I\_{k,t}$ denotes the residual (yet-unfused) information of modality $k$ at step $t$, i.e., how much information from modality $k$ has not been incorporated into the current fused image $x_t$. Since $I_{k,t}$ is unobservable in typical fusion tasks, we use the empirical $\mathrm{DIG}_k(t)$ as its monotone proxy. On MFI-WHU, because all-in-focus GT is available, $I\_{k,t}$ can be directly measured using GT by computing region-wise $\ell_2$ distances between the fused image and GT, separately on foreground and background regions. This enables quantitative verification of two key implications:
>
> 1. **$\mathrm{DIG}_k(t)$ as a proxy of $I\_{k,t}$.** We compute per-modality $\mathrm{DIG}_k(t)$ and $I\_{k,t}$ along the sampling trajectory and measure their correlation. Fig. 6 shows a Pearson correlation of 0.9345, indicating that $\mathrm{DIG}_k(t)$ monotonically and reliably tracks $I\_{k,t}$.
>
> 2. **$I_{k,t}$ vs. error reduction.** We construct fusion weights using softmax($I_{k,t}$) and compare three settings: weights positively correlated with $I_{k,t}$, uncorrelated/fixed weights, and negatively correlated (inverse-$I_{k,t}$) weights. It shows that the positively correlated setting achieves the best fusion quality, while inverse-$I_{k,t}$ weighting degrades performance, supporting our prediction that aligning weights with residual information $I_{k,t}$ reduces error, whereas misalignment harms generalization.
>
>    [r1] Tiankai Hang, Shuyang Gu, Chen Li, Jianmin Bao, Dong Chen, Han Hu, Xin Geng, Baining Guo. Efficient Diffusion Training via Min-SNR Weighting Strategy. In *Proceedings of the IEEE/CVF International Conference on Computer Vision (ICCV)*, 2023.
>
>    [r2] Tianyi Zheng, Peng-Tao Jiang, Ben Wan, Hao Zhang, Jinwei Chen, Jia Wang, Bo Li. Beta-Tuned Timestep Diffusion Model. In *European Conference on Computer Vision (ECCV)*, 2024.
>
>    [r3] Myunsoo Kim, Donghyeon Ki, Seong-Woong Shim, Byung-Jun Lee. Adaptive Non-Uniform Timestep Sampling for Accelerating Diffusion Model Training. In *Proceedings of the IEEE/CVF Conference on Computer Vision and Pattern Recognition (CVPR)*, 2025.
>
>    [r4] Hao Zhang, Zhuliang Le, Zhenfeng Shao, Han Xu, Jiayi Ma. MFF-GAN: An Unsupervised Generative Adversarial Network with Adaptive and Gradient Joint Constraints for Multi-Focus Image Fusion. *Information Fusion*, 66:40–53, 2021.

---

> ### Author Response · Authors · 2025-11-23
> **Authors' Rebuttal to Reviewer u7Jm (4/10)**
>
> > **W4 Heuristic nature of step-skipping**
>
> **Response W4.** We thank the reviewer for this clarification. We agree that our statement that “steps with low information gain can be skipped” currently has no formal proof of diffusion consistency, and should therefore be regarded as an empirical, efficiency-oriented heuristic rather than a theoretically guaranteed property.
>
> At the same time, our theoretical result in Section 3 and Appendix B is stated for a diffusion-based fusion operator $F$ defined by a reverse trajectory  $\lbrace x_t \rbrace _{t=0}^T$ , and does not impose any specific choice of time grid or number of steps. In other words, once a particular sampler (with or without step-skipping) is fixed, Theorem 1 and the covariance bound describe how DIG-based weights should correlate with the residual modality information along that trajectory; the theorem itself does not require or forbid step-skipping.
>
> In practice, we instantiate $F$ with a reduced-step sampler and use the information gain as a heuristic signal to decide where the trajectory can be discretized more coarsely to save computation. Table 7 (originally Table 5) shows that moderate skipping (e.g., “DIG-25” vs. “DIG-50”) yields large speedups with very similar fusion quality, whereas overly aggressive skipping (“DIG-15”) clearly degrades performance. In the revised version, we will state explicitly that the step-skipping policy is a DIG-guided efficiency heuristic evaluated empirically, while the formal generalization analysis applies to any fixed diffusion-based sampler. Investigating whether information-gain–driven step allocation itself can be brought into the same generalization-error framework is left for future work.

---

> ### Author Response · Authors · 2025-11-23
> **Authors' Rebuttal to Reviewer u7Jm (5/10)**
>
> > **W5 & W6 Definition of residual information $I_{k,t}$ and its link to DIG**
>
> **Response W5 & W6.** We thank the reviewer for carefully pointing out the ambiguity in the definition of the residual information term $I_{k,t}$ and its connection to DIG. Our understanding of the concern is twofold. First, in Eq. (4) we introduce $I_{k,t} = \Delta_I(c_k, x_t, x^{*}(c))$ only in symbolic form, without specifying its exact mathematical form or how it would be instantiated if an ideal fused image were available. Second, Section 3.3 states that $I_{k,t}$ is unobservable and is replaced by the empirical DIG metric, but the link between this substitution and the later geometric analysis in the appendix (for example Eq. (37)) is not explained explicitly.
>
> Conceptually, in the main text $I\_{k,t}$ is meant to quantify how much information from modality $k$ has not yet been incorporated into the current fused image $x\_t$. Under the thought experiment where an ideal fused image $x^{\ast}(c)$ exists, this can be instantiated as the squared norm of the component of the residual  $x^{*}(c)-x\_t$ that lies in the subspace associated with modality $c_k$, that is,
>
> $$ I\_{k,t} \triangleq \bigl|\Pi_k\bigl(x^{*}(c) - x_t\bigr)\bigr|^2, $$
>
> where $\Pi_k$ denotes the projection operator onto the modality-$k$ subspace. This makes it clear that $I_{k,t}$ is a scalar, defined directly in terms of the ideal fusion and the current fusion. In the initial version we did not write out this explicit form in order to keep the main text concise, but we agree that omitting it makes $\Delta_I(c_k,x_t,x^{\ast}(c))$ harder to interpret.
>
> In the appendix, we instantiate exactly this idea in a simplified geometric model. We decompose the residual between the ideal fusion and the current fusion as
>
> $$ x^{*}(c) - x_t = au_{\mathrm{IR}} + bu_{\mathrm{RGB}} + cu_s, $$
>
> where $u_{\mathrm{IR}}$ and $u_{\mathrm{RGB}}$ span modality-specific high-frequency subspaces and $u_s$ captures shared low-frequency structure. In this setting, the residual information carried by each modality is represented by the coefficients $a$ and $b$, that is,
>
> $$ I_{\mathrm{IR},t} \propto |a|,\qquad I_{\mathrm{RGB},t} \propto |b|. $$
>
> The subsequent angle analysis studies the angles $\theta_{t,k}$ between the ideal-fusion direction and the modality-$k$ guidance direction, and shows that, outside a negligible corner region, the ordering of $I_{k,t}$ across modalities coincides with the ordering of $\cos\theta_{t,k}$. In other words, the appendix does not introduce $I_{k,t}$ after the fact, but realizes $I_{k,t}$ as the residual coefficients $a,b$ in this orthogonal decomposition and then establishes a monotone relationship between these residuals and the geometric alignment.
>
> We agree that our current presentation does not make this identification sufficiently explicit. In the revision, we will therefore (i) add in the main text a concrete example showing that, under the existence of an ideal fusion $x^{\ast}(c)$, $\Delta_I(c_k,x_t,x^{\ast}(c))$ can be realized as the projection energy $|\Pi_k(x^{\ast}(c) - x_t)|^2$; and (ii) rewrite the beginning of the appendix so that the coefficients $a$ and $b$ are explicitly denoted as $I_{\mathrm{IR},t}$ and $I_{\mathrm{RGB},t}$.
>
> Finally, we clarify the chain of connections between $I_{k,t}$ and DIG. In Section 3.2, the theoretical bound is derived in terms of the covariance between the fusion weights and a directional alignment term. This alignment term is proportional to $\cos\theta_{t,k}$, the cosine between the ideal-fusion direction and the modality-$k$ guidance direction. In the appendix, the geometric analysis then shows that, in the simplified model, the ordering of $\cos\theta_{t,k}$ across modalities is consistent with the ordering of the residual information $I_{k,t}$: modalities that still carry more unfused information have larger $\cos\theta_{t,k}$. Thus, $I_{k,t}$ is the theoretically appropriate residual term that appears in the covariance of Theorem 1.
>
> At the same time, $x^{\ast}(c)$ and the projections $\Pi_k(x^{\ast}(c) - x_t)$ are not observable in practice, so $I_{k,t}$ cannot be computed directly. Diffusion-model studies have repeatedly observed that different frequency bands and different regions are recovered at different speeds along the reverse trajectory, and that this recovery speed is dictated by the underlying information content. Motivated by this, in Section 3.3 we introduce $\mathrm{DIG}_k(t) = l(\hat c_k^{t}, c_k)$ as an observable quantity computed from single-modality reconstructions.In the revision we will make this chain explicit: (i) the bound is expressed in terms of $\cos\theta\_{t,k}$; (ii) the appendix establishes a monotone relationship between $\cos\theta\_{t,k}$ and $I\_{k,t}$; and (iii) $\mathrm{DIG}\_k(t)$ is used as a practical proxy that is designed to be monotone with $I\_{k,t}$.

---

> ### Author Response · Authors · 2025-11-23
> **Authors' Rebuttal to Reviewer u7Jm (6/10)**
>
> **W7 & W8 Probabilistic assumptions**
>
> **Response W7 & W8.** We sincerely thank the reviewer for the important theoretical feedback, which helped us realize that the probabilistic exposition around Eq. (15) and Eq. (16) should be made more explicit. In the revision, we will clarify two points so that the derivation is Bayes consistent.
>
> First, DDFM’s EM and IRLS are performed under a modality separable approximate probabilistic model. Multi modality fusion is cast as robust, weighted inference on a shared latent variable, namely the residual or the fused estimate, where each modality contributes one independent conditional likelihood term together with its own robustness scale or weight variable. This modality-separable likelihood modeling (i.e., a joint objective written as a sum/product of per-modality terms) directly leads to modality-wise computable updates in practice, and adopting such separable likelihoods is a common and standard strategy in Bayesian/diffusion-based fusion. Therefore, both the probabilistic formulation and the EM updates factorize over modalities. Modalities interact only implicitly through the shared latent variable, and no explicit cross modality interaction term, such as covariance or cross energy, is modeled. Hence our separable conditional score form in Eq. (16) is consistent with DDFM’s inference assumption.
>
> Second, following the reviewer’s suggestion, we will define an explicit weighted joint conditional distribution
> $$p_w(c \mid x_t) \propto \prod_{k=1}^K p(c_k \mid x_t)^{w_k}, \quad w_k \ge 0,\ \sum_{k=1}^K w_k = 1.$$
> Under this model, the conditional score becomes
> $$\nabla_{x_t} \log p_w(c \mid x_t) = \sum_{k=1}^K w_k \nabla_{x_t} \log p(c_k \mid x_t),$$
> which places Eq. (16) on a Bayes consistent approximate joint distribution. Intuitively, $w_k$ adjusts the relative confidence or strength of each modality likelihood, and such weighted score composition in multi condition guidance is widely used in diffusion based guidance and multi condition generation or fusion [r5-7].
>
> Regarding correlated modalities and potential cross terms, we agree that a full $\log p(c_1,\ldots,c_K \mid x_t)$ may contain cross-modality gradients. However, treating each condition/modality as a separable expert and composing guidance via a (weighted) sum of individual scores is the standard approximation in compositional / multi-guidance diffusion, which corresponds to a product-of-experts–style approximate joint model and yields tractable, stable conditional guidance [r8-9]. Diffusion-based image fusion methods follow the same separable-guidance-then-fusion paradigm, i.e., they inject modality-specific priors/guidance independently and combine them through explicit fusion mechanisms within the diffusion process, rather than explicitly modeling cross-modality score interactions [r10]. We will state this assumption more clearly in the paper, and in future work we will investigate the effect of explicitly adding cross-modality interaction energies or covariances on the right-hand side of Eq. (16), and how they refine the relationship between dynamic weights and residual information.
>
> [r5] Jonathan Ho and Tim Salimans. Classifier-Free Diffusion Guidance. arXiv preprint arXiv:2207.12598, 2022.
>
> [r6] Nan Liu, Shuang Li, Yilun Du, Antonio Torralba, and Joshua B. Tenenbaum. Compositional Visual Generation with Composable Diffusion Models. In ECCV, 2022.
>
> [r7] Huixuan Zhang, Junzhe Zhang, and Xiaojun Wan. How Much To Guide: Revisiting Adaptive Guidance in Classifier-Free Guidance Text-to-Vision Diffusion Models. arXiv preprint arXiv:2506.08351, 2025.
>
> [r8] Yilun Du, Conor Durkan, Robin Strudel, Joshua B. Tenenbaum, Sander Dieleman, Rob Fergus, Jascha Sohl-Dickstein, Arnaud Doucet, and Will S. Grathwohl. Reduce, Reuse, Recycle: Compositional Generation with Energy-Based Diffusion Models and MCMC. In ICML, 2023.
>
> [r9] Marta Skreta, Tara Akhound-Sadegh, Viktor Ohanesian, Roberto Bondesan, Alán Aspuru-Guzik, Arnaud Doucet, Rob Brekelmans, Alexander Tong, Kirill Neklyudov. Feynman–Kac Correctors in Diffusion: Annealing, Guidance, and Product of Experts. In ICML (Spotlight), 2025.
>
> [r10] Hao Zhang, Lei Cao, and Jiayi Ma. Text-DiFuse: An Interactive Multi-Modal Image Fusion Framework based on Text-modulated Diffusion Model. In NeurIPS, 2024.

---

> ### Author Response · Authors · 2025-11-23
> **Authors' Rebuttal to Reviewer u7Jm (7/10)**
>
> > **W9 Redundant  description**
>
> **Response W9**.  Thank you for pointing out the redundancy in Sec. C.1. Our original intention in including this EM description was to make the appendix self-contained, so that readers could understand the concrete update forms without needing to consult DDFM. We agree that this section largely repeats prior material and does not add new formulation details. In the revised version, we will remove Sec. C.1 for conciseness and refer readers directly to DDFM where appropriate.
>
> > **W10 Equation label**
>
> **Response W10**. We thank the reviewer for pointing out the inconsistency in equation references. In the revised version, we unify all equation citations to use **“Eq.”** for clarity and consistency.

---

> ### Author Response · Authors · 2025-11-23
> **Authors' Rebuttal to Reviewer u7Jm (8/10)**
>
> > **W11   Problematic geometric analysis**
>
> **Response W10.** We thank the reviewer for carefully scrutinizing the rigor of our geometric analysis. In the revision, we will make the underlying assumptions and the scope of the geometric argument explicit to remove potential ambiguity. Our clarifications are as follows.
>
> 1. **Explicit metric and inner-product space, and the context of the decomposition.**
>    Our geometric analysis is carried out in the discrete pixel domain. Concretely, each image is treated as a vector $x\in\mathbb{R}^{HWN}$, equipped with the standard Euclidean dot product $\langle x,y\rangle=\sum_{p}x_p y_p$ and the induced notion of length. All energy comparisons, orthogonal decompositions, and projections in the appendix are defined under this dot-product geometry.
>    Under this setting, $u_{\mathrm{IR}},u_{\mathrm{RGB}},u_s$ are interpreted as approximately orthogonal frequency-band directions: $u_{\mathrm{IR}}$ and $u_{\mathrm{RGB}}$ capture modality-private high-frequency details, while $u_s$ captures shared low-frequency structure. This private-plus-shared orthogonal decomposition viewpoint aligns with classical frequency-domain and multi-scale orthogonal decompositions, as well as subspace-projection principles widely used in image fusion[r11]. We will explicitly state this dot-product space and the approximate orthogonality assumption in the paper.
>
> 2. **Stochastic dependency and the scope of the geometric argument.**
>    Our main theorem and the generalization bound are derived under the actual diffusion sampling process, taking expectation $\mathbb{E}_{c,z}[\cdot]$; thus stochasticity is already rigorously accounted for in the bound. The geometric analysis does not introduce a separate stochastic model. Instead, it analyzes a particular reverse step $t$ along a given sampling trajectory to explain the local directional alignment term appearing in the bound. Therefore, we do not ignore stochastic dependencies of diffusion variables, nor do we assume cross-step independence. The geometric part is a step-wise local interpretation consistent with the statistical framework of the theorem. We will make this “per-step local analysis” context clearer in the revision.
>
> 3. **Clarifying covariance versus cosine alignment.**
>    We do not conflate statistical covariance with geometric cosine similarity. The theorem links the generalization-error upper bound to the statistical covariance $\mathrm{Cov}(w_k,\text{alignment}_{k,t})$ over samples from the data distribution. Meanwhile, the alignment term decomposes as
>
> $$\mathrm{alignment}\_{k,t} =\||\nabla_{x_t}\log_p(c\_k\mid x\_t)\||\cos\theta_{t,k}.$$
>
>
>
>    where $\cos\theta_{t,k}$ is a single-sample geometric alignment factor. Our appendix provides a further analysis of this cosine factor, proving that the residual modality information $I_{k,t}$ is monotonically proportional to $\cos\theta_{t,k}$ in the stated working regime. This explains why making $w_k$ positively correlated with $I_{k,t}$ (equivalently with $\cos\theta_{t,k}$) increases the covariance term and tightens the bound.
>
> We again thank the reviewer for the constructive feedback. These revisions will make the metric setting, assumptions, and the exact role of the geometric proof in our bound much clearer, emphasizing that our geometric argument is a rigorous analysis of the cosine term inside the covariance, rather than a purely intuitive explanation.
>
> [r11] Shutao Li, Xudong Kang, Leyuan Fang, Jianwen Hu, and Haitao Yin. *Pixel-level Image Fusion: A Survey of the State of the Art*. Information Fusion, 33:100–112, 2017.

---

> ### Author Response · Authors · 2025-11-23
> **Authors' Rebuttal to Reviewer u7Jm (9/10)**
>
> >**Q1 Applicability Beyond Diffusion Models**
>
> **Response Q1.** As explained in Response W1, our theorem and DIG weighting are derived along the specific reverse-diffusion trajectory generated by our sampler, i.e., the sequence $\\{x_t\\}_{t=0}^T$ from Eq. (2). Therefore, the current theoretical guarantee is scoped to diffusion-type generative fusion processes that produce intermediate states during sampling, and it is not claimed to directly cover Transformer- or GAN-based fusion without a new, model-specific analysis. We will make this scope and limitation explicit in the revised manuscript.
>
> >**Q2 Relationship Between Step-Skipping and $S$**
>
> **Response Q2.** “Step-skipping” corresponds to the DIG-$N$ variants (e.g., DIG-25, DIG-50), where $N$ is the total number of reverse sampling steps, i.e., the actual denoising updates executed; a smaller $N$ means more skipping and faster inference. $S$ is a separate hyperparameter that controls the DIG weight refresh interval: DIG is recomputed and $w_k(t)$ is updated only every $S$ reverse steps. It affects the frequency and cost of weight updates, but does not change the total sampling steps, so it is not the step-skipping mechanism.
> In short, DIG-$N$ sets how many steps are sampled for skipping, while $S$ sets how often weights are updated within those steps.
>
> >**Q3 Ablation for DIG–Residual Correlation and Error Reduction**
>
> **Response Q3.** This is addressed in Response W3 with new experiments on MFI-WHU. Because GT all-in-focus images are available there, we can directly measure residual information $I\_{k,t}$ and show that $\mathrm{DIG}\_k(t)$ tracks it monotonically along the trajectory (Pearson correlation $0.9345$ in Fig. 6). We further verify that weights positively correlated with $I\_{k,t}$ achieve the best fusion quality, while inverse-$I\_{k,t}$ weighting degrades performance, matching our theoretical prediction.

---

> ### Author Response · Authors · 2025-11-23
> **Authors' Rebuttal to Reviewer u7Jm (10/10)**
>
> >**Q4 Effect of Metric $L(\cdot,\cdot)$ on DIG Stability**
>
> **Response Q4.** As discussed in Response W2, the theory only requires $L(\hat c_k^t, c_k)$ to be a monotone proxy of residual information, not a specific analytic form. Empirically, Table 4 shows that several reasonable choices (e.g., $\ell_1$, SSIM, $\ell_2$) all yield clear gains over uniform weighting, with only modest differences among them. This indicates DIG is stable as long as the metric reflects reconstruction discrepancy.
>
> >**Q5 Nonlinear Inter-Modality Interactions**
>
> **Response Q5.** We thank the reviewer for the question. Although we use time-dependent linear weights to superpose modality-specific guidance when driving the diffusion sampler, the DIG framework itself is not purely linear: computing $\mathrm{DIG}_k(t)=L(\hat c_k^t,c_k)$ requires obtaining $\hat c_k^t$ through the diffusion model’s denoising / reverse-sampling process, whose dynamics are highly nonlinear, and $L(\cdot,\cdot)$ is typically nonlinear as well; hence each modality’s information gain evolves nonlinearly along the sampling trajectory. Moreover, a widely accepted baseline formulation in image fusion is to model the fused image as a linear weighted superposition (weighted-sum/linear combination) of multiple source modalities, and many pixel-level and classical fusion approaches are built upon this linear superposition consensus. Accordingly, linearity in our method is confined to the normalized superposition of guidance signals, while the effective nonlinear inter-modality coupling is introduced by the diffusion denoising dynamics and the nonlinear DIG measurement.
>
> >**Q6 Space/Norm for the Angle and Stochasticity Handling**
>
> **Response Q6.** This is clarified in Response W11. The angle $\theta_{t,k}$ is defined in the discrete pixel Euclidean inner-product space where each image is a vector in $\mathbb{R}^{HWN}$ with dot product $\langle x,y\rangle=\sum_p x_p y_p$, so cosine alignment is computed under this standard geometry. The stochastic nature of diffusion is handled in the main theorem via expectations over sampling randomness; the geometric analysis is a per-step local interpretation of the cosine factor inside the statistically defined covariance term, not a separate independence assumption.

---

> ### Author Response · Authors · 2025-11-27
> **Summary of responses to Reviewer u7Jm**
>
> We sincerely thank Reviewer u7Jm for the detailed and helpful review. In our rebuttal and revision, we:
>
> • Clarify the scope of the theory: the generalization bound is proved along the specific reverse-diffusion trajectory ${x_t}$ of our sampler, and we explicitly scope the current guarantee to diffusion-type fusion models (not generic Transformer / GAN fusion) (W1, Q1).
>
> • Clarify the practical meaning and robustness of DIG: we explain DIG as a one-step reconstruction discrepancy measuring yet-unfused modality information, show that several reasonable metrics l(·) (ℓ2 / SSIM / ℓ1) all outperform uniform weights, and add GT-based experiments on MFI-WHU showing strong DIG–residual correlation and that positively correlated weights give the best fusion quality (W2, W3, Q3, Q4).
>
> • Separate formal theory from heuristics on step-skipping: the covariance bound itself is agnostic to skipping; DIG-K denotes the total number of reverse steps (how much we skip), while τ only controls how often DIG/weights are refreshed. We now state clearly that DIG-guided step-skipping is an empirical efficiency heuristic evaluated via ablations (W4, Q2).
>
> • Make the residual term and probabilistic assumptions explicit: we instantiate $I_{k,t}$ as projection energy of the fusion residual onto modality-k subspaces and connect it to the cosine-alignment analysis, then define an explicit weighted joint conditional whose score is a weighted sum of modality scores, consistent with standard product-of-experts / multi-guidance diffusion (W5, W6, W7, W8, Q5, Q6).
>
> • Improve exposition and geometric analysis: we unify all equation references to “Eq.”, and explicitly state the Euclidean inner-product space and per-step nature of the geometric argument as an analysis of the cosine term inside the covariance, not a separate probabilistic model (W9, W10, W11).

---

### Author Response · Authors · 2025-12-03
**Author Final Remarks**

Dear SACs, ACs, and Reviewers,

We sincerely thank you for the time and care you devoted to reviewing our submission, and for the many detailed comments and suggestions that helped us clarify and strengthen Dig2DIG, a diffusion-based fusion framework where Diffusion Information Gain (DIG) drives dynamic modality weighting.

We are encouraged that the reviews highlighted both our theoretical analysis and practical advantages, including a **“Novel Theoretical Foundation”** and clear **“Efficiency Improvement”** (Reviewer **u7Jm**), the spatio-temporal imbalance perspective and DIG as “**an intuitive response**” to this issue (Reviewer **beN9**), the fact that the paper “**thoroughly analyzes**” the motivation for dynamic fusion and “**systematically compares**” with strong baselines (Reviewer **aQV2**), and the **meaningful** idea of correlating guidance weights with modality residual information estimated via information gain (Reviewer **UcWT**).

In response, we made three focused sets of improvements:

1. **Theoretical clarifications**. We clarify in the main text and appendix that the generalization-error bound is proved along the specific reverse-diffusion trajectory of our sampler and is developed within the framework of diffusion-type fusion models. We also provide more explicit definitions and explanations for the residual-information term and the corresponding geometric analysis. Following reviewers’ suggestions, we introduce an explicit weighted joint conditional distribution with weighted modality scores, which makes the theoretical treatment more complete.

2.  **DIG as a proxy for residual information**. We further clarify the intuition behind using DIG as a proxy for modality residual information, and briefly distinguish the illustrative “information gain” visualizations from the formal DIG quantity. To empirically support our theory, we add GT-based multi-focus experiments on a dataset with all-in-focus ground truth, showing that DIG is strongly correlated with a ground-truth residual term and that weights positively correlated with this residual reduce error, while inverse correlation degrades performance, in line with our covariance-based theory.

3.  **Additional experiments, baselines, and ablations**. To strengthen empirical support, we add several recent strong fusion methods as comparisons, demonstrate scalability beyond bimodal fusion in K>2 multi-exposure settings, and include ablations on DIG spatio-temporal weighting and weight-mapping functions, showing that fully spatio-temporal DIG-based dynamic weighting performs best and that the gains are robust to reasonable design choices.

In the revised manuscript, the corresponding new or refined content is highlighted in **blue** to make the changes easy to track. We again thank the SACs, ACs, and Reviewers, for their constructive feedback and careful evaluation, and we believe these clarifications, additional experiments, and strengthened presentation make Dig2DIG clearer, more rigorous, and more useful to the diffusion-based fusion community.

Best regards,
Authors of Submission 688

---

### Meta-Review · Area_Chair_ZcVh · 2026-01-07

**Summary:**

The authors propose Dig2DIG, a dynamic image fusion framework for diffusion models, introducing Diffusion Information Gains (DIG) to guide modality weighting throughout the denoising process. While the paper offers a novel theoretical perspective and adds several supporting experiments during the rebuttal, all reviewers raised substantial concerns about the clarity of theoretical formulations, the appropriateness of DIG as a proxy for residual information, and the rigor of generalization guarantees. The rebuttal addresses these points at length but does not convincingly resolve the mathematical ambiguities or bridge the theoretical-empirical gap that underpins the central claims.

**Reviewer Concerns:**

The authors made a commendable effort to respond in detail, including extended theoretical clarifications and additional ablations. However, several critical issues remain unresolved: (1) DIG’s reliance on loosely defined or heuristic approximations remains theoretically fragile; (2) the generalization error bound, while formally stated, is only weakly connected to the unsupervised nature of image fusion; and (3) many empirical gains are marginal, with improvements potentially attributable to architectural heuristics rather than a sound theoretical foundation.

**Reviewer Scores:**

Reviewer u7Jm raised deep theoretical concerns about the probabilistic assumptions, DIG’s proxy validity, and inconsistencies in the derivation. Despite a detailed rebuttal, the core issues regarding Bayes consistency and residual term definition were only partially addressed. This reviewer does not give feedback. Score likely unchanged (4).

Reviewer beN9 appreciated the intuition behind DIG but found the generalization bound questionable and visualizations misleading. The rebuttal adds details, but the foundational disconnect between theory and fusion task remains. This reviewer does not give feedback. Score likely unchanged (4).

Reviewer aQV2 acknowledged strong experiments but noted the insufficient scaling to complex scenarios. Despite more baselines and ablations, the central innovation remains incremental. This reviewer does not give feedback. Score likely unchanged (4).

Reviewer UcWT raised foundational objections regarding DIG’s meaning, the loss function’s appropriateness, and fusion loss compatibility. Despite clarifications, the use of DIG remains empirically driven rather than theoretically grounded. This reviewer does not give feedback. Score likely unchanged (4).

---

### Decision · Program_Chairs · 2026-01-26

Reject